# Geometric Median (GM) Matching for
# Robust $k$-Subset Selection from Noisy Data

**Anish Acharya** [1 2]  **Sujay Sanghavi** [1 2]  **Alexandros G. Dimakis** [3 4]  **Inderjit S. Dhillon** [1 5]

## Abstract

Data pruning – the combinatorial task of selecting a small and representative subset from a large dataset, is crucial for mitigating the enormous computational costs associated with training data-hungry modern deep learning models at scale. Since large-scale data collections are invariably noisy, developing data pruning strategies that remain robust even in the presence of corruption is critical in practice. Existing data pruning methods often fail under high corruption rates due to their reliance on empirical mean estimation, which is highly sensitive to outliers. In response, this work proposes Geometric Median (GM) Matching, a novel k-subset selection strategy that leverages the GM, a robust estimator with an optimal breakdown point of 1/2; to enhance resilience against noisy data. Our method iteratively selects a $k$-subset such that the mean of the subset approximates the GM of the (potentially) noisy dataset, ensuring robustness even under arbitrary corruption. We provide theoretical guarantees, showing that GM MATCHING enjoys an improved $\mathcal{O}(1/k)$ convergence rate, outperforming $\mathcal{O}(1/\sqrt{k})$ scaling of uniform sampling, even under arbitrary corruption. Extensive experiments across image classification and image generation tasks demonstrate that GM MATCHING consistently outperforms existing pruning approaches, particularly in high-corruption settings; making it a strong baseline for robust data pruning.

## 1. Introduction

The recent success of deep learning has been largely fueled by the training of gigantic models on vast amounts of training data (Radford et al., 2018; 2021b; Brown et al., 2020; Touvron et al., 2023; Kaplan et al., 2020; Hestness et al., 2017). However, such large-scale training is usually associated with enormous computational costs, hindering the path to democratizing AI (Paul et al., 2021).

Data pruning – the combinatorial task of reducing a large training set into a small informative subset (Feldman, 2020; Agarwal et al., 2005; Muthukrishnan et al., 2005; Har-Peled, 2011; Feldman & Langberg, 2011), is a promising approach to reduce the enormous computational and storage costs of modern deep learning.

### 1.1. Robustness vs Diversity

Consequently, a large body of recent work has been proposed to solve the **combinatorial subset selection** problem. At a high level, these approaches rely on some carefully designed importance scoring criterion to rank the training samples, retaining a fraction of them as representative samples (super samples) used for training the downstream model. For example, spatial sampling approaches (Xia et al., 2022; Joshi & Mirzasoleiman, 2023; Sorscher et al., 2022; Needell et al., 2014) calculate the importance score of a sample in terms of the distance from the centroid of its corresponding class marginal. Samples closer to the centroid are considered the most prototypical (easy) and those far from the centroid are treated as least prototypical (hard). Canonical scoring criteria have also been developed in terms of gradients (Paul et al., 2021), uncertainty (Pleiss et al., 2020), and forgetfulness (Toneva et al., 2018). Notably, the spatial distance-based score is closely related to the gradient / uncertainty / forgetting-based score. Samples close (far away) to the class centroid are often associated with smaller (larger) gradient norm / lower (higher) uncertainty / harder (easier) to forget (Paul et al., 2021; Sorscher et al., 2022; Xia et al., 2022).

In the ideal scenario i.e. in the **absence of any corruption**, hard examples are known to contribute the most towards downstream generalization performance (Katharopoulos & Fleuret, 2018; Joshi et al., 2009; Huang et al., 2010; Balcan et al., 2007) as they often encode the most discriminative and task-relevant information in the dataset (Xu et al., 2020).

[1]University of Texas at Austin [2]Amazon [3]University of California, Berkeley [4]Bespoke Labs [5]Google. Correspondence to: Anish Acharya <anishacharya@utexas.edu>.

*Proceedings of the $42^{nd}$ International Conference on Machine Learning*, Vancouver, Canada. PMLR 267, 2025. Copyright 2025 by the author(s).

However, in the **realistic noisy scenarios involving outliers**, this strategy often fails since the noisy examples are wrongly deemed informative for training (Zhang & Sabuncu, 2018; Park et al., 2024). To mitigate this issue, existing pruning methods, tailored for such noisy scenarios, aim to retain the most prototypical / easy samples (Pleiss et al., 2020; Jiang et al., 2018; Har-Peled et al., 2007; Shah et al., 2020; Shen & Sanghavi, 2019). Yet, by prioritizing samples far from the decision boundary, these methods overlook less prototypical but uncorrupted and highly informative examples. This bias introduces a fundamental robustness vs. diversity trade-off (Xia et al., 2022), where *emphasizing robustness can lead to a lack of diversity* in the pruned subset. This limitation not only results in suboptimal generalization performance but, in extreme cases, can also lead to degenerate solutions (Sugiyama & Kawanabe, 2012).

Moreover, real-world scenarios frequently deviate from idealized assumptions, making it challenging or infeasible to adapt selection criteria and methodologies to diverse and unpredictable conditions. Consequently, despite its drawbacks in prioritizing informative examples, random sampling remains the industry standard due to its simplicity, efficiency, and ease of implementation. Due to space constraints, a more detailed related work is deferred to Appendix C.

### 1.2. Overview of Our Approach

To go beyond these limitations, we investigate the $k$-subset selection problem under the Gross Corruption Framework (Definition 1), where $0 \leq \psi < \frac{1}{2}$ fraction of the samples are allowed to be **arbitrarily perturbed**. This allowance for **arbitrary** corruption enables us to capture many practical robustness scenarios – including **corrupt features / labels** and **adversarial attacks**.

We make a key observation that: *existing pruning methods typically use the empirical mean to calculate the centroid of the samples, which then guides the selection process based on how representative those samples are*. However, the **empirical mean is highly susceptible to outliers** – in fact, it is possible to construct a single adversarial example to arbitrarily perturb the empirical mean (Figure 1).

As a consequence, in the presence of arbitrary corruption, the conventional distinction between easy (robust) and hard samples breaks down, leading to the selection of subsets that are significantly compromised by corruptions (2).

In response, we propose a novel subset selection strategy that fosters balanced diversity, effectively navigating various regions of the distribution while avoiding distant, noisy points. Specifically, we formulate the subset selection problem as one of minimizing the Maximum Mean Discrepancy (Gretton et al., 2012) between the empirical distribution induced by the selected subset and that of the underlying

---

**Algorithm 1 GEOMETRIC MEDIAN MATCHING**

(initialization)
A finite collection of grossly corrupted (Definition 1) observations $\mathcal{D} = \{\mathbf{x}_i \in \mathbb{R}^d\}_{i=1}^n$; pretrained encoder $\phi(\cdot) : \mathbb{R}^d \to \mathbb{R}^s$ e.g. CLIP (Radford et al., 2021b); initial weight vector $\boldsymbol{\theta}_0 \in \mathbb{R}^s$; number of sampling batches $B$, population fraction for GM computation $0 < \gamma_{\text{GM}} \leq 1$.

(compute embeddings)
$\Phi = \{\omega_i = \phi(\mathbf{x}_i) \in \mathbb{R}^s : \forall \mathbf{x}_i \in \mathcal{D}\}$
(pick random $n_{\text{GM}}$-subset for GM computation)
$\Phi_{\text{GM}} \overset{i.i.d}{\sim} \Phi, \quad \text{where,} |\Phi_{\text{GM}}| = \gamma_{\text{GM}}|\Phi| \leq n$
(compute $\epsilon$-approximate geometric median)
$\boldsymbol{\mu}_\epsilon^{\text{GM}}(\Phi_{\text{GM}}) = \arg\min_{\mathbf{z} \in \mathbb{R}^s} \sum_{\omega_i \in \Phi_{\text{GM}}} \|\mathbf{z} - \omega_i\|$
(partition data into batches)
$\mathcal{D} = \bigcup_{b=1}^B \mathcal{D}_b$
(initialize subset)
$\mathcal{D}_\mathcal{S} \leftarrow \emptyset$
**for** *batch index* $b = 1, \ldots B$ **do**
   (load batch embeddings)
   $\Phi_b = \{\omega_i \in \Phi : \mathbf{x}_i \in \mathcal{D}_b\}$
   **for** *iterations* $t = 0, 1, \ldots, k/B$ **do**
      (find embedding closest to $\boldsymbol{\theta}_t$)
      $\omega := \arg\max_{\omega_i \in \Phi_b} \langle \boldsymbol{\theta}_t, \omega_i \rangle$
      (update direction parameter)
      $\boldsymbol{\theta}_{t+1} := \boldsymbol{\theta}_t + \left[ \boldsymbol{\mu}_\epsilon^{\text{GM}}(\Phi_b) - \omega \right]$
      (update selected subset)
      $\mathcal{D}_\mathcal{S} := \mathcal{D}_\mathcal{S} \cup \mathbf{x}$ where, $\omega = \phi(\mathbf{x})$
      (update the batch embedding set)
      $\Phi_b := \Phi_b \setminus \omega$
   **end**
**end**
**return:** $\mathcal{D}_\mathcal{S}$

---

true (uncorrupted) distribution. Our key idea is to replace the empirical mean with a robust surrogate – Geometric Median (GM)(Definition 3) (Weber et al., 1929; Weiszfeld, 1937) – a classical estimator of the central tendency, inherently robust to outliers.

In particular, we optimize over finding a $k$-subset that minimizes the distance between the subset's empirical mean and the GM of the (potentially noisy) dataset over some appropriate embedding space, using herding (Welling, 2009) style greedy iterative updates. We call our algorithm Geometric Median (GM) Matching as described in Algorithm 1.

Intuitively, GM MATCHING can be viewed as a robust generalization of Kernel Herding (Chen et al., 2010). By replacing its vulnerable empirical mean with the GM, we inherit the favorable $\mathcal{O}(1/k)$ convergence of Kernel Herding while adding robustness to adversarial and heavy-tailed noise, bridging moment-matching and robust estimation in a unified framework.

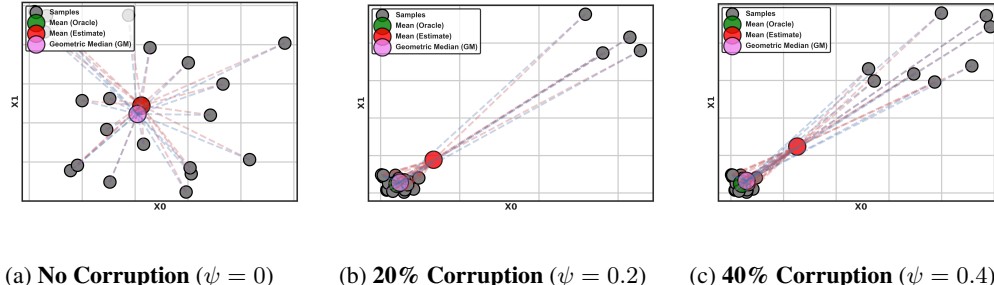

(a) **No Corruption** ($\psi = 0$)       (b) **20% Corruption** ($\psi = 0.2$)       (c) **40% Corruption** ($\psi = 0.4$)

Figure 1: ROBUST MEAN ESTIMATION: As the corruption rate $0 \leq \psi < \frac{1}{2}$ increases ( representing the fraction of samples drawn from an adversary-chosen distribution), the empirical mean increasingly deviates from the true uncorrupted mean. In contrast, the geometric median (GM) remains robust and stays closer to the uncorrupted (oracle) mean, demonstrating its resilience to outliers.

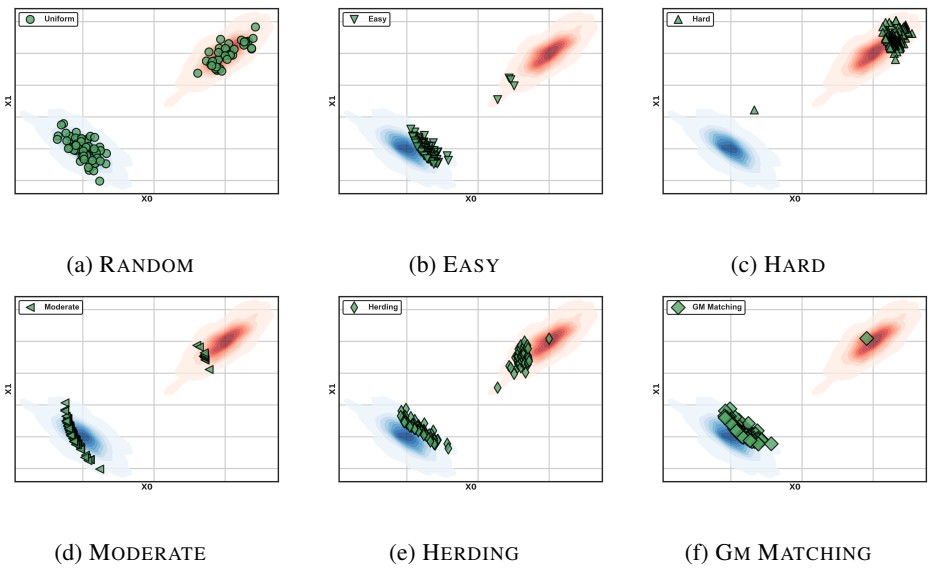

(a) RANDOM       (b) EASY       (c) HARD

(d) MODERATE       (e) HERDING       (f) GM MATCHING

Figure 2: DATA PRUNING IN THE WILD: subset 10% of the examples from anisotropic Gaussian (blue) where 40% replaced by an adversarial distribution (red). We compare GM MATCHING with (RANDOM), (EASY) samples closest to the centroid, (HARD) samples farthest from the centroid, (MODERATE) samples closest to the median distance from the centroid, and (KERNEL HERDING) (6). GM MATCHING yields significantly more robust subset than the other approaches.

### 1.3. Contributions

Overall, our contributions can be summarized as follows:

- We introduce a principled formulation of the $k$-subset selection problem under the Gross Corruption model (Definition 1), allowing up to 50% of the data to be arbitrarily corrupted. This generalizes prior noise models such as Huber contamination and Byzantine failure, and captures a wide range of real-world noise scenarios such as label noise, outliers, and adversarial attacks. To the best of our knowledge, this is the first theoretical and algorithmic treatment of robust data pruning under such a general corruption model.

- We demonstrate that state-of-the-art pruning strategies – including those designed for robustness – fail under gross corruption due to their reliance on the empirical mean, which has an asymptotic breakdown point of zero. Through both formal analysis (Lemma 2) and illustrative examples (Figure 1, Figure 8), we show how even a single adversarial point can arbitrarily skew moment-based selection methods.

- Motivated by this key observation, we propose a robust alternative: Geometric Median(GM) Matching. Our method selects a subset whose empirical mean approximates the Geometric Median (GM) of the full (potentially corrupted) dataset. We formalize this as a robust moment matching objective (6), and solve it via a greedy, herding (Chen et al., 2010) style iterative selection procedure (Algorithm 1). Unlike prior approaches, GM MATCHING leverages the optimal 1/2 breakdown point of GM, ensuring robustness even under adversarial perturbations.

- We provide rigorous theoretical analysis showing that

GM MATCHING converges to a bounded neighborhood of the true (uncorrupted) underlying mean at an $\mathcal{O}(1/k)$ rate (Theorem 1). This matches the best-known rates from Kernel Herding in clean settings, and crucially, continues to hold under arbitrary corruption. As a corollary, we derive bounds on the Maximum Mean Discrepancy (MMD) between the selected subset and the clean distribution (Lemma 1), establishing formal generalization guarantees.

- We also propose a practical, batched version of the algorithm, that enables efficient computation at scale as outlined in Algorithm 1. Our method incorporates two key engineering strategies: (i) sub-sampling for GM estimation, and (ii) batched greedy selection. We analyze the computational complexity and demonstrate that batching yields near-linear speedups without significantly compromising performance.

- We conduct comprehensive experiments across a range of tasks — including image classification, unsupervised distribution matching, and image generation. Our benchmarks cover diverse noise types: feature corruptions, label noise, and adversarial attacks. In all settings, the proposed approach consistently outperforms existing methods, especially under high corruption and aggressive pruning (often by >10%), establishing it as a state-of-the-art robust coreset selection strategy.

- Our analysis and visualizations (Figure 2) highlight how GM MATCHING balances robustness and diversity — avoiding the degeneracies of "easy-only" pruning while excluding adversarial outliers. This addresses a key open problem in the pruning literature: how to retain task-relevant examples near the decision boundary without being misled by corrupted points.

## 2. Problem Setup : Robust Data Pruning

Given a set of $n$ samples, data pruning (or coreset selection) aims to find a $k$-subset that is representative of the underlying distribution. If such a subset can be found compute-efficiently, then training a parametric model on the subset typically yields similar generalization performance as training on the entire dataset, while resulting in significant speedup when $k \ll n$.

**CORRUPTION MODEL:** However, real-world data is often noisy and imperfect due to the difficulty and expense of obtaining perfect semantic annotations, adversarial attacks, or simply measurement noise. To account for such realistic scenarios, we study the **combinatorial $k$-subset selection** problem under Gross Corruption Framework (Definition 1) (Diakonikolas et al., 2019) that generalizes both the Huber Contamination (Huber, 1992), and the Byzantine Failure model (Lamport et al., 1982).

**Definition 1** (**Gross Corruption**). *Given observations $\mathcal{D} =$*

$\{\mathbf{x}_i \in \mathbb{R}^d \overset{i.i.d}{\sim} p(x)\}_{i=1}^n$, *an adversary **inspects all the samples** and **arbitrarily perturbs** $\psi \in [0, \frac{1}{2})$ fraction of them. We will refer to such a set of samples $\mathcal{D} = \mathcal{D}_\mathcal{G} \cup \mathcal{D}_\mathcal{B}$ as $\psi$-corrupted , where $\mathcal{D}_\mathcal{B}, \mathcal{D}_\mathcal{G}$ denote the sets of corrupt and clean samples respectively.*

The goal of Robust Data Pruning is thus to select a $k$-subset $\mathcal{D}_\mathcal{S} \subseteq \mathcal{D}$ where $|\mathcal{D}_\mathcal{S}| = k$ that *encapsulates the underlying uncorrupted distribution $p(x)$ induced by subset $\mathcal{D}_\mathcal{G}$ without any a-priori knowledge about the corrupted samples*.

Let, $\hat{p}_\mathcal{S}$ denote the empirical measure induced by $\mathcal{D}_\mathcal{S}$. Then, we aim to solve:

$$\min_{\mathcal{D}_\mathcal{S} \subseteq \mathcal{D}} \Lambda\left(\hat{p}_\mathcal{S}(x) \,\Big\|\, p(x)\right) \tag{1}$$

for some appropriate divergence measure $\Lambda$. As discussed in Section 3, the proposed GM MATCHING (Algorithm 1) finds the subset by minimizing the **Maximum Mean Discrepancy (MMD)** (Gretton et al., 2012) between $\hat{p}_\mathcal{S}$ and the true underlying uncorrupted distribution $p$.

We measure the robustness of subset selection algorithms via breakdown point (Donoho & Huber, 1983; Lopuhaa et al., 1991; Davies & Gather, 2007) – a classic tool in robust optimization to assess the resilience of an estimator.

**Definition 2** (**Breakdown Point**). *The breakdown point $\varepsilon_T$ of an estimator $T(\cdot)$ is the smallest fraction of corrupted samples that can cause it to diverge arbitrarily:*

$$\zeta_T = \inf\left\{\psi \geq 0 : \sup_{\mathcal{D}_\mathcal{B}} \left\|T(\mathcal{D}) - T(\mathcal{D}_\mathcal{G})\right\| = \infty\right\} \tag{2}$$

$T(\cdot)$ achieves the **optimal breakdown point** $\zeta_T^* = 1/2$ if it remains bounded $\forall\, 0 \leq \psi < 1/2$.

**PROXY ENCODER:** Identifying sample importance is an ill-posed problem without some notion of similarity among the samples. Hence, we assume access to a *proxy encoder* $\phi : \mathbb{R}^d \to \mathcal{H}$ that maps raw inputs into a *separable embedding space* (potentially infinite-dimensional), *i.e.*, a *Reproducing Kernel Hilbert Space (RKHS)* $\mathcal{H}$ with inner product determined by a positive-definite kernel $\omega(\mathbf{x}, \mathbf{x}') = \langle \phi(\mathbf{x}), \phi(\mathbf{x}') \rangle_\mathcal{H}$ The RKHS structure ensures that inner products in $\mathcal{H}$ define a well-behaved notion of similarity i.e. semantically similar samples remain close while dissimilar ones are well separated. We further assume that $\phi(\cdot)$ is a characteristic feature map i.e. the mapping $p \mapsto \mu_p = \mathbb{E}_{\mathbf{x} \sim p}[\phi(\mathbf{x})]$ is *injective*. Simply put, for any two distributions $p$ and $q$: $\mu_p = \mu_q \implies p = q$. In practice, we instantiate such embeddings with pretrained *foundation models*, *e.g.*, CLIP encoders (Radford et al., 2021a) – explicitly trained via a contrastive objective (Chen et al., 2020) to map semantically similar examples closer together while pushing dissimilar ones apart.

| | | | CIFAR-100 | | | | |
|---|---|---|---|---|---|---|---|
| **Method / Ratio** | **20%** | **30%** | **40%** | **60%** | **80%** | **100%** | **Mean ↑** |
| Random | 50.26±3.24 | 53.61±2.73 | 64.32±1.77 | 71.03±0.75 | 74.12±0.56 | 78.14±0.55 | 62.67 |
| Herding | 48.39±1.42 | 50.89±0.97 | 62.99±0.61 | 70.61±0.44 | 74.21±0.49 | 78.14±0.55 | 61.42 |
| Forgetting | 35.57±1.40 | 49.83±0.91 | 59.65±2.50 | **73.34±0.39** | **77.50±0.53** | 78.14±0.55 | 59.18 |
| GraNd-score | 42.65±1.39 | 53.14±1.28 | 60.52±0.79 | 69.70±0.68 | 74.67±0.79 | 78.14±0.55 | 60.14 |
| EL2N-score | 27.32±1.16 | 41.98±0.54 | 50.47±1.20 | 69.23±1.00 | 75.96±0.88 | 78.14±0.55 | 52.99 |
| Optimization-based | 42.16±3.30 | 53.19±2.14 | 58.93±0.98 | 68.93±0.70 | 75.62±0.33 | 78.14±0.55 | 59.77 |
| Self-sup.-selection | 44.45±2.51 | 54.63±2.10 | 62.91±1.20 | 70.70±0.82 | 75.29±0.45 | 78.14±0.55 | 61.60 |
| Moderate-DS | 51.83±0.52 | 57.79±1.61 | 64.92±0.93 | 71.87±0.91 | 75.44±0.40 | 78.14±0.55 | 64.37 |
| **GM Matching** | **55.93± 0.48** | **63.08± 0.57** | **66.59± 1.18** | 70.82± 0.59 | 74.63± 0.86 | 78.14± 0.55 | **66.01** |
| | | | Tiny ImageNet | | | | |
| Random | 24.02±0.41 | 29.79±0.27 | 34.41±0.46 | 40.96±0.47 | 45.74±0.61 | 49.36±0.25 | 34.98 |
| Herding | 24.09±0.45 | 29.39±0.53 | 34.13±0.37 | 40.86±0.61 | 45.45±0.33 | 49.36±0.25 | 34.78 |
| Forgetting | 22.37±0.71 | 28.67±0.54 | 33.64±0.32 | 41.14±0.43 | **46.77±0.31** | 49.36±0.25 | 34.52 |
| GraNd-score | 23.56±0.52 | 29.66±0.37 | 34.33±0.50 | 40.77±0.42 | 45.96±0.56 | 49.36±0.25 | 34.86 |
| EL2N-score | 19.74±0.26 | 26.58±0.40 | 31.93±0.28 | 39.12±0.46 | 45.32±0.27 | 49.36±0.25 | 32.54 |
| Optimization-based | 13.88±2.17 | 23.75±1.62 | 29.77±0.94 | 37.05±2.81 | 43.76±1.50 | 49.36±0.25 | 29.64 |
| Self-sup.-selection | 20.89±0.42 | 27.66±0.50 | 32.50±0.30 | 39.64±0.39 | 44.94±0.34 | 49.36±0.25 | 33.13 |
| Moderate-DS | 25.29±0.38 | 30.57±0.20 | 34.81±0.51 | 41.45±0.44 | 46.06±0.33 | 49.36±0.25 | 35.64 |
| **GM Matching** | **27.88±0.19** | **33.15±0.26** | **36.92±0.40** | **42.48±0.12** | 46.75±0.51 | 49.36±0.25 | **37.44** |

Table 1: (CLEAN) IMAGE CLASSIFICATION: Comparing (Test Accuracy) pruning algorithms on CIFAR-100 and Tiny-ImageNet in the uncorrupted setting. ResNet-50 is used both as proxy and for downstream classification.

## 3. Geometric Median Matching

In the uncorrupted setting i.e. when $\psi = 0$, a natural and theoretically grounded approach for data pruning is to formulate it as a combinatorial MOMENT MATCHING objective:

$$\underset{\substack{\mathcal{D}_{\mathcal{S}} \subseteq \mathcal{D} \\ |\mathcal{D}_{\mathcal{S}}|=k}}{\arg\min} \left[ \Delta^2 := \left\| \frac{1}{|\mathcal{D}|} \sum_{\mathbf{x}_i \in \mathcal{D}} \phi(\mathbf{x}_i) - \frac{1}{k} \sum_{\mathbf{x}_i \in \mathcal{D}_{\mathcal{S}}} \phi(\mathbf{x}_i) \right\|^2 \right]$$
(3)

where the goal is to find a $k$-subset such that the empirical mean of the subset closely approximates that of the full dataset. Note that, (3) is an instance of the famous set function maximization problem – known to be NP-hard via a reduction from $k$-set cover (Feige, 1998). Remarkably, (Mirzasoleiman et al., 2020) demonstrated a transformation into a submodular set cover problem, allowing efficient solutions via greedy algorithms (Nemhauser et al., 1978). Despite strong theoretical guarantees in the uncorrupted setting, the moment matching objective can result in arbitrarily poor solutions under gross corruption (Definition 1) $\forall \psi > 0$. The vulnerability can be attributed to the estimation of target moment via empirical mean – notorious for its sensitivity to outliers. To illustrate, consider a single adversarial sample $\mathbf{x}^{\mathcal{B}} = \left( |\mathcal{D}| \boldsymbol{\mu}^{\mathcal{B}} - \sum_{\mathbf{x} \in \mathcal{D} \setminus \mathbf{x}^{\mathcal{B}}} \phi(\mathbf{x}) \right)$ forcing the empirical mean to an adversary chosen arbitrary target $\boldsymbol{\mu}^{\mathcal{B}}$. This implies that the empirical mean can't tolerate even a single grossly corrupted sample i.e. yields **lowest possible asymptotic breakdown point of 0**. Consequently, optimizing over (3) no longer guarantees convergence to the true underlying (uncorrupted) moment $\boldsymbol{\mu}^{\mathcal{G}} = \mathbb{E}_{\mathbf{x} \in \mathcal{D}_{\mathcal{G}}} \phi(\mathbf{x})$. Instead, the subset selection can be hijacked by a single bad sample, warping the solution towards an adversarial target.

ROBUST MOMENT MATCHING : Building on this key observation, we propose to solve a robust variant (6) of the moment matching objective (3) instead. *The key idea is to replace the empirical mean with a robust surrogate estimator of the target moment*, mitigating its susceptibility to corrupted samples.

A well-designed robust estimator $\tilde{\boldsymbol{\mu}}$ should guarantee that the estimation error, $\Delta = \|\tilde{\boldsymbol{\mu}} - \boldsymbol{\mu}^{\mathcal{G}}\| \leq \delta$ remains bounded, even under $\psi$-corruption (Definition 1). In this context, Geometric Median (GM) ( Definition 3) – a well-studied spatial estimator, known for several nice properties like rotation and translation invariance and **optimal breakdown point of 1/2** under gross corruption (Minsker et al., 2015; Kemperman, 1987) ( Figure 1,7,8). Moreover, GM is guaranteed to lie in the relative interior of the convex hull of the majority (good) points i.e. $\boldsymbol{\mu}^{\text{GM}} \in \mathcal{C}_{\mathcal{G}}$, making it an attractive choice to estimate the target moment.

**Definition 3** (**Geometric Median**). *Given a finite collection of observations $\{\phi(\mathbf{x}_1), \phi(\mathbf{x}_2), \ldots \phi(\mathbf{x}_n)\}$ defined over Hilbert space $\mathcal{H} \in \mathbb{R}^d$, equipped with norm $\|\cdot\|$ and inner $\langle\cdot\rangle$ operators, the geometric median (Fermat-Weber point) (Weber et al., 1929) $\boldsymbol{\mu}^{\text{GM}}$ is defined as:*

$$\boldsymbol{\mu}^{\text{GM}} = \underset{\mathbf{z} \in \mathcal{H}}{\arg\min} \left[ \rho(\mathbf{z}) := \sum_{i=1}^{n} \left\| \mathbf{z} - \phi(\mathbf{x}_i) \right\| \right] \quad (4)$$

Computing the GM exactly, is known to be algebraically intractable (Bajaj, 1988), making it necessary to rely on approximation methods to estimate the GM (Weiszfeld, 1937; Vardi & Zhang, 2000; Cohen et al., 2016). We call a point

| Method / Selection ratio | 20% | 30% | 40% | 60% | 80% | 100% | Mean ↑ |
|---|---|---|---|---|---|---|---|
| | | | CIFAR-100 with 20% corrupted images | | | | |
| Random | 40.99±1.46 | 50.38±1.39 | 57.24±0.65 | 65.21±1.31 | 71.74±0.28 | 74.92±0.88 | 57.11 |
| Herding | 44.42±0.46 | 53.57±0.31 | 60.72±1.78 | 69.09±1.73 | 73.08±0.98 | 74.92±0.88 | 60.18 |
| Forgetting | 26.39±0.17 | 40.78±2.02 | 49.95±2.31 | 65.71±1.12 | 73.67±1.12 | 74.92±0.88 | 51.30 |
| GraNd-score | 36.33±2.66 | 46.21±1.48 | 55.51±0.76 | 64.59±2.40 | 70.14±1.36 | 74.92±0.88 | 54.56 |
| EL2N-score | 21.64±2.03 | 23.78±1.66 | 35.71±1.17 | 56.32±0.86 | 69.66±0.43 | 74.92±0.88 | 41.42 |
| Optimization-based | 33.42±1.60 | 45.37±2.81 | 54.06±1.74 | 65.19±1.27 | 70.06±0.83 | 74.92±0.88 | 54.42 |
| Self-sup.-selection | 42.61±2.44 | 54.04±1.90 | 59.51±1.22 | 68.97±0.96 | 72.33±0.20 | 74.92±0.88 | 60.01 |
| Moderate-DS | 42.98±0.87 | 55.80±0.95 | 61.84±1.96 | 70.05±1.29 | 73.67±0.30 | 74.92±0.88 | 60.87 |
| **GM Matching** | **47.12±0.64** | **59.17±0.92** | **63.45±0.34** | **71.70±0.60** | **74.60±1.03** | 74.92±0.88 | **63.21** |
| | | | Tiny ImageNet with 20 % corrupted images | | | | |
| Random | 19.99±0.42 | 25.93±0.53 | 30.83±0.44 | 37.98±0.31 | 42.96±0.62 | 46.68±0.43 | 31.54 |
| Herding | 19.46±0.14 | 24.47±0.33 | 29.72±0.39 | 37.50±0.59 | 42.28±0.30 | 46.68±0.43 | 30.86 |
| Forgetting | 18.47±0.46 | 25.53±0.23 | 31.17±0.24 | 39.35±0.44 | 44.55±0.67 | 46.68±0.43 | 31.81 |
| GraNd-score | 20.07±0.49 | 26.68±0.40 | 31.25±0.40 | 38.21±0.49 | 42.84±0.72 | 46.68±0.43 | 30.53 |
| EL2N-score | 18.57±0.30 | 24.42±0.44 | 30.04±0.15 | 37.62±0.44 | 42.43±0.61 | 46.68±0.43 | 30.53 |
| Optimization-based | 13.71±0.26 | 23.33±1.84 | 29.15±2.84 | 36.12±1.86 | 42.94±0.52 | 46.88±0.43 | 29.06 |
| Self-sup.-selection | 20.22±0.23 | 26.90±0.50 | 31.93±0.49 | 39.74±0.52 | 44.27±0.10 | 46.68±0.43 | 32.61 |
| Moderate-DS | 23.27±0.33 | 29.06±0.36 | 33.48±0.11 | 40.07±0.36 | 44.73±0.39 | 46.68±0.43 | 34.12 |
| **GM Matching** | **27.19±0.92** | **31.70±0.78** | **35.14±0.19** | **42.04±0.31** | **45.12±0.28** | 46.68±0.43 | **36.24** |

Table 2: (**FEATURE CORRUPTION**) **IMAGE CLASSIFICATION:** Experiments comparing pruning methods when 20% of the images are corrupted. ResNet-50 is used for both proxy (data pruning) and downstream training.

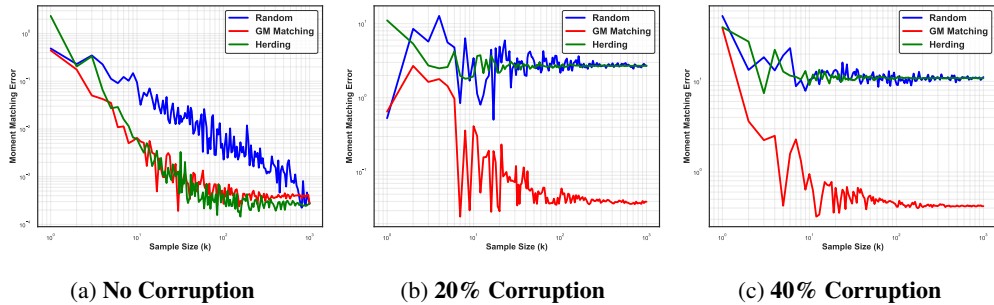

(a) **No Corruption**  (b) **20% Corruption**  (c) **40% Corruption**

Figure 3: **SAMPLING CONVERGENCE** (**RECOVERING ANISOTROPIC GAUSSIAN**): Comparison of convergence rates for Mean Discrepancy, defined as $\Delta^2 = \|\mu_\mathcal{G} - \mu_k\|^2$, as a function of sample size $k$ under varying corruption rates. Without corruption, both Herding and GM Matching achieve a quadratic improvement ($1/k$) over random sampling ($1/\sqrt{k}$). In the presence of corruption, GM Matching demonstrates superior robustness, maintaining convergence while Herding is significantly impacted.

$\boldsymbol{\mu}_\epsilon^{\mathrm{GM}} \in \mathcal{H}$ an $\epsilon$ accurate GM if:

$$\sum_{i=1}^n \left\| \boldsymbol{\mu}_\epsilon^{\mathrm{GM}} - \phi(\mathbf{x}_i) \right\| \leq (1+\epsilon) \sum_{i=1}^n \left\| \boldsymbol{\mu}^{\mathrm{GM}} - \phi(\mathbf{x}_i) \right\| \quad (5)$$

We adopt the popular Weiszfeld Algorithm ( Algorithm 2) for approximating the GM due of its efficiency, numerical stability and simplicity. We refer the reader to Appendix G, for more details and results on computing GM.

Leveraging the breakdown and translation invariance properties of GM we solve for the following objective:

$$\underset{\substack{\mathcal{D}_\mathcal{S} \subseteq \mathcal{D} \\ |\mathcal{D}_\mathcal{S}|=k}}{\arg\min} \left( \Delta_{\mathrm{GM}}^2 := \left\| \boldsymbol{\mu}_\epsilon^{\mathrm{GM}} - \frac{1}{k} \sum_{\mathbf{x}_i \in \mathcal{D}_\mathcal{S}} \phi(\mathbf{x}_i) \right\|^2 \right) \quad (6)$$

In other words, we aim to find a $k$-subset $\mathcal{D}_\mathcal{S}$ such that the empirical mean of the subset $\boldsymbol{\mu}_k = \frac{1}{k} \sum_{\mathbf{x}_i \in \mathcal{D}_\mathcal{S}} \phi(\mathbf{x}_i)$ approximately matches the GM of the input dataset $\mathcal{D}$.

Consequently, we perform herding style greedy minimization (Chen et al., 2010) of the error (6). Starting with a suitably chosen $\boldsymbol{\theta}_0 \in \mathcal{H}$; we repeatedly perform the following updates, adding one sample at a time, $k$ times:

$$\mathbf{x}_{t+1} := \underset{\mathbf{x} \in \mathcal{D}}{\arg\max} \, \langle \boldsymbol{\theta}_t, \phi(\mathbf{x}) \rangle \quad (7)$$

$$\boldsymbol{\theta}_{t+1} := \boldsymbol{\theta}_t + \left( \boldsymbol{\mu}_\epsilon^{\mathrm{GM}} - \phi(\mathbf{x}_{t+1}) \right) \quad (8)$$

We refer to the resulting robust data pruning approach as GM MATCHING (Algorithm 1).

### 3.1. Theoretical Analysis

Notably, GM MATCHING is an infinite memory, deterministic process as at any iteration $t = T$, $\boldsymbol{\theta}_T$ encapsulates the

entire sampling history:

$$\boldsymbol{\theta}_T = \boldsymbol{\theta}_0 + T\boldsymbol{\mu}_\epsilon^{\text{GM}} - \sum_{t=1}^{T} \phi(\mathbf{x}_t)$$

Conceptually, $\boldsymbol{\theta}_T$ *represents the vector pointing towards under-sampled regions of the target distribution induced by $\mathcal{D}$ at iteration $T$.* Greedy updates in the direction that reduces the accumulated error encourage the algorithm to explore underrepresented regions of the feature space, **promoting diversity**. By matching the GM rather than the empirical mean, the algorithm imposes larger penalties on outliers, which lie farther from the core distribution, **prioritizing samples near the convex hull of uncorrupted points** $\mathcal{C}_\mathcal{G} = \text{conv}\{\phi_\mathcal{B}(\mathbf{x}) | \mathbf{x} \in \mathcal{D}_\mathcal{G}\}$. As a result, the algorithm promotes diversity in a balanced manner, effectively exploring different regions of the distribution while avoiding distant, noisy points, thus **mitigating the robustness vs. diversity trade-off** discussed in Section 1.

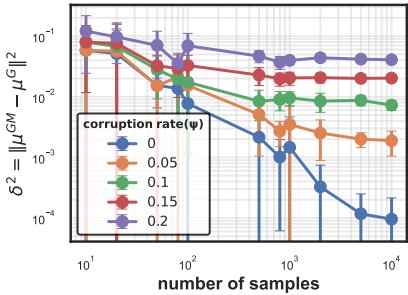

Figure 4: **GM SCALING (CONVERGENCE):** Comparison of Mean Estimation Error i.e. deviation of GM from the oracle mean $\|\boldsymbol{\mu}_\epsilon^{\text{GM}} - \boldsymbol{\mu}_\mathcal{G}\|^2$ w.r.t sample size $|\mathcal{D}|$ and corruption rate $\psi$.

**CONVERGENCE GUARANTEE:** Our convergence analysis leverages two key properties of GM – its robustness under gross corruption and the fact that it is guaranteed to lie in the interior of the convex hull of the majority (good) samples (Boyd & Vandenberghe, 2004). First, we exploit the robustness property of GM to get an upper bound on the estimation error w.r.t the underlying oracle mean (Acharya et al., 2022; Chen et al., 2017). Next, we use the property that GM is guaranteed to lie in the interior of the convex hull of the majority of the samples (Boyd & Vandenberghe, 2004). Combining these two results, we establish the following convergence guarantee:

**Theorem 1.** *Suppose that we are given a set of grossly corrupted samples $\mathcal{D} = \mathcal{D}_\mathcal{G} \cup \mathcal{D}_\mathcal{B}$ (Definition 1), an $\epsilon$ approx. $\text{GM}(\cdot)$ oracle (4) and characteristic feature map $\phi(\cdot) : \mathbb{R}^d \to \mathcal{H}$. Further assume that $\|\phi(\mathbf{x})\| \leq R \; \forall \mathbf{x} \in \mathcal{D}$ for some constant $R$. Then, GM MATCHING guarantees that the mean of the selected $k$-subset $\mathcal{D}_\mathcal{S} \subseteq \mathcal{D}$ converges to a $\delta$-neighborhood of the uncorrupted (true) mean*

$\boldsymbol{\mu}^\mathcal{G} = \mathbb{E}_{\mathbf{x} \in \mathcal{D}_\mathcal{G}}(\phi(\mathbf{x}))$ *at the rate $\mathcal{O}(1/k)$ such that:*

$$\delta^2 \leq \frac{8|\mathcal{D}_\mathcal{G}|^2}{(|\mathcal{D}_\mathcal{G}| - |\mathcal{D}_\mathcal{B}|)^2}\sigma_\mathcal{G}^2 + \frac{2\epsilon^2}{(|\mathcal{D}_\mathcal{G}| - |\mathcal{D}_\mathcal{B}|)^2} \quad (9)$$

*where, $\sigma_\mathcal{G}^2 = \frac{1}{|\mathcal{D}_\mathcal{G}|}\sum_{\mathbf{x} \in \mathcal{D}_\mathcal{G}} \mathbb{E}\|\phi(\mathbf{x}) - \boldsymbol{\mu}^\mathcal{G}\|^2$ denotes the variance of the uncorrupted samples.*

This result suggests that, even under gross corruption, GM MATCHING converges to a neighborhood of the true mean, where the neighborhood radius depends on two terms – the first term depends on the variance of the uncorrupted samples and the second term depends on how accurately the GM is calculated. Furthermore, the bound holds $\forall \alpha = \mathcal{D}_\mathcal{B}/\mathcal{D}_\mathcal{G} < 1$ implying GM Matching remains robust even when $1/2$ of the samples are arbitrarily corrupted, i.e. it achieves the optimal breakdown point of 1/2. Furthermore, GM MATCHING achieves a convergence rate of $\mathcal{O}(1/k)$ — a quadratic improvement over random sampling $\mathcal{O}(1/\sqrt{k})$. As a straightforward consequence of Theorem 1, we have:

**Lemma 1.**

$$\Delta^2 = \left\|\boldsymbol{\mu}(\mathcal{D}_\mathcal{S}) - \boldsymbol{\mu}(\mathcal{D}_\mathcal{G})\right\|^2 \leq \mathcal{O}\left(\frac{1}{k^2}\right)$$
$$+ \frac{16}{(1-\alpha)^2}\sigma_\mathcal{G}^2 + \frac{4\epsilon^2}{|\mathcal{D}_\mathcal{G}|^2(1-\alpha)^2} \quad (10)$$

By matching $\boldsymbol{\mu}(\mathcal{D}_\mathcal{S})$ to $\boldsymbol{\mu}(\mathcal{D}_\mathcal{G})$, we ensure that $\mathcal{D}_\mathcal{S}$ captures the uncorrupted distribution's first moment in the RKHS. Specifically, because $\phi(\cdot)$ is a *characteristic* feature map, bounding $\|\boldsymbol{\mu}(\mathcal{D}_\mathcal{S}) - \boldsymbol{\mu}(\mathcal{D}_\mathcal{G})\|$ immediately bounds $\Lambda_{\text{MMD}}(\hat{p}_\mathcal{S}, p)$. In turn, If the subset satisfies $\Delta \leq \delta$, then the expected risk difference between models trained on $\mathcal{D}_\mathcal{S}$ and $\mathcal{D}$ can be bounded as $\mathcal{O}(\delta^2)$, assuming the model class exhibits Lipschitz continuity in expectation (Shalev-Shwartz & Ben-David, 2014). Detailed proofs in Appendix E.

### 3.2. Computational Complexity Analysis

Ensuring computational efficiency is paramount in large-scale, real-world applications. Below, we outline the three major cost components in GM MATCHING:

**(A. Compute Embeddings)** For inputs divided into $L \approx \mathcal{O}(d)$ patches / tokens, the computational cost of computing CLIP ViT (Radford et al., 2021a) $\phi(\cdot) : \mathbb{R}^d \to \mathbb{R}^s$ embedding is $\approx \mathcal{O}(dh^2 + d^2h)$ where $h$ is the hidden size of the encoder. **(B. Compute GM)** Weiszfeld's algorithm(Algorithm 2) converges in the worst case as $\mathcal{O}(s/\epsilon)$ i.e. to compute $\boldsymbol{\mu}_\epsilon^{\text{GM}}$ over the embeddings incurs $\approx \mathcal{O}(ns/\epsilon)$. **(C. Sampling Iterations)** Each iteration of GM MATCHING computes inner products between all points in $\mathcal{D}$ and a direction vector $\boldsymbol{\theta} \in \mathbb{R}^s$ incurring $\approx \mathcal{O}(nks)$ compute. Additionally, storing the embeddings gives rise to

$\mathcal{O}(ns)$ space complexity. Thus, the overall time complexity can be expressed as:

$$T_{\text{GMM}} \approx \mathcal{O}(n(dh^2 + d^2h + s/\epsilon + ks))$$

To improve efficiency, we propose a two-fold batching strategy as outlined in Algorithm 1: **(1. Subsampling for GM)**: Instead of computing GM over the entire dataset, we compute it over a randomly chosen $\gamma_{\text{GM}}$-fraction $(0 < \gamma_{\text{GM}} \leq 1)$, reducing complexity to $\mathcal{O}(n\gamma_{\text{GM}}s/\epsilon)$. The deviation of GM from the true mean depends on the variance of the clean population (Theorem 1). In practice, for sufficiently large $\gamma_{\text{GM}}$, the variance increase is negligible compared to the savings (Figure 4). **(2. Batched GM MATCHING Iterations)**: Instead of selecting all $k$ samples at once, we divide the iterations into $B$ batches, selecting $k/B$ samples per batch. This results in a linear reduction of inner product computations per iteration, lowering the complexity to $\mathcal{O}(nsk/B)$.

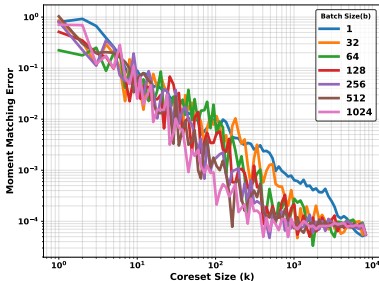

Figure 5: **GMM SCALING (CONVERGENCE):** Comparing moment matching error $\|\boldsymbol{\mu}_{\mathcal{G}} - \boldsymbol{\mu}_k\|^2$ of GM MATCHING across different batch size ($b$).

However, choosing too small of a batch size can increase variance and slow down convergence (Figure 5). In practice, with a moderate batch size, this additional error is often negligible, while the speedup is substantial. Incorporating these two ideas results in overall complexity:

$$\tilde{T}_{\text{GMM}} = \mathcal{O}\big(n(dh^2 + d^2h + ks/B + \gamma_{\text{GM}}s/\epsilon)\big)$$

We provide additional discussion and wall clock experiments in Appendix G.1,H.

## 4. Experiments

In this section, we outline our experimental setup, present our key empirical findings, and discuss deeper insights into the performance of GM MATCHING.

Due to space constraint, we only present a small subset of results in the main paper; a larger set of experiments and implementation details can be found in Appendix I.

### 4.1. Image Classification

To ensure reproducibility, our experimental setup is identical to (Xia et al., 2022). We compare GM MATCHING with the following baselines: (1) Random; (2) Herding (Welling, 2009); (3) Forgetting (Toneva et al., 2018); (4) GraNd-score (Paul et al., 2021); (5) EL2N-score (Paul et al., 2021); (6) Optimization-based (Yang et al., 2022); (7) Self-sup.-selection (Sorscher et al., 2022) and (8) Moderate (Xia et al., 2022) across three popular *Image Classification* datasets – Tiny-ImageNet, CIFAR10/100. Our experiments span popular deep nets including ResNet-18/50, VGG-16, ShuffleNet, SENet, EfficientNet-B0.

**IDEAL (NO CORRUPTION) SCENARIO:** Our first sets of experiments involve performing data pruning across selection ratio ranging from 20% - 80% in the uncorrupted setting. The corresponding results, presented in Table 1, indicate that while GM Matching is developed with robustness scenarios in mind, it outperforms the existing strong baselines even in the clean setting. Overall, on both CIFAR-100 and Tiny ImageNet, GM Matching improves over the prior methods > 2% on average. In particular, we note that GM Matching enjoys larger gains in the low data selection regime, while staying competitive at low pruning rates.

**CORRUPTION SCENARIOS:** To understand the performance of data pruning strategies in presence of corruption, we experiment with three different sources of corruption – image corruption, label noise and adversarial attacks.

**ROBUSTNESS TO IMAGE CORRUPTION:** In these experiments, the input images are corrupted – a popular robustness setting, often encountered when training models on real-world data (Hendrycks & Dietterich, 2019; Szegedy et al., 2013). To corrupt images, we apply: Gaussian noise, random occlusion, resolution reduction, fog, and motion blur to parts of the corrupt samples i.e. to say if $m$ samples are corrupted, each type of noise is added to one a random $m/5$ of them, while the other partitions are corrupted with a different noise. The results are presented in Table 2. We observe that GM Matching outperforms all the baselines across all pruning rates improving $\approx$3% across both datasets on an average. We note that, the gains are more consistent and profound in this setting than the clean setting.

**ROBUSTNESS TO LABEL CORRUPTION:** Next, we consider another important corruption scenario where a fraction of the training examples are mislabeled. We conduct experiments with synthetically injected symmetric label noise (Li et al., 2022; Patrini et al., 2017; Xia et al., 2020). The results are summarized in Table 3(Left). Encouragingly, GM Matching **outperforms the baselines by $\approx$ 12%**. Since, mislabeled samples come from different class - they tend to be spatially quite dissimilar, being less likely to be picked by GM matching, explaining the superior performance.

**ROBUSTNESS TO ADVERSARIAL ATTACKS:** Finally, we experiment with adversarial attacks that add imperceptible but adversarial noise on natural examples (Szegedy et al.,

| Method / Ratio | CIFAR-100 (PGD Attack) | | CIFAR-100 (GS Attack) | | Method / Ratio | CIFAR-100 (Label noise) | | Tiny ImageNet (Label noise) | |
| --- | --- | --- | --- | --- | --- | --- | --- | --- | --- |
| | 20% | 30% | 20% | 30% | | 20% | 30% | 20% | 30% |
| Random | 43.23±0.31 | 52.86±0.34 | 44.23±0.41 | 53.44±0.44 | Random | 34.47±0.64 | 43.26±1.21 | 17.78±0.44 | 23.88±0.42 |
| Herding | 40.21±0.72 | 49.62±0.65 | 39.92±1.03 | 50.14±0.15 | Herding | 42.29±1.75 | 50.52±3.38 | 18.98±0.44 | 24.23±0.29 |
| Forgetting | 35.90±1.30 | 47.37±0.99 | 37.55±0.53 | 46.88±1.91 | Forgetting | 36.53±1.11 | 45.78±1.04 | 13.20±0.38 | 21.79±0.43 |
| GraNd-score | 40.87±0.84 | 50.13±0.30 | 40.77±1.11 | 49.88±0.83 | GraNd-score | 31.72±0.67 | 42.80±0.30 | 18.28±0.32 | 23.72±0.18 |
| EL2N-score | 26.61±0.58 | 34.50±1.02 | 26.72±0.66 | 35.55±1.30 | EL2N-score | 29.82±1.19 | 33.62±2.35 | 13.93±0.69 | 18.57±0.31 |
| Optimization-based | 38.29±1.77 | 46.25±1.82 | 41.36±0.92 | 49.10±0.81 | Optimization-based | 32.79±0.62 | 41.80±1.14 | 14.77±0.95 | 22.52±0.77 |
| Self-sup.-selection | 40.53±1.15 | 49.95±0.50 | 40.74±1.66 | 51.23±0.25 | Self-sup.-selection | 31.08±0.78 | 41.87±0.63 | 15.10±0.73 | 21.01±0.36 |
| Moderate-DS | 43.60±0.97 | 51.66±0.39 | 44.69±0.68 | 53.71±0.37 | Moderate-DS | 40.25±0.12 | 48.53±1.60 | 19.64±0.40 | 24.96±0.30 |
| **GM Matching** | **45.41±0.86** | **51.80±1.01** | **49.78±0.27** | **55.50±0.31** | **GM Matching** | **52.64±0.72** | **61.01±0.47** | **25.80±0.37** | **31.71±0.24** |

Table 3: **(Left)** (LABEL NOISE) IMAGE CLASSIFICATION: Comparing (Test Accuracy) pruning methods on CIFAR-100 and TinyImageNet datasets, under 20% Symmetric Label Corruption, at 20% and 30% selection ratio. ResNet-50 is used both as proxy and for downstream classification. Side-by-side comparison of (Test Accuracy) pruning methods under adversarial attacks (left) and label noise (right) across CIFAR-100 and Tiny ImageNet datasets. ResNet-50 is used both as proxy and for downstream classification. **(Right)** (ADVERSARIAL ATTACK) IMAGE CLASSIFICATION: . Comparing (Test Accuracy) pruning methods under PGD and GS attacks. ResNet-50 is used both as proxy and for downstream classification.

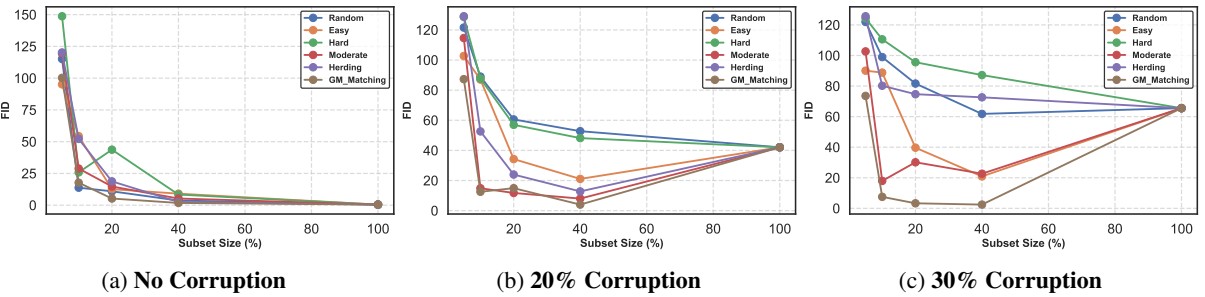

(a) **No Corruption**      (b) **20% Corruption**      (c) **30% Corruption**

Figure 6: DIFFUSION (FID): We train DDPM on MNIST across different sampling fraction and corruption rates using different sampling strategies. GM MATCHING consistently outperformed other approaches especially under corruption.

2013; Huang et al., 2010). Specifically, we employ two popular adversarial attack algorithms – PGD attack (Madry et al., 2017) and GS Attacks (Goodfellow et al., 2014) on models trained with CIFAR-100 and Tiny-ImageNet to generate adversarial examples. Following this, various pruning methods are applied to these adversarial examples, and the models are retrained on the curated subset of data. The results are summarized in Table 3(Right). Similar to other corruption scenarios, even in this setting, GM MATCHING outperforms the baselines yielding ≈ 3% average gain over the best performing baseline.

### 4.2. Unconditional Image Generation

To further validate our approach, we conduct experiments on an unconditional image generation task using a diffusion model. Specifically, we train a U-Net with Denoising Diffusion Probabilistic Models (DDPM) (Ho et al., 2020) on the MNIST dataset. Given the importance of selecting informative training samples, we compare our proposed geometric median-based sampling method (GM MATCHING) against multiple baseline selection strategies: (RANDOM) selection, (EASY) samples closest to the centroid, (HARD) samples farthest from the centroid, (MODERATE) samples with distances closest to the median, and (KERNEL HERDING) (Welling, 2009). To assess the resilience of different

selection methods, we introduce a range of structured perturbations in the training data, including Gaussian noise, uniform noise, random patches, Cutout augmentations, and entirely random images (Figure 18). We perform sample selection over CLIP ViT-B/32 (Radford et al., 2021a) embeddings (Figure 19). The performance of each sampling method is summarized (FID Score) in Figure 6, where GM MATCHING consistently outperforms other approaches by a significant margin. These findings underscore the effectiveness of geometric median-based selection in robustly curating training subsets for generative modeling.

Code is available publicly at Github.

## 5. Conclusion and Limitations

We introduced GM MATCHING, a novel data pruning strategy that ensures resilience under corruption, with theoretical and empirical validation. However, it relies on approximate GM computation, which can face challenges in degenerate cases. Additionally, its performance depends on the quality of the embedding space, which may be suboptimal in biased or poorly calibrated encoders. Addressing these challenges with efficient approximations and adaptive embedding strategies is a promising avenue for future work.

## Impact Statement

This work presents a method for robust data pruning, aimed at improving the efficiency and reliability of machine learning systems trained on large, potentially noisy datasets. By enabling models to learn from smaller, cleaner subsets, our approach reduces the computational burden and improves robustness – making it easier to train models in a data efficient manner, even in imperfect real world settings. The proposed method is task-agnostic and broadly applicable, with potential benefits across a wide array of domains — particularly where resilience to corrupted or adversarial data is critical. While the algorithm itself is neutral, any data pruning approach may risk amplifying biases if the embedding space or selection criteria reflect underlying societal or structural imbalances. We therefore encourage users to pair this method with fair, well-audited encoders and to critically evaluate the representativeness of selected subsets, especially in high-stakes applications.

Overall, this work contributes to a central goal in machine learning: learning effectively from imperfect data. We believe the potential benefits are significant, but we also emphasize the importance of careful, responsible deployment and continued ethical scrutiny.

## Acknowledgments

This work was supported by NSF Encore Tripods (2217069), NSF's AI Institute IFML (2019844) and grant from Meta.

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

# Supplementary Material for GM MATCHING

## Contents

# A. Notations and Abbreviations

| | |
|---|---|
| $a$ | A scalar (integer or real) |
| $\mathbf{a}$ | A vector |
| $\mathbf{A}$ | A matrix |
| $\mathrm{a}$ | A scalar random variable |
| $\mathbf{a}$ | A vector-valued random variable |
| $\mathbb{A}, \mathcal{A}$ | A set |
| $[a, b]$ | The real interval including $a$ and $b$ |
| $\mathbb{A} \backslash \mathbb{B}$ | Set subtraction, i.e., the set containing the elements of $\mathbb{A}$ that are not in $\mathbb{B}$ |
| $\mathrm{a}_i$ | Element $i$ of the random vector $\mathbf{a}$ |
| $P(\mathrm{a})$ | A probability distribution over a discrete variable |
| $p(\mathrm{a})$ | A probability distribution over a continuous variable, or over a variable whose type has not been specified |
| $f : \mathbb{A} \to \mathbb{B}$ | The function $f$ with domain $\mathbb{A}$ and range $\mathbb{B}$ |
| $f \circ g$ | Composition of the functions $f$ and $g$ |
| $f(\mathbf{x}; \boldsymbol{\theta})$ | A function of $\mathbf{x}$ parametrized by $\boldsymbol{\theta}$. (Sometimes we write $f(\mathbf{x})$ and omit the argument $\boldsymbol{\theta}$ to lighten notation) |
| $\|\mathbf{x}\|_p$ | $L^p$ norm of $\mathbf{x}$ |
| $\mathbf{1}(condition)$ | is 1 if the condition is true, 0 otherwise |
| GM MATCHING | Geometric Median Matching |

# B. Additional Definitions

**Definition 4** (**Multivariate Gaussian**). *A Multivariate Gaussian (or normal) distribution for a random vector $\mathbf{x} \in \mathbb{R}^d$ with mean $\boldsymbol{\mu}$ and covariance matrix $\boldsymbol{\Sigma}$ has the probability density function:*

$$p(\mathbf{x}) = \frac{1}{(2\pi)^{d/2}|\boldsymbol{\Sigma}|^{1/2}} \exp\left(-\frac{1}{2}(\mathbf{x} - \boldsymbol{\mu})^\top \boldsymbol{\Sigma}^{-1}(\mathbf{x} - \boldsymbol{\mu})\right) \tag{11}$$

**Definition 5** (**Isotropic Gaussian**). *A Gaussian distribution is said to be* isotropic *if its covariance matrix is a scalar multiple of the identity matrix:*

$$\boldsymbol{\Sigma} = \sigma^2 \mathbf{I} \tag{12}$$

*where, $\sigma^2 > 0$ is the variance (common to all dimensions) and $\mathbf{I}$ is the $d \times d$ identity matrix.*

*In this case, the density function simplifies to:*

$$p(\mathbf{x}) = \frac{1}{(2\pi\sigma^2)^{d/2}} \exp\left(-\frac{1}{2\sigma^2}\|\mathbf{x} - \boldsymbol{\mu}\|^2\right) \tag{13}$$

*Key characteristics:*

- *Equal variance in all directions: Every component of $\mathbf{x}$ has the same variance $\sigma^2$.*

- *No correlations: The off-diagonal elements of the covariance matrix are zero.*

- *Circular (or spherical in higher dimensions): Level sets of the density (contours) are circles (or spheres) centered at $\boldsymbol{\mu}$.*

**Definition 6** (**Anisotropic Gaussian**). *A Gaussian distribution is* anisotropic *if its covariance matrix is a general symmetric positive definite matrix that is not a scalar multiple of the identity:*

$$\mathbf{\Sigma} \neq \sigma^2 \mathbf{I} \tag{14}$$

*In this case, the full density function remains:*

$$p(\mathbf{x}) = \frac{1}{(2\pi)^{d/2}|\mathbf{\Sigma}|^{1/2}} \exp\left(-\frac{1}{2}(\mathbf{x} - \boldsymbol{\mu})^\top \mathbf{\Sigma}^{-1}(\mathbf{x} - \boldsymbol{\mu})\right). \tag{15}$$

*Key characteristics:*

- *Different variances along different axes: The eigenvalues of $\mathbf{\Sigma}$ determine the variance along the corresponding eigen-directions.*

- *Possible correlations: Off-diagonal entries may be nonzero, indicating correlations between components.*

- *Elliptical contours: The level sets of the density are ellipsoids, which can be elongated in some directions and compressed in others.*

**Definition 7** (**Discrete Derivative**). *For a set function $f : 2^{\mathbb{E}} \to \mathbb{R}$ , $\mathbb{S} \subseteq \mathbb{E}$ , $e \in \mathbb{E} \setminus \mathbb{S}$ the discrete derivative or the marginal gain of $f$ at $\mathbb{S}$ with respect to $e$ is defined as:*

$$\Delta_f(e|\mathbb{S}) = f(\mathbb{S} \cup \{e\}) - f(\mathbb{S}) \tag{16}$$

**Definition 8** (**Submodularity**). *A set function $f : 2^{\mathbb{E}} \to \mathbb{R}$ is submodular if $\forall \mathbb{A} \subseteq \mathbb{B} \subseteq \mathbb{E}$ and $e \in \mathbb{E} \setminus \mathbb{B}$ the following holds:*

$$\Delta_f(e|\mathbb{A}) \geq \Delta_f(e|\mathbb{B}) \tag{17}$$

*It can be shown that it is equivalent to the following condition:*

$$f(\mathbb{A}) + f(\mathbb{B}) \geq f(\mathbb{A} \cap \mathbb{B}) + f(\mathbb{A} \cup \mathbb{B}) \tag{18}$$

**Definition 9** (**Convex Hull**). *Given a set $\mathcal{S} \subseteq \mathbb{R}^d$, the* convex hull *of $\mathcal{S}$ is defined as the set of all convex combinations of points in $\mathcal{S}$, i.e.*

$$\text{conv}(\mathcal{S}) = \left\{ \sum_{i=1}^m \lambda_i \mathbf{x}_i : \mathbf{x}_i \in \mathcal{S}, \ \lambda_i \geq 0, \ \sum_{i=1}^m \lambda_i = 1, \ m \in \mathbb{N} \right\}.$$

## C. Related Work

A large body of recent works have been proposed to solve the data selection problem as detailed below:

IMPORTANCE SAMPLING :

One set of data pruning approaches rely on some carefully designed pruning metrics to rank the training samples based on the scores and retain a fraction of them as representative samples, used for training the downstream model. For example, (Xia et al., 2022; Joshi & Mirzasoleiman, 2023; Sorscher et al., 2022) calculate the importance score of a sample in terms of the distance from the centroid of its corresponding class marginal. Samples closer to the centroid are considered most prototypical (easy) and those far from the centroid are treated as least prototypical (hard). While this work primarily focuses on spatial approaches, it is worth mentioning that the canonical importance scoring criterion have been proposed in terms gradient norm (Paul et al., 2021; Needell et al., 2014), uncertainty (Pleiss et al., 2020; Garg & Roy, 2023) and forgetfulness (Toneva et al., 2018; Feldman & Zhang, 2020). Typically, samples closer to the class centroid in feature space tend to have lower gradient norms, exhibit lower uncertainty, and are harder to forget during training. In contrast, samples farther from the centroid generally have higher gradient norms, greater uncertainty, and are easier to forget (Paul et al., 2021; Sorscher et al., 2022; Xia et al., 2022). However, such scoring-based selection methods typically rely on the empirical mean

to define centroids, making them brittle under data corruption—a vulnerability that becomes critical in noisy or adversarial settings.

### MOMENT MATCHING METHODS :

A second family of methods tackles data selection as an optimization problem, aiming to match certain statistical moments—such as means, losses, or gradients—between the subset and the full dataset. These techniques, often rooted in coreset construction, select a subset whose empirical properties approximate those of the entire dataset (Chen et al., 2010; Campbell & Broderick, 2018; Dwivedi & Mackey, 2021). A notable extension is gradient-based moment matching (Mirzasoleiman et al., 2020), which seeks to preserve the full dataset's gradient dynamics within the selected subset, enabling efficient yet faithful training.

### SUBSET SELECTION FROM NOISY DATA :

Despite substantial advances in subset selection algorithms for large-scale datasets, most existing methods are designed for idealized, clean training data. In practical scenarios, however, real-world datasets are often noisy—affected by mislabeled examples, corrupted features, or adversarial attacks. Unfortunately, extending traditional subset selection methods to such noisy settings remains an under-explored challenge. A key limitation of many classical approaches lies in their dependence on the empirical mean—a statistic known to be highly sensitive to outliers and heavy-tailed distributions. As we formally demonstrate in Lemma 2, this renders them brittle under arbitrary corruption, where even a small fraction of adversarial points can severely skew the selection process. Consequently, any method that uses the empirical mean as a reference—either directly (e.g., centroid-based selection) or indirectly (e.g., gradient- or uncertainty-based ranking)—inherits the same fundamental vulnerability.

Several recent works have attempted to address robustness, but often in narrowly scoped settings. For instance, methods tailored to label noise (Pleiss et al., 2020; Park et al., 2024) typically rely on techniques such as relabeling or sample reweighting, assuming access to a relatively clean subset or side information. These strategies can be effective in controlled scenarios but do not generalize well to broader corruption types such as input perturbations, distribution shift, or structural outliers.

Another class of methods mitigate noise by aggressively pruning away uncertain or atypical examples—often selecting only the most prototypical or easy-to-learn samples (Shah et al., 2020; Toneva et al., 2018; Jiang et al., 2018; Har-Peled et al., 2007). While such selection improves robustness by avoiding noisy outliers, it inadvertently sacrifices diversity, discarding informative hard examples that are essential for learning decision boundaries. This introduces a well-known robustness–diversity tradeoff (Xia et al., 2022; Feldman & Zhang, 2020), where safe selection leads to suboptimal generalization. To balance this trade-off, Xia et al. (2022) proposed a moderation strategy that retains samples closest to the median distance from the class centroid—aiming to avoid both overly hard and overly easy examples. However, this approach still fundamentally relies on spatial distances measured from the empirical mean, making it vulnerable to corruption that distorts this reference point. Furthermore, while a few recent works have studied robust subset selection in a theoretical manner, they are either limited to linear models(Xu et al., 2025) or assume simplified theoretical frameworks(Thompson, 2022; Park et al., 2024), strong assumptions (Qian et al., 2017). These works stop short of providing practical algorithms or generalization guarantees applicable to modern, highly nonlinear deep learning systems.

In contrast, our work introduces a simple, theoretically grounded algorithm for subset selection under arbitrary data corruption, bridging the gap between robustness and diversity. GM MATCHING generalizes to deep models and real-world noisy data, offering provable guarantees under the Gross Corruption.

### CONNECTION TO KERNEL HERDING :

GM MATCHING builds upon the classical Kernel Herding framework (Welling, 2009; Chen & Welling, 2010; Bach et al., 2012), which selects representative subsets that match the mean embedding of the data distribution in a reproducing kernel Hilbert space (RKHS). While Kernel Herding offers favorable convergence guarantees $\mathcal{O}(1/k)$ vs. $\mathcal{O}(1/\sqrt{k})$ for random sampling; it relies on the empirical mean, making it vulnerable to outliers and adversarial corruption. GM MATCHING preserves the moment-matching spirit of Kernel Herding but replaces the empirical mean with the geometric median (GM), a robust estimator with a breakdown point of 1/2. This simple substitution significantly improves robustness without compromising convergence. To our knowledge, GM MATCHING is the first method to combine robust estimation with herding-style greedy selection, enabling provably fast and stable subset selection even under high rates of data corruption.

## D. Lemma 2 : Vulnerability of Importance Score based Pruning

In the ideal setting, given a batch of i.i.d samples $\boldsymbol{\mu}_y = \boldsymbol{\mu}_y^{\mathcal{G}} = \mathbb{E}_{\mathbf{x} \sim \mathcal{D}_{\mathcal{G}}}(\mathbf{x})$. However, the presence of even a single grossly corrupted sample can cause the centroid estimate to deviate arbitrarily from the true mean. Consider a single grossly corrupt sample $(\mathbf{x}_i^{\mathcal{B}}, y_i)$ such that :

$$\mathbf{x}_i^{\mathcal{B}} = \sum_{(\mathbf{x}_i, y_i) \in \mathcal{D}} \mathbf{1}(y_i = y)\boldsymbol{\mu}_y^{\mathcal{B}} - \sum_{(\mathbf{x}_i, y_i) \in \mathcal{D} \setminus (\mathbf{x}_i^{\mathcal{B}}, y_i)} \mathbf{1}(y_i = y)\mathbf{x}_i \tag{19}$$

resulting in shifting the estimated centroid $\Delta\boldsymbol{\mu}_y = \boldsymbol{\mu}_y^{\mathcal{B}} - \boldsymbol{\mu}_y^{\mathcal{G}}$

**Lemma 2.** *A single gross corrupted sample* (19) *causes the importance scores to deviate arbitrarily:*

$$\Delta d(\mathbf{x}_i, y_i) = \|\Delta\boldsymbol{\mu}_y\|^2 - 2\left(\mathbf{x}_i - \boldsymbol{\mu}_y^{\mathcal{G}}\right)^T \Delta\boldsymbol{\mu}_y \tag{20}$$

*Implying, these methods yield the* **lowest possible asymptotic breakdown of 0**.

### D.1. Proof of Lemma 2

*Proof.* The original importance score without the corrupted sample is:

$$d(\mathbf{x}_i, y_i) = \|\mathbf{x}_i - \mu_y^{\mathcal{G}}\|_2^2 \tag{21}$$

The importance score with the corrupted sample affecting the centroid is:

$$d'(\mathbf{x}_i, y_i) = \|\mathbf{x}_i - \mu_y^{\mathcal{B}}\|_2^2 \tag{22}$$

We can calculate the deviation as:

$$\Delta d(\mathbf{x}_i, y_i) = d(\mathbf{x}_i, y_i) - d'(\mathbf{x}_i, y_i)$$
$$= \left(\mathbf{x}_i - \mu_y^{\mathcal{B}}\right)^T \left(\mathbf{x}_i - \mu_y^{\mathcal{B}}\right) - \left(\mathbf{x}_i - \mu_y^{\mathcal{G}}\right)^T \left(\mathbf{x}_i - \mu_y^{\mathcal{G}}\right)$$

The result follows by expanding and defining $\Delta\mu_y = \mu_y^{\mathcal{B}} - \mu_y^{\mathcal{G}}$ ∎

## E. Proof of Theorem 1

We restate the theorem for convenience:

**Theorem** 1 Given a set of Grossly-corrupted samples $\mathcal{D} = \mathcal{D}_{\mathcal{G}} \cup \mathcal{D}_{\mathcal{B}}$ (Definition 1) and an $\epsilon$ approx. GM$(\cdot)$ oracle (4). Further assume that $\|\mathbf{x}\| \leq R \ \forall \mathbf{x} \in \mathcal{D}$ for some constant $R$. Then, GM MATCHING guarantees that the mean of the selected $k$-subset $\mathcal{D}_{\mathcal{S}} \subseteq \mathcal{D}$ converges to a $\delta$-neighborhood of the uncorrupted (true) mean $\boldsymbol{\mu}(\mathcal{D}_{\mathcal{G}}) = \mathbb{E}_{\mathbf{x} \in \mathcal{D}_{\mathcal{G}}}(\mathbf{x})$ at the rate $\mathcal{O}(\frac{1}{k})$ such that:

$$\delta^2 = \mathbb{E}\left\|\boldsymbol{\mu}_{\epsilon}^{\mathrm{GM}}(\mathcal{D}) - \boldsymbol{\mu}(\mathcal{D}_{\mathcal{G}})\right\|^2 \leq \frac{8|\mathcal{D}_{\mathcal{G}}|}{(|\mathcal{D}_{\mathcal{G}}| - |\mathcal{D}_{\mathcal{B}}|)^2} \sum_{\mathbf{x} \in \mathcal{D}_{\mathcal{G}}} \mathbb{E}\left\|\mathbf{x} - \boldsymbol{\mu}(\mathcal{D}_{\mathcal{G}})\right\|^2 + \frac{2\epsilon^2}{(|\mathcal{D}_{\mathcal{G}}| - |\mathcal{D}_{\mathcal{B}}|)^2} \tag{23}$$

*Proof.* For notational simplicity, WLOG we will assume that the samples are already in RKHS i.e. we drop the notation $\phi(\cdot)$ for the proof i.e. we assume that $\mathbf{x}_i$ is already projected on the positive definite kernel space using a characteristic mapping $\phi(\cdot)$. We prove Theorem 1 in two main steps:

### E.1. Bounding Estimation Error from Approximate Geometric Median

We will first establish the following result which follows from the definition of GM; see also (Lopuhaa et al., 1991; Minsker et al., 2015; Cohen et al., 2016; Chen et al., 2017; Li et al., 2019; Wu et al., 2020; Acharya et al., 2022) for similar adaptations.

**Lemma 3.** *Given a set of $\alpha$-corrupted samples $\mathcal{D} = \mathcal{D}_\mathcal{G} \cup \mathcal{D}_\mathcal{B}$ ( Definition 1), and an $\epsilon$-approx. $\mathrm{GM}(\cdot)$ oracle (4), then we have:*

$$\mathbb{E}\left\|\boldsymbol{\mu}_\epsilon^{\mathrm{GM}}(\mathcal{D}) - \boldsymbol{\mu}(\mathcal{D}_\mathcal{G})\right\|^2 \leq \frac{8|\mathcal{D}_\mathcal{G}|}{(|\mathcal{D}_\mathcal{G}| - |\mathcal{D}_\mathcal{B}|)^2} \sum_{\mathbf{x} \in \mathcal{D}_\mathcal{G}} \mathbb{E}\left\|\mathbf{x} - \boldsymbol{\mu}(\mathcal{D}_\mathcal{G})\right\|^2 + \frac{2\epsilon^2}{(|\mathcal{D}_\mathcal{G}| - |\mathcal{D}_\mathcal{B}|)^2} \tag{24}$$

*where, $\boldsymbol{\mu}_\epsilon^{\mathrm{GM}}(\mathcal{D})$ is the $\epsilon$-approximate $\mathrm{GM}$ over the entire ($\alpha$-corrupted) dataset; and $\boldsymbol{\mu}(\mathcal{D}_\mathcal{G}) = \frac{1}{|\mathcal{D}_\mathcal{G}|} \sum_{\mathbf{x}_i \in \mathcal{D}_\mathcal{G}} \mathbf{x}_i$ denotes the mean of the (underlying) uncorrupted set.*

Now we prove this bound:

Note that, by using the triangle inequality, we can write:

$$\sum_{\mathbf{x}_i \in \mathcal{D}} \left\|\boldsymbol{\mu}_\epsilon^{\mathrm{GM}}(\mathcal{D}) - \mathbf{x}_i\right\| \geq \sum_{\mathbf{x}_i \in \mathcal{D}_\mathcal{B}} \left(\left\|\mathbf{x}_i\right\| - \left\|\boldsymbol{\mu}_\epsilon^{\mathrm{GM}}(\mathcal{D})\right\|\right) + \sum_{\mathbf{x}_i \in \mathcal{D}_\mathcal{G}} \left(\left\|\boldsymbol{\mu}_\epsilon^{\mathrm{GM}}(\mathcal{D})\right\| - \left\|\mathbf{x}_i\right\|\right) \tag{25}$$

$$= \left(\sum_{\mathbf{x}_i \in \mathcal{D}_\mathcal{G}} - \sum_{\mathbf{x}_i \in \mathcal{D}_\mathcal{B}}\right)\left\|\boldsymbol{\mu}_\epsilon^{\mathrm{GM}}(\mathcal{D})\right\| + \sum_{\mathbf{x}_i \in \mathcal{D}_\mathcal{B}} \left\|\mathbf{x}_i\right\| - \sum_{\mathbf{x}_i \in \mathcal{D}_\mathcal{G}} \left\|\mathbf{x}_i\right\| \tag{26}$$

$$= \left(|\mathcal{D}_\mathcal{G}| - |\mathcal{D}_\mathcal{B}|\right)\left\|\boldsymbol{\mu}_\epsilon^{\mathrm{GM}}(\mathcal{D})\right\| + \sum_{\mathbf{x}_i \in \mathcal{D}} \left\|\mathbf{x}_i\right\| - 2\sum_{\mathbf{x}_i \in \mathcal{D}_\mathcal{G}} \left\|\mathbf{x}_i\right\|. \tag{27}$$

Now, by definition (5); we have that:

$$\sum_{\mathbf{x}_i \in \mathcal{D}} \left\|\boldsymbol{\mu}_\epsilon^{\mathrm{GM}}(\mathcal{D}) - \mathbf{x}_i\right\| \leq \inf_{\mathbf{z} \in \mathcal{H}} \sum_{\mathbf{x}_i \in \mathcal{D}} \left\|\mathbf{z} - \mathbf{x}_i\right\| + \epsilon \leq \sum_{\mathbf{x}_i \in \mathcal{D}} \left\|\mathbf{x}_i\right\| + \epsilon \tag{28}$$

Combining these two inequalities, we get:

$$\left(|\mathcal{D}_\mathcal{G}| - |\mathcal{D}_\mathcal{B}|\right)\left\|\boldsymbol{\mu}_\epsilon^{\mathrm{GM}}(\mathcal{D})\right\| \leq \sum_{\mathbf{x}_i \in \mathcal{D}} \left\|\mathbf{x}_i\right\| - \sum_{\mathbf{x}_i \in \mathcal{D}} \left\|\mathbf{x}_i\right\| + 2\sum_{\mathbf{x}_i \in \mathcal{D}_\mathcal{G}} \left\|\mathbf{x}_i\right\| + \epsilon \tag{29}$$

This implies:

$$\left\|\boldsymbol{\mu}_\epsilon^{\mathrm{GM}}(\mathcal{D})\right\| \leq \frac{2}{\left(|\mathcal{D}_\mathcal{G}| - |\mathcal{D}_\mathcal{B}|\right)} \sum_{\mathbf{x}_i \in \mathcal{D}_\mathcal{G}} \left\|\mathbf{x}_i\right\| + \frac{\epsilon}{\left(|\mathcal{D}_\mathcal{G}| - |\mathcal{D}_\mathcal{B}|\right)} \tag{30}$$

Squaring both sides,

$$\left\|\boldsymbol{\mu}_\epsilon^{\mathrm{GM}}(\mathcal{D})\right\|^2 \leq \left[\frac{2}{\left(|\mathcal{D}_\mathcal{G}| - |\mathcal{D}_\mathcal{B}|\right)} \sum_{\mathbf{x}_i \in \mathcal{D}_\mathcal{G}} \left\|\mathbf{x}_i\right\| + \frac{\epsilon}{\left(|\mathcal{D}_\mathcal{G}| - |\mathcal{D}_\mathcal{B}|\right)}\right]^2 \tag{31}$$

$$\leq 2\left[\frac{2}{\left(|\mathcal{D}_\mathcal{G}| - |\mathcal{D}_\mathcal{B}|\right)} \sum_{\mathbf{x}_i \in \mathcal{D}_\mathcal{G}} \left\|\mathbf{x}_i\right\|\right]^2 + 2\left[\frac{\epsilon}{\left(|\mathcal{D}_\mathcal{G}| - |\mathcal{D}_\mathcal{B}|\right)}\right]^2 \tag{32}$$

Where the last step is a well-known consequence of the triangle inequality and AM-GM inequality.

Taking expectation on both sides, we have:

$$\mathbb{E}\left\|\boldsymbol{\mu}_\epsilon^{\mathrm{GM}}(\mathcal{D})\right\|^2 \leq \frac{8|\mathcal{D}_\mathcal{G}|}{\left(|\mathcal{D}_\mathcal{G}| - |\mathcal{D}_\mathcal{B}|\right)^2} \sum_{\mathbf{x}_i \in \mathcal{D}_\mathcal{G}} \mathbb{E}\left\|\mathbf{x}_i\right\|^2 + \frac{2\epsilon^2}{\left(|\mathcal{D}_\mathcal{G}| - |\mathcal{D}_\mathcal{B}|\right)^2} \tag{33}$$

Since, GM is **translation equivariant**, we can write:

$$\mathbb{E}\left[\text{GM}\left(\left\{\mathbf{x}_i - \boldsymbol{\mu}(\mathcal{D}_{\mathcal{G}})|\mathbf{x}_i \in \mathcal{D}\right\}\right)\right] = \mathbb{E}\left[\text{GM}\left(\left\{\mathbf{x}_i|\mathbf{x}_i \in \mathcal{D}\right\}\right) - \boldsymbol{\mu}(\mathcal{D}_{\mathcal{G}})\right] \tag{34}$$

Consequently, we have that :

$$\mathbb{E}\left\|\boldsymbol{\mu}_\epsilon^{\text{GM}}(\mathcal{D}) - \boldsymbol{\mu}(\mathcal{D}_{\mathcal{G}})\right\|^2 \le \frac{8|\mathcal{D}_{\mathcal{G}}|}{\left(|\mathcal{D}_{\mathcal{G}}| - |\mathcal{D}_{\mathcal{B}}|\right)^2} \sum_{\mathbf{x}_i \in \mathcal{D}_{\mathcal{G}}} \mathbb{E}\left\|\mathbf{x}_i - \boldsymbol{\mu}(\mathcal{D}_{\mathcal{G}})\right\|^2 + \frac{2\epsilon^2}{\left(|\mathcal{D}_{\mathcal{G}}| - |\mathcal{D}_{\mathcal{B}}|\right)^2}$$

This concludes the proof of Lemma 3

### E.2. Convergence of the Greedy Updates

We will now show that GM MATCHING converges to $\boldsymbol{\mu}_\epsilon^{\text{GM}}(\mathcal{D})$ at $\mathcal{O}(\frac{1}{k})$.

It suffices to show that the error $\delta = \left\|\boldsymbol{\mu}_\epsilon^{\text{GM}}(\mathcal{D}) - \frac{1}{k}\sum_{\mathbf{x}_i \in \mathcal{S}} \mathbf{x}_i\right\| \to 0$ asymptotically. We will follow the proof technique in (Chen et al., 2010) mutatis-mutandis to prove this result. We also assume that $\mathcal{D}$ contains the support of the resulting noisy distribution.

We start by defining a GM-centered marginal polytope as the convex hull –

$$\mathcal{M}_\epsilon := \text{conv}\left\{\mathbf{x} - \boldsymbol{\mu}_\epsilon^{\text{GM}}(\mathcal{D}) \,|\mathbf{x} \in \mathcal{D}\right\} \tag{35}$$

Then, we can rewrite the update equation (8) as:

$$\boldsymbol{\theta}_{t+1} = \boldsymbol{\theta}_t + \boldsymbol{\mu}_\epsilon^{\text{GM}}(\mathcal{D}) - \mathbf{x}_{t+1} \tag{36}$$

$$= \boldsymbol{\theta}_t - (\mathbf{x}_{t+1} - \boldsymbol{\mu}_\epsilon^{\text{GM}}(\mathcal{D}) \tag{37}$$

$$= \boldsymbol{\theta}_t - \left(\arg\max_{\mathbf{x}\in\mathcal{D}}\langle\boldsymbol{\theta}_t, \mathbf{x}\rangle - \boldsymbol{\mu}_\epsilon^{\text{GM}}(\mathcal{D})\right) \tag{38}$$

$$= \boldsymbol{\theta}_t - \arg\max_{\mathbf{m}\in\mathcal{M}_\epsilon}\langle\boldsymbol{\theta}_t, \mathbf{m}\rangle \tag{39}$$

$$= \boldsymbol{\theta}_t - \mathbf{m}_t \tag{40}$$

Now, squaring both sides we get :

$$\|\boldsymbol{\theta}_{t+1}\|^2 = \|\theta_t\|^2 + \|\mathbf{m}_t\|^2 - 2\langle\boldsymbol{\theta}_t, \mathbf{m}_t\rangle \tag{41}$$

Rearranging the terms we get:

$$\|\boldsymbol{\theta}_{t+1}\|^2 - \|\theta_t\|^2 = \|\mathbf{m}_t\|^2 - 2\langle\boldsymbol{\theta}_t, \mathbf{m}_t\rangle \tag{42}$$

$$= \|\mathbf{m}_t\|^2 - 2\|\mathbf{m}_t\|\|\boldsymbol{\theta}_t\|\langle\frac{\boldsymbol{\theta}_t}{\|\boldsymbol{\theta}_t\|}, \frac{\mathbf{m}_t}{\|\mathbf{m}_t\|}\rangle \tag{43}$$

$$= 2\|\mathbf{m}_t\|\left(\frac{1}{2}\|\mathbf{m}_t\| - \|\boldsymbol{\theta}_t\|\langle\frac{\boldsymbol{\theta}_t}{\|\boldsymbol{\theta}_t\|}, \frac{\mathbf{m}_t}{\|\mathbf{m}_t\|}\rangle\right) \tag{44}$$

Assume that $\|\mathbf{x}_i\| \le r \,\forall \mathbf{x}_i \in \mathcal{D}$. Then we note that,

$$\|\mathbf{x}_i - \boldsymbol{\mu}_\epsilon^{\text{GM}}(\mathcal{D})\| \le \|\mathbf{x}_i\| + \|\boldsymbol{\mu}_\epsilon^{\text{GM}}(\mathcal{D})\| \le 2r$$

Plugging this in, we get:

$$\|\boldsymbol{\theta}_{t+1}\|^2 - \|\theta_t\|^2 \leq 2\|\mathbf{m}_t\| \left( r - \|\boldsymbol{\theta}_t\| \langle \frac{\boldsymbol{\theta}_t}{\|\boldsymbol{\theta}_t\|}, \frac{\mathbf{m}_t}{\|\mathbf{m}_t\|} \rangle \right) \tag{45}$$

Recall that, $\boldsymbol{\mu}_\epsilon^{\text{GM}}(\mathcal{D})$ is guaranteed to be in the relative interior of $\text{conv}\{\mathbf{x} \,|\, \mathbf{x} \in \mathcal{D}\}$ (Lopuhaa et al., 1991; Minsker et al., 2015). Consequently, $\exists \kappa$-ball around $\boldsymbol{\mu}_\epsilon^{\text{GM}}(\mathcal{D})$ contained inside $\mathcal{M}$ and we have $\forall t > 0$

$$\langle \frac{\boldsymbol{\theta}_t}{\|\boldsymbol{\theta}_t\|}, \frac{\mathbf{m}_t}{\|\mathbf{m}_t\|} \rangle \geq \kappa > 0 \tag{46}$$

This implies, $\forall t > 0$

$$\|\boldsymbol{\theta}_t\| \leq \frac{r}{\kappa} \tag{47}$$

Expanding the value of $\boldsymbol{\theta}_t$, we have:

$$\|\boldsymbol{\theta}_k\| = \left\| \boldsymbol{\theta}_0 + k\boldsymbol{\mu}_\epsilon^{\text{GM}}(\mathcal{D}) - \sum_{i=1}^{k} \mathbf{x}_k \right\| \leq \frac{r}{\kappa} \tag{48}$$

Apply Cauchy-Schwarz inequality:

$$\left\| k\boldsymbol{\mu}_\epsilon^{\text{GM}}(\mathcal{D}) - \sum_{i=1}^{k} \mathbf{x}_k \right\| \leq \|\boldsymbol{\theta}_0\| + \frac{r}{\kappa} \tag{49}$$

Normalizing both sides by the number of iterations $k$

$$\left\| \boldsymbol{\mu}_\epsilon^{\text{GM}}(\mathcal{D}) - \frac{1}{k} \sum_{i=1}^{k} \mathbf{x}_k \right\| \leq \frac{1}{k} \left( \|\boldsymbol{\theta}_0\| + \frac{r}{\kappa} \right) \tag{50}$$

Thus, we have that GM MATCHING converges to $\boldsymbol{\mu}_\epsilon^{\text{GM}}$ at the rate $\mathcal{O}(\frac{1}{k})$.

Combining this with Lemma 3, completes the proof of Theorem 1. ∎

## F. Proof of Lemma 1

We restate the lemma for convenience:

**Lemma** 1: Suppose that we are given a set of grossly corrupted samples $\mathcal{D} = \mathcal{D}_\mathcal{G} \cup \mathcal{D}_\mathcal{B}$ (Definition 1), an $\epsilon$ approx. GM$(\cdot)$ oracle (4) and bounded, characteristic feature map $\phi(\cdot) : \mathbb{R}^d \rightarrow \mathcal{H}$. Then, GM MATCHING guarantees that:

$$\Delta^2 = \left\| \boldsymbol{\mu}(\mathcal{D}_\mathcal{S}) - \boldsymbol{\mu}(\mathcal{D}_\mathcal{G}) \right\|^2 \leq \mathcal{O}\left( \frac{1}{k^2} \right) + \frac{16}{(1-\alpha)^2} \sigma_\mathcal{G}^2 + \frac{4\epsilon^2}{|\mathcal{D}_\mathcal{G}|^2 (1-\alpha)^2} \tag{51}$$

where, $\alpha = |\mathcal{D}_\mathcal{B}|/|\mathcal{D}_\mathcal{G}| < 1$ is the ratio of corrupted and clean samples. $\boldsymbol{\mu}(\mathcal{D}_\mathcal{S})$ and $\boldsymbol{\mu}(\mathcal{D}_\mathcal{G})$ denote the mean of the selected subset and the true uncorrupted mean respectively.

*Proof.* We begin by decomposing the overall error using the triangle inequality:

$$\Delta^2 = \left\| \boldsymbol{\mu}(\mathcal{D}_\mathcal{S}) - \boldsymbol{\mu}(\mathcal{D}_\mathcal{G}) \right\|^2 \tag{52}$$

$$= \left\| \left( \boldsymbol{\mu}(\mathcal{D}_\mathcal{S}) - \boldsymbol{\mu}_\epsilon^{\text{GM}}(\mathcal{D}) \right) + \left( \boldsymbol{\mu}_\epsilon^{\text{GM}}(\mathcal{D}) - \boldsymbol{\mu}(\mathcal{D}_\mathcal{G}) \right) \right\|^2 \tag{53}$$

$$\leq 2 \left\| \boldsymbol{\mu}(\mathcal{D}_\mathcal{S}) - \boldsymbol{\mu}_\epsilon^{\text{GM}}(\mathcal{D}) \right\|^2 + 2 \left\| \boldsymbol{\mu}_\epsilon^{\text{GM}}(\mathcal{D}) - \boldsymbol{\mu}(\mathcal{D}_\mathcal{G}) \right\|^2 \tag{54}$$

**Bounding the first term.**  The herding-style greedy procedure is designed to iteratively reduce the discrepancy between the empirical mean of the selected subset and the robust target moment $\boldsymbol{\mu}_\epsilon^{\text{GM}}(\mathcal{D})$. Standard results from the analysis of kernel herding imply that

$$\left\|\boldsymbol{\mu}(\mathcal{D}_\mathcal{S}) - \boldsymbol{\mu}_\epsilon^{\text{GM}}(\mathcal{D})\right\| = \mathcal{O}\!\left(\frac{1}{k}\right) \tag{55}$$

Thus, there exists a constant $C_1 > 0$ such that

$$\left\|\boldsymbol{\mu}(\mathcal{D}_\mathcal{S}) - \boldsymbol{\mu}_\epsilon^{\text{GM}}(\mathcal{D})\right\|^2 \leq \frac{C_1}{k^2}. \tag{56}$$

**Bounding the second term.**  By Theorem 1, the robust estimator $\boldsymbol{\mu}_\epsilon^{\text{GM}}(\mathcal{D})$ satisfies

$$\left\|\boldsymbol{\mu}_\epsilon^{\text{GM}}(\mathcal{D}) - \boldsymbol{\mu}(\mathcal{D}_\mathcal{G})\right\|^2 \leq \frac{8|\mathcal{D}_\mathcal{G}|^2}{(|\mathcal{D}_\mathcal{G}| - |\mathcal{D}_\mathcal{B}|)^2}\,\sigma^2(\mathcal{D}_\mathcal{G}) + \frac{2\epsilon^2}{(|\mathcal{D}_\mathcal{G}| - |\mathcal{D}_\mathcal{B}|)^2}. \tag{57}$$

Since $|\mathcal{D}_\mathcal{G}| - |\mathcal{D}_\mathcal{B}| = |\mathcal{D}_\mathcal{G}|(1 - \alpha)$ with $\alpha = |\mathcal{D}_\mathcal{B}|/|\mathcal{D}_\mathcal{G}| < 1$, we can rewrite the bound as

$$\left\|\boldsymbol{\mu}_\epsilon^{\text{GM}}(\mathcal{D}) - \boldsymbol{\mu}(\mathcal{D}_\mathcal{G})\right\|^2 \leq \frac{8}{(1 - \alpha)^2}\,\sigma^2(\mathcal{D}_\mathcal{G}) + \frac{2\epsilon^2}{|\mathcal{D}_\mathcal{G}|^2(1 - \alpha)^2}. \tag{58}$$

Multiplying both sides by 2 yields:

$$2\left\|\boldsymbol{\mu}_\epsilon^{\text{GM}}(\mathcal{D}) - \boldsymbol{\mu}(\mathcal{D}_\mathcal{G})\right\|^2 \leq \frac{16}{(1 - \alpha)^2}\,\sigma^2(\mathcal{D}_\mathcal{G}) + \frac{4\epsilon^2}{|\mathcal{D}_\mathcal{G}|^2(1 - \alpha)^2}. \tag{59}$$

**Combining the bounds.**  Substituting the bounds from (56) and (59) into (54), we obtain

$$\Delta^2 \leq \frac{2C_1}{k^2} + \frac{16}{(1 - \alpha)^2}\,\sigma^2(\mathcal{D}_\mathcal{G}) + \frac{4\epsilon^2}{|\mathcal{D}_\mathcal{G}|^2(1 - \alpha)^2}.$$

Since the constant $2C_1$ can be absorbed into the $\mathcal{O}(1/k^2)$ term, we conclude that

$$\Delta^2 \leq \mathcal{O}\!\left(\frac{1}{k^2}\right) + \frac{16}{(1 - \alpha)^2}\,\sigma^2(\mathcal{D}_\mathcal{G}) + \frac{4\epsilon^2}{|\mathcal{D}_\mathcal{G}|^2(1 - \alpha)^2}.$$

This completes the proof. ∎

## G. Computing Geometric Median

As discussed in Section 3, the Geometric Median (GM) also known as the Fermat-Weber point (Weber et al., 1929), is a robust measure of central tendency for a set of observations. Given a finite collection of observations $\{\mathbf{x}_1, \mathbf{x}_2, \dots \mathbf{x}_n\}$ defined over Hilbert space $\mathcal{H} \in \mathbb{R}^d$, equipped with norm $\|\cdot\|$ and inner $\langle\cdot\rangle$ operators, the geometric median(Definition 3) (Weber et al., 1929) $\text{GM}(\{\mathbf{x}_1, \mathbf{x}_2, \dots \mathbf{x}_n\})$ is defined as the solution to the optimization problem:

$$\boldsymbol{\mu}^{\text{GM}} = \arg\min_{\mathbf{z} \in \mathbb{R}^d} \rho(\mathbf{z}), \quad \text{where} \quad \rho(\mathbf{z}) = \sum_{i=1}^{n} \|\mathbf{x}_i - \mathbf{z}\| \tag{60}$$

where $\rho(\mathbf{z})$ is the sum of Euclidean distances from $\mathbf{z}$ to the data points, as illustrated in Figure 7.

Note that, in contrast, the empirical mean $\hat{\mu}$ is defined as the minimizer of the squared Euclidean distances:

$$\hat{\boldsymbol{\mu}} = \arg\min_{\mathbf{z} \in \mathbb{R}^d} \rho(\mathbf{z}), \quad \text{where} \quad \rho(\mathbf{z}) = \sum_{i=1}^{n} \|\mathbf{x}_i - \mathbf{z}\|^2 \tag{61}$$

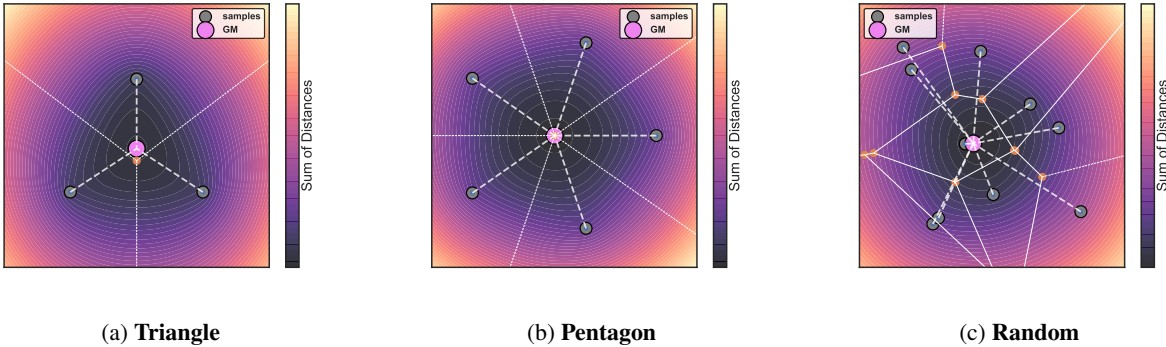

(a) **Triangle**       (b) **Pentagon**       (c) **Random**

Figure 7: GEOMETRIC MEDIAN VISUALIZATION: The plots illustrate the computation of the geometric median (denoted by the pink circle) for three different spatial point configurations: (a) Triangle, (b) Pentagon, and (c) Random. The color gradient represents the sum of distances $\rho(\mathbf{z})$ from a candidate point $\mathbf{z}$ to all data points, with darker regions indicating smaller values of $\rho(\mathbf{z})$. The white dashed lines show the connections between the geometric median and the data points, emphasizing how the geometric median minimizes the total Euclidean distance to all points. Additionally, the Voronoi regions formed around the data points visually partition the space based on proximity, offering insight into how the geometric median balances contributions from each point. In symmetric configurations such as (a) and (b), the Voronoi structure highlights the symmetry in influence regions, leading to a geometric median located at the center. For the random configuration (c), the irregular Voronoi regions illustrate the varying influence of data points, with the geometric median robustly adapting to minimize the total distance while down-weighting the effect of outlier-like points.

Minimizing the squared Euclidean distances ensures computational simplicity, as a closed-form solution exists:

$$\hat{\mu} = \frac{1}{n} \sum_{i=1}^{n} \mathbf{x}_i \tag{62}$$

which follows from the first-order optimality conditions of convex quadratic minimization. However, this also makes the empirical mean sensitive to outliers, as extreme values have a disproportionately large effect on the sum of squared distances. On the other hand, the linear penalty in the GM computation ensures that the objective is less influenced by outliers, as deviations are not amplified quadratically.

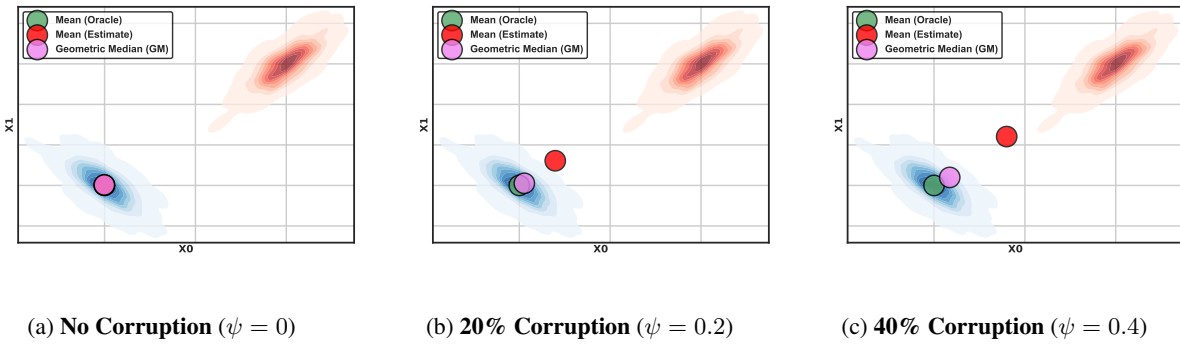

(a) **No Corruption** ($\psi = 0$)   (b) **20% Corruption** ($\psi = 0.2$)   (c) **40% Corruption** ($\psi = 0.4$)

Figure 8: ROBUST MEAN ESTIMATION: This figure illustrates the deviation of the Empirical Mean (red) and Geometric Median (purple) from the Oracle Mean (green) as the fraction of corrupted samples $0 \leq \psi < 1/2$ is varied. The uncorrupted distribution is shown in **Blue**, while the adversarial distribution is shown in red. At $\psi = 0$, all estimators align closely with the oracle mean. As $\psi$ increases, the empirical mean is significantly influenced by the corrupted samples, drifting toward the corrupted distribution. In contrast, the GM remains robust, staying closer to the oracle mean demonstrating the resilience of the GM in the presence of corruptions.

### G.1. Approximate GM : The Weiszfeld Algorithm

Note that the GM optimization problem is inherently non-smooth due to the presence of the Euclidean norm $\|\mathbf{x}_i - \mathbf{z}\|$ which leads to non-differentiability at points where multiple distances are equal, making gradient-based optimization difficult. Furthermore, computing the exact geometric median is NP-hard in dimensions $d \geq 2$ (Bajaj, 1988), as it involves solving a

high-degree polynomial system without a closed-form solution.

As a result, a long line of research has focused on developing efficient approximation methods, and several algorithms have been proposed (Weiszfeld, 1937; Vardi & Zhang, 2000; Cohen et al., 2016) to compute $\epsilon$-approx GM instead.

Recall that, an $\epsilon$-approximate geometric median (5), denoted by $\boldsymbol{\mu}_\epsilon^{\text{GM}}$, satisfies:

$$\sum_{i=1}^n \left\| \boldsymbol{\mu}_\epsilon^{\text{GM}} - \mathbf{x}_i \right\| \leq (1 + \epsilon) \sum_{i=1}^n \left\| \boldsymbol{\mu}^{\text{GM}} - \mathbf{x}_i \right\|$$

where $\boldsymbol{\mu}^{\text{GM}}$ is the exact geometric median.

In this work, we adopt the celebrated Weiszfeld algorithm (Algorithm 2) – an iterative procedure for approximating the geometric median by leveraging a re-weighted averaging scheme.

---

**Algorithm 2 (WEISZFELD, 1937) COMPUTING $\epsilon$-APPROX. GM**

---

**Input:**
Observations $\{\mathbf{x}_i \in \mathbb{R}^d\}_{i=1}^n$, initial guess $\mathbf{z}^{(0)} \in \mathbb{R}^d$, convergence threshold $\epsilon > 0$, and regularization parameter $\delta > 0$.
**Initialization:**
Initialize $\mathbf{z}^{(0)}$ (e.g., as the arithmetic mean of $\{\mathbf{x}_i\}$), set iteration counter $k \leftarrow 0$.
**while** *not converged (i.e., $\|\mathbf{z}^{(k+1)} - \mathbf{z}^{(k)}\| \geq \epsilon$)* **do**

> (compute update step)

$$\mathbf{z}^{(k+1)} \leftarrow \frac{\sum_{i=1}^n \frac{\mathbf{x}_i}{\|\mathbf{x}_i - \mathbf{z}^{(k)}\| + \delta}}{\sum_{i=1}^n \frac{1}{\|\mathbf{x}_i - \mathbf{z}^{(k)}\| + \delta}}$$

> (update iteration counter)
> $k \leftarrow k + 1$

**end**
**Return: $\mathbf{z}^* \leftarrow \mathbf{z}^{(k)}$**

---

Note that the subgradient of $\rho(\mathbf{z})$ at any point $\mathbf{z}$ is given by:

$$\partial \rho(\mathbf{z}) = \sum_{i=1}^n \frac{\mathbf{z} - \mathbf{x}_i}{\|\mathbf{x}_i - \mathbf{z}\|}. \tag{63}$$

Let $f(\mathbf{z}) = \|\mathbf{z} - \mathbf{x}_i\|$, where:

$$f(\mathbf{z}) = \sqrt{(\mathbf{z} - \mathbf{x}_i)^\top (\mathbf{z} - \mathbf{x}_i)}.$$

The Euclidean norm can be expressed as:

$$\|\mathbf{z} - \mathbf{x}_i\| = \sqrt{\sum_{j=1}^d (z_j - x_{ij})^2}.$$

Define $g(\mathbf{z}) = (\mathbf{z} - \mathbf{x}_i)^\top (\mathbf{z} - \mathbf{x}_i)$, so:

$$f(\mathbf{z}) = \sqrt{g(\mathbf{z})}.$$

Using the chain rule, the gradient of $f(\mathbf{z})$ with respect to $\mathbf{z}$ is:

$$\nabla_{\mathbf{z}} f(\mathbf{z}) = \frac{1}{2\sqrt{g(\mathbf{z})}} \nabla_{\mathbf{z}} g(\mathbf{z}).$$

The function $g(\mathbf{z})$ is given by:

$$g(\mathbf{z}) = (\mathbf{z} - \mathbf{x}_i)^\top (\mathbf{z} - \mathbf{x}_i),$$

and its gradient is:

$$\nabla_{\mathbf{z}} g(\mathbf{z}) = 2(\mathbf{z} - \mathbf{x}_i).$$

Substituting $\nabla_{\mathbf{z}} g(\mathbf{z})$ into the chain rule:

$$\nabla_{\mathbf{z}} f(\mathbf{z}) = \frac{1}{2\sqrt{g(\mathbf{z})}} \cdot 2(\mathbf{z} - \mathbf{x}_i).$$

Since $\sqrt{g(\mathbf{z})} = \|\mathbf{z} - \mathbf{x}_i\|$, we have:

$$\nabla_{\mathbf{z}} \|\mathbf{z} - \mathbf{x}_i\| = \frac{\mathbf{z} - \mathbf{x}_i}{\|\mathbf{z} - \mathbf{x}_i\|}.$$

thus we have that the subgradient of the Euclidean norm $\|\mathbf{z} - \mathbf{x}_i\|$ with respect to $\mathbf{z}$ is:

$$\nabla_{\mathbf{z}} \|\mathbf{z} - \mathbf{x}_i\| = \frac{\mathbf{z} - \mathbf{x}_i}{\|\mathbf{z} - \mathbf{x}_i\|}.$$

To find the optimal solution $\mathbf{z}^*$, we can solve the condition:

$$\mathbf{0} \in \partial \rho(\mathbf{z}^*)$$

Substituting $\mathbf{z} = \mathbf{z}^*$ and re-arranging the terms, we get:

$$\mathbf{z}^* = \frac{\sum_{i=1}^n \frac{\mathbf{x}_i}{\|\mathbf{x}_i - \mathbf{z}^*\|}}{\sum_{i=1}^n \frac{1}{\|\mathbf{x}_i - \mathbf{z}^*\|}}.$$

Since this equation is non-linear in $\mathbf{z}^*$, solving it directly is infeasible. Instead, the Weiszfeld algorithm approximates $\mathbf{z}^*$ iteratively using the update:

$$\mathbf{z}^{(k+1)} = \frac{\sum_{i=1}^n \frac{\mathbf{x}_i}{\|\mathbf{x}_i - \mathbf{z}^{(k)}\|}}{\sum_{i=1}^n \frac{1}{\|\mathbf{x}_i - \mathbf{z}^{(k)}\|}}$$

where $\mathbf{z}^{(k)}$ is the estimate at the $k$-th iteration.

This update step can be interpreted as a re-weighted average, where the weights are inversely proportional to the distance of each point $\mathbf{x}_i$ from the current estimate $\mathbf{z}^{(k)}$. Points closer to $\mathbf{z}^{(k)}$ contribute most to the next estimate.

**Handling Non-Differentiability :** At points where $\mathbf{z}^{(k)} = \mathbf{x}_i$ i.e. the current estimate coincides with one of the observations, the term $\|\mathbf{x}_i - \mathbf{z}^{(k)}\| = 0$ results in division by zero. To address this, the algorithm excludes the term corresponding to $\mathbf{x}_i$ from the summation. Alternatively, the subgradient of $\|\mathbf{x}_i - \mathbf{z}\|$ at $\mathbf{z} = \mathbf{x}_i$ can be defined as:

$$\partial \|\mathbf{x}_i - \mathbf{z}\| = \begin{cases} \frac{\mathbf{z} - \mathbf{x}_i}{\|\mathbf{x}_i - \mathbf{z}\|} & \text{if } \mathbf{z} \neq \mathbf{x}_i, \\ \{\mathbf{g} : \|\mathbf{g}\| \leq 1\} & \text{if } \mathbf{z} = \mathbf{x}_i. \end{cases}$$

**Convergence and Regularization:** Notably, the Weiszfeld algorithm converges under mild conditions if the initial point $\mathbf{z}^{(0)}$ is not chosen at one of the data points. Convergence can be shown using fixed-point theory or by analyzing the decrease in the objective function $\rho(\mathbf{z})$ at each iteration. However, convergence is only guaranteed to a local minimum if the data points $\mathbf{x}_i$ are not in general position (e.g., collinear points in $\mathbb{R}^2$). To handle singularities, we modify the denominator to avoid division by zero:

$$\mathbf{z}^{(k+1)} = \frac{\sum_{i=1}^n \frac{\mathbf{x}_i}{\|\mathbf{x}_i - \mathbf{z}^{(k)}\| + \delta}}{\sum_{i=1}^n \frac{1}{\|\mathbf{x}_i - \mathbf{z}^{(k)}\| + \delta}}$$

where $\delta > 0$ is a small regularization term.

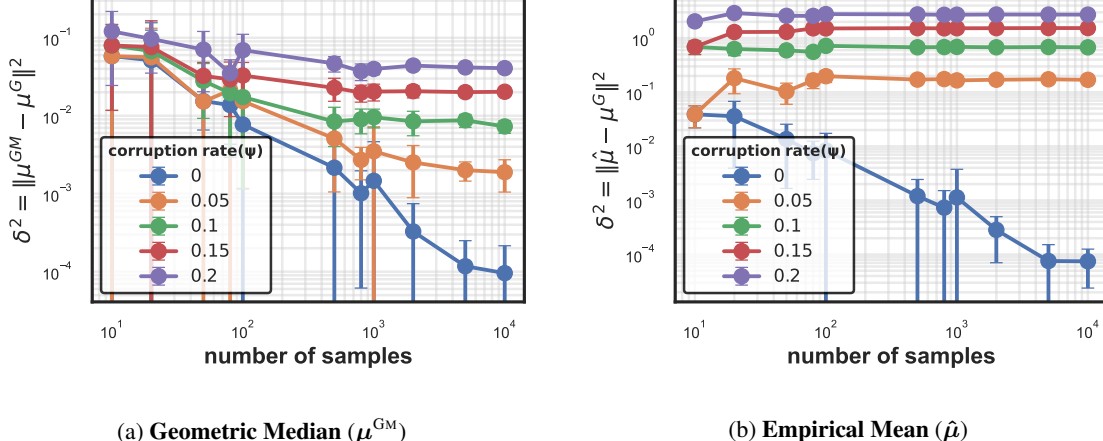

(a) **Geometric Median ($\boldsymbol{\mu}^{\text{GM}}$)**  (b) **Empirical Mean ($\hat{\mu}$)**

Figure 9: **MEAN ESTIMATION ERROR SCALING FOR RECOVERING ANISOTROPIC GAUSSIAN :** Comparison of Mean Estimation Error i.e. deviation from the oracle mean using Geometric Median (GM) $\mu^{\text{GM}}$ and Empirical Mean (EM) $\hat{\mu}$ as a function of sample size and amount of corruption. GM is computed using Weiszfeld Algorithm Algorithm 2. The $x$ - axis represents the sample size i.e. the number of samples drawn i.i.d. from an anisotropic Gaussian, with a fraction $0 \leq \psi < 1/2$ of them corrupted by an adversarially chosen distribution. The plots depict the convergence behavior of both methods across varying corruption rates, highlighting the robustness of GM in the presence of outliers. The plots further emphasize the effect of sample size on estimation error for both the estimators.

## H. Additional Details on Computational Complexity Analysis

### H.1. GM Scaling Law

We now discuss the computational complexity of GM computation over $n$ points in $\mathbb{R}^d$:

Weiszfeld's algorithm proceeds by iteratively updating the current estimate $\mathbf{z}^{(k)}$ via a re-weighted average of the data points. Each iteration requires:

1. *Distance evaluations:* Computing $\|\mathbf{x}_i - \mathbf{z}^{(k)}\|$ for all $i = 1, 2, \ldots, n$. Each distance in $\mathbb{R}^d$ is evaluated in $\mathcal{O}(d)$ time, so this step costs $\mathcal{O}(nd)$ per iteration.

2. *Update step:* Forming the weighted average $\mathbf{z}^{(k+1)} = \left(\sum_i \frac{\mathbf{x}_i}{\|\mathbf{x}_i - \mathbf{z}^{(k)}\|}\right) \big/ \left(\sum_i \frac{1}{\|\mathbf{x}_i - \mathbf{z}^{(k)}\|}\right)$ also takes $\mathcal{O}(nd)$ time.

Hence, each iteration costs $\mathcal{O}(nd)$.

Although Weiszfeld's algorithm is guaranteed to converge under mild conditions, the *rate* of convergence depends on the geometric configuration of the data points:

- **Non-degenerate (typical) case:** If the geometric median $\boldsymbol{\mu}^{\text{GM}}$ lies strictly in the interior (i.e., it does *not* coincide with any $\mathbf{x}_i$ and is at a positive distance from each data point), then Weiszfeld's iteration converges *linearly* near the optimum. The number of iterations to achieve an $\epsilon$-accurate solution typically scales as $\mathcal{O}(\log(1/\epsilon))$.

- **Degenerate (worst) case:** If the geometric median coincides with (or lies extremely close to) some data point(s), the objective can lose strict convexity and the Weiszfeld update may progress more slowly. In the worst theoretical analyses, the iteration count can grow as $\mathcal{O}(1/\epsilon)$.

Putting it together, let $K$ denote the iteration count required for an $\epsilon$-accurate estimate. We have:

$$\text{Time per iteration} = \mathcal{O}(n\,d),$$

$$K = \begin{cases} \mathcal{O}\big(\log(1/\epsilon)\big) & \text{(typical, non-degenerate)}, \\ \mathcal{O}\big(1/\epsilon\big) & \text{(worst-case, degenerate)}. \end{cases}$$

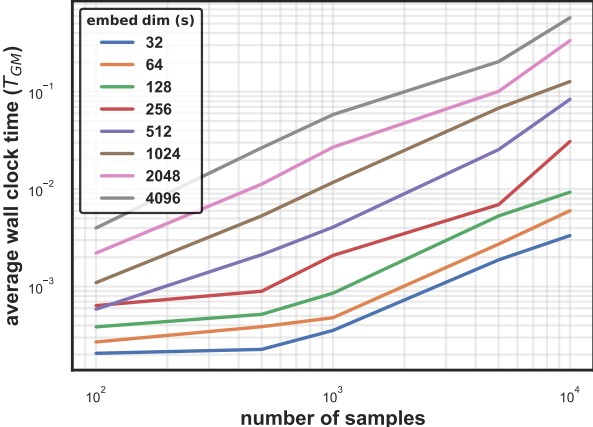

Figure 10: **Scaling Laws (Wall Clock) for Gm Computation**: This figure shows the scaling behavior for computing the geometric median using Weiszfeld algorithm ( Algorithm 2) in terms of the wall clock time ($T_{GM}$) as a function of population size. The x-axis represents the number of samples ($n$) on a logarithmic scale, and the y-axis shows the average wall clock time ($T_{GM}$) on a logarithmic scale. Each curve corresponds to a different embedding dimension ($d$), ranging from 32 to 4096. The results were generated using synthetic data drawn from a standard normal distribution, with the geometric median computed iteratively until convergence (tolerance $\epsilon = 10^{-5}$, maximum iterations = 100). For each combination of $n$ and $s$, wall clock time was averaged across 10 random seed. The computational cost increases with both $n$ and $s$: for fixed $n$, the scaling with $s$ is approximately linear, while for fixed $s$, scaling with $n$ exhibits sub-linear to near-linear growth. These results emphasize the trade-offs in selecting $n$ and $s$ for practical applications. Experiments were run on a single-threaded CPU setup. These results confirm that the practical runtime aligns with the expected time complexity of $\mathcal{O}(ns \log(1/\epsilon))$ in typical scenarios, showcasing the efficiency of Weiszfeld algorithm for high-dimensional data.

Thus, the *overall time complexity* is either $\mathcal{O}\big(n\,d\,\log(1/\epsilon)\big)$ in the typical scenario or $\mathcal{O}\big(n\,d\,(1/\epsilon)\big)$ in the worst case. In practice, degeneracies are rare for generic data, and Weiszfeld's algorithm often converges quite rapidly to a high-accuracy solution. In Figure 10, we empirically validate the theoretical scaling behavior of the geometric median computation, demonstrating how the wall clock time $T^{\mathrm{GM}}$ scales with the number of samples $n$ and embedding dimension $s$.

### H.2. Gm Matching Scaling Law

We show how the batching scheme scales w.r.t wall-clock time in Figure 11 – almost linear scaling can be observed.

### H.3. Variance Considerations

In reducing computational costs, the use of sub-sampling for the robust Gm, and batching in the Gm Matching selection process inevitably introduces additional variance into the estimation and selection outcomes. In this section, we dissect the variance implications of these strategies, stressing that any gain in efficiency incurs a trade-off with statistical precision—a quintessential *no free lunch* scenario: faster computation comes with the cost of reduced statistical precision.

Note that the deviation of Gm from the true mean depends on the variance of the clean population (Theorem 1). This implies that, when estimating Gm over a subset of the data, as the variance of the population increases :

$$\Delta(\sigma^2) \approx \mathcal{O}\bigg( \big(\frac{1 - \lambda_{\mathrm{GM}}}{\lambda_{\mathrm{GM}}}\big) \cdot \frac{\sigma^2(\mathcal{D}_{\mathcal{G}})}{n} \bigg) \tag{64}$$

Thus, while sub-sampling offers a significant computational speedup by reducing the number of points used, it also amplifies the estimation variance—a trade-off that must be carefully balanced by choosing an appropriate sub-sampling fraction relative to the intrinsic noise level of the data. In practice, however, we notice that for sufficiently large $\gamma_{\mathrm{GM}}$, the variance increase is negligible compared to the savings as evidenced in Figure 9.

Similarly, batching in the Gm Matching selection process, where the selection of the $k$-subset is partitioned into $B$ smaller batches, inherently introduces additional error terms in the convergence. Firstly, for each batch, the herding convergence suffers a constant bias penalty because each batch only selects $k/B$ samples instead of $k$ samples, thereby incurring an error

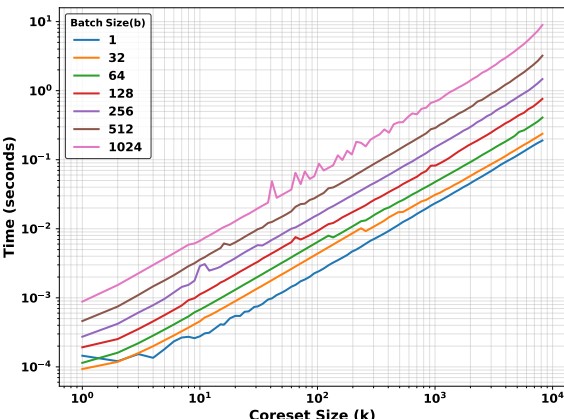

Figure 11: **SCALING LAWS (WALL CLOCK) FOR BATCH GM MATCHING** as a function of coreset size (k) across varying batch sizes (b). Log-log scale highlights the computational cost trends for batch sizes ranging from 1 to 1024.

term on the order of $\mathcal{O}(B/k)$. Moreover, the overall estimate is formed as the mean of the batch estimates, which introduces an additional variance component arising from the averaging of independent errors across batches. This variance term scales as $\mathcal{O}(1/\sqrt{B})$ i.e.

$$\left\| \boldsymbol{\mu}(\mathcal{D}_{\mathcal{S}}) - \boldsymbol{\mu}_{\epsilon}^{\text{GM}}(\mathcal{D}) \right\| \leq \mathcal{O}\left(\frac{B}{k}\right) + \mathcal{O}\left(\frac{1}{\sqrt{B}}\right). \tag{65}$$

This implies that, while batching can result in significant speedups – too small a batch size can also slow down convergence. In practice, we find that, with a moderate batch size, this additional error is often negligible, while the speedup is substantial as demonstrated in Figure 5.

# I. Additional Experimental Details

In this section, we will provide more detail on our experiments from Section 4, share additional results and provide more detailed discussion.

Our experiments are divided into three fundamental learning paradigms:

- *Approximating from Noisy Distributions*

- *Image Classification*

- *Image Generation*

By dividing the experiments into these categories, we ensure that GM MATCHING is tested across a wide spectrum of tasks, covering unsupervised, supervised, and generative learning scenarios. Detailed analyses in each paradigm help uncover not just the strengths of GM MATCHING but also potential limitations or areas where further improvements can be made. In each experiment, the performance of GM MATCHING is rigorously compared to established baselines, offering clear evidence of its competitive edge in handling noisy data, enhancing classification, and generating realistic outputs – demonstrating the practical applicability and versatility of GM MATCHING across a wide array of machine learning challenges.

## I.1. Baselines

We compare GM MATCHING (see Algorithm 1) against several baseline subset selection methods.

These baselines include:

- **Random:** Samples are selected uniformly at random. This approach serves as a **strong** baseline that does not incorporate any structure from the data – widely adopted in practical applications due to its simplicity and strong performance under idealized conditions.

- **Easy:** (Shah et al., 2020): This strategy selects samples that are closest to the centroid of the dataset. These "easy" samples are presumed to be representative of the core data distribution, but may under-represent the data's diversity, especially in the presence of noise.

- **Hard:** (Joshi et al., 2009): This approach selects samples that are farthest from the centroid. While these "hard" samples may capture edge cases or outliers, they can also be overly sensitive to noise and may not adequately reflect the central structure of the clean distribution.

- **Moderate:** (Xia et al., 2022): This strategy selects samples that are closest to the median distance from the centroid. This approach aims to balance the representation of the data by avoiding the extremes of the distribution, thereby providing a more stable and robust subset.

- **Kernel Herding:** (Chen et al., 2010) kernel herding employs a greedy algorithm to select samples that minimize the discrepancy between the empirical distribution and the target distribution in a reproducing kernel Hilbert space which forms the basis of this work.

- **Forgetting** (Toneva et al., 2018): Data points that are easily forgotten during optimization are chosen.

- **GraNd-score** (Paul et al., 2021): Data points with larger loss gradient norms are included.

- **EL2N-score** (Paul et al., 2021): This focuses on data points with larger norms of the error vector, which is the difference between the predicted class probabilities and the one-hot label encoding.

- **Optimization-based** (Yang et al., 2022): This method uses the influence function (Koh & Liang, 2017) to select data points that minimize the generalization gap under strict constraints.

- **Self-sup.-selection** (Sorscher et al., 2022): After self-supervised pre-training and clustering, data points are selected based on their distance to the nearest cluster centroid, with the number of clusters set to the number of classes to avoid tuning. Points with larger distances are chosen.

For GM MATCHING, the GM is approximated via Algorithm 2 (Weiszfeld, 1937). The optimization routine terminates, either when the GM approximation error is small $\epsilon_0$ or after a maximum number of iterations $T_{\max}$ is reached. $\epsilon_0 = 10^{-8}$ and $T_{\max} = 1000$ are used for all the experiments.

### I.2. Approximating from Noisy Distributions

We simulate a Gaussian Mixture Model (GMM) comprising both clean and adversarial components to evaluate robust moment estimation in noisy datasets. The clean and corrupt samples are drawn from two anisotropic Gaussian distributions:

$$p_{\text{clean}}(x) = \mathcal{N}(x; \mu, \Sigma) , \; p_{\text{adv}}(x) = \mathcal{N}(x; \mu', \Sigma'). \tag{66}$$

In our experiments, the two Gaussians are well-separated in the 2D plane, ensuring a clear notion of "clean" vs. "adversarial" clusters with means and covariances:

$$\boldsymbol{\mu} = \begin{bmatrix} 0 \\ 0 \end{bmatrix} , \; \Sigma = \begin{bmatrix} 1 & -0.5 \\ -0.5 & 0.5 \end{bmatrix} , \; \mu' = \begin{bmatrix} 10 \\ 6 \end{bmatrix} , \; \Sigma' = \begin{bmatrix} 1 & 0.5 \\ 0.5 & 0.5 \end{bmatrix}. \tag{67}$$

Assuming that a fraction $\psi$ of the data is corrupted, the input data is modeled as a mixture:

$$p_{\text{corrupt}}(x) = (1 - \psi) \, p_{\text{clean}}(x) + \psi \, p_{\text{adv}}(x). \tag{68}$$

The objective is to select a subset $\mathcal{D}_S$ of samples such that the subset induced distribution $p_S(x)$ is as close to the clean distribution $p_{\text{clean}}(x)$. We generate a total of $n = 10^3$ samples, where $\psi n$ samples are generated from $p_{\text{adv}}(x)$ and the rest of the $(1 - \psi)n$ samples are generated from $p_{\text{clean}}(x)$, and the sampling algorithms subset 10% of the samples.

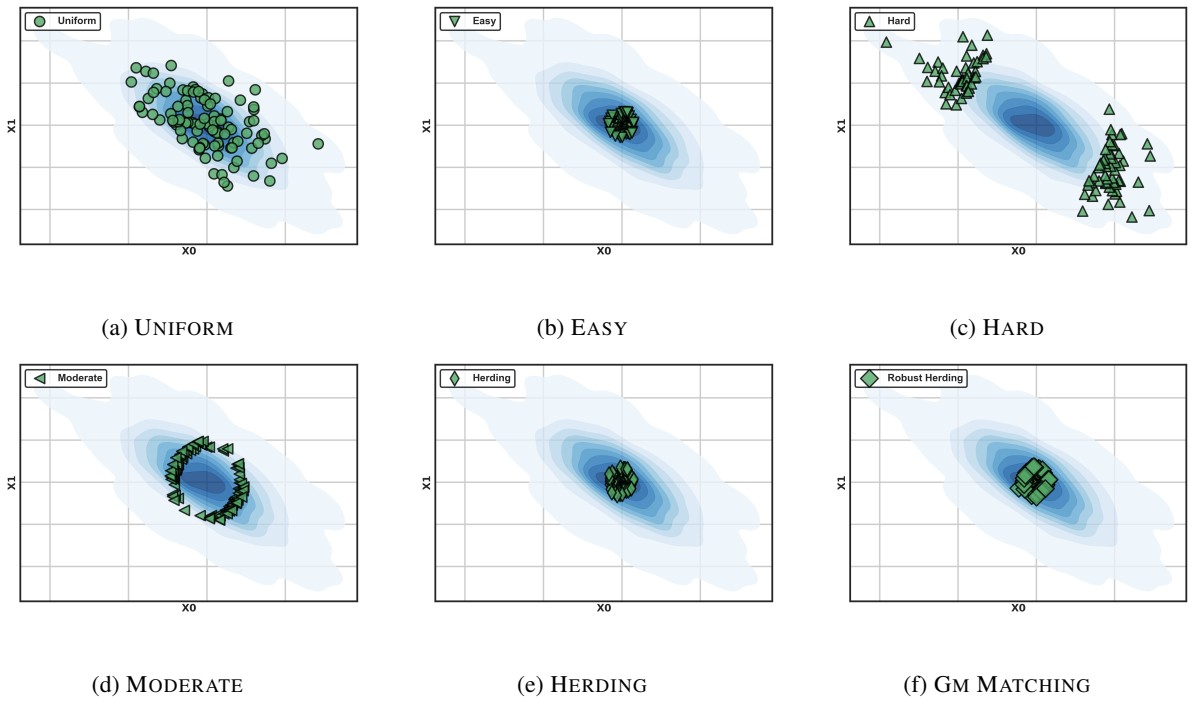

(a) UNIFORM  (b) EASY  (c) HARD

(d) MODERATE  (e) HERDING  (f) GM MATCHING

Figure 12: **No Corruption :** We select 10% of the samples using: (UNIFORM) Random Sampling, (EASY) Selection of samples closest to the centroid. (HARD) Selection of samples farthest from the centroid. (MODERATE) Selection of samples closest to the median distance from the centroid. (HERDING) Moment Matching, (GM MATCHING) Robust Moment (GM) Matching (6).

### I.2.1. ROBUSTNESS TO CORRUPTION

Firstly, in **Figure 1,8**, we observe that as the fraction of adversarial samples $\psi$ increases, the empirical mean of the noisy data set drifts noticeably towards the adversarial group. In contrast, GM remains anchored near the true uncorrupted mean $\boldsymbol{\mu}$. This divergence underscores the vulnerability of classical moment estimators to even modest levels of noise – a phenomenon well documented in robust statistics (Huber, 1992; Diakonikolas & Kane, 2019).

In our robust sampling experiments, we observe that when $\psi = 0$ i.e. the no-corruption scenario, all methods trivially capture the clean distribution, but as soon as a fraction of the data is adversarial, many standard strategies fail to consistently filter out the adversarial cluster. However, as $\psi$ increases as demonstrated in **Figure 12,13,2, 14** – methods like uniform sampling or even heuristic approaches (Easy, Hard, and Moderate), as well as standard Kernel Herding, begin to pick up an increasing number of adversarial points. Their selected subsets start to show a clear dual-group pattern, which results in a mixed empirical distribution that deviates from $p_{\text{clean}}(x)$.

In contrast, GM MATCHING yields a significantly more robust subset than these baseline methods, effectively filtering out adversarial noise while preserving the intrinsic structure of the clean data. The robustness of GM MATCHING becomes particularly prominent at high corruption levels ($\psi = 0.4$, $\psi = 0.45$) where most other methods fail. This demonstrates the potential of the proposed strategy to serve as a powerful tool for robust moment estimation in noisy and adversarial settings.

### I.2.2. ABLATIONS WITH PROXY ENCODER

Next, we examine the impact of various kernel maps on the performance of GM MATCHING, particularly in the presence of data corruption. Using a Gaussian mixture model as our testbed, we empirically evaluated different kernels to understand their ability to capture the underlying data structure and robustness under noise.

In Figure 15, we experiment with polynomial kernels:

$$\omega(\mathbf{x}, \mathbf{x}') = (\langle \mathbf{x}, \mathbf{x}' \rangle + c)^d \tag{69}$$

where, $c$ is a constant (typically set to 1) and $d$ is the degree of the polynomial. We tried varying degrees $d = \{1, 3, 5, 10\}$,

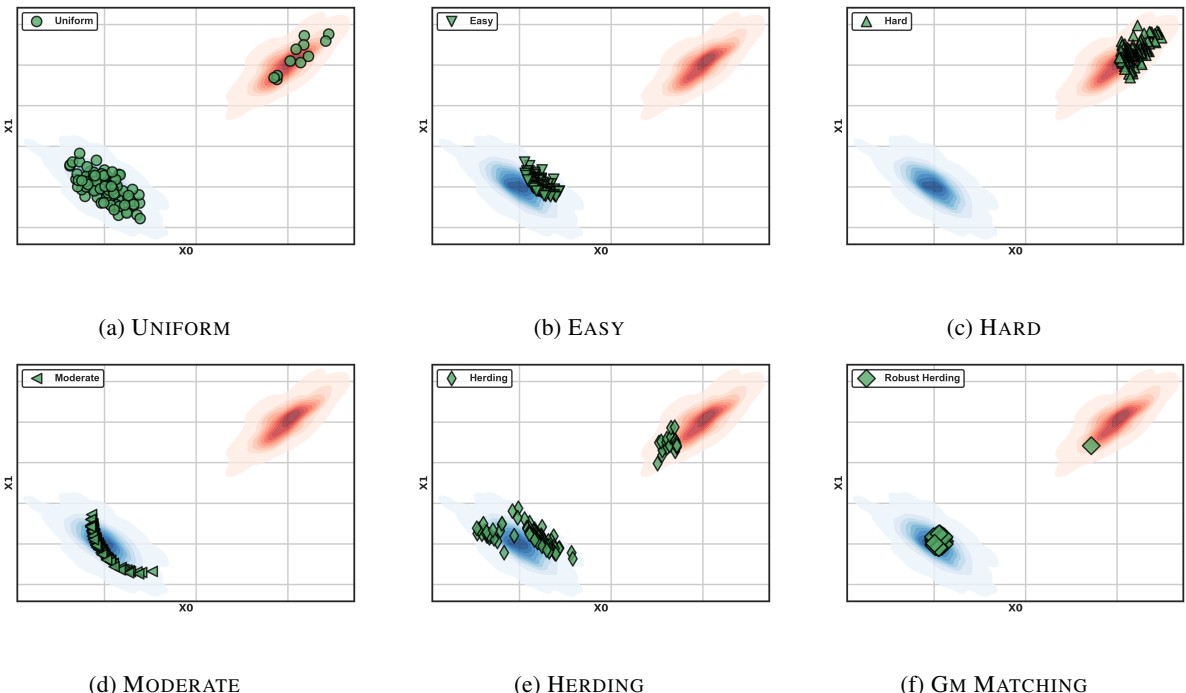

(a) UNIFORM          (b) EASY          (c) HARD

(d) MODERATE          (e) HERDING          (f) GM MATCHING

Figure 13: **20% Corruption**: In this experiment, 20% of the samples are corrupted – drawn from a adversary chosen distribution (red). We select 10% samples using: (UNIFORM) Random Sampling, (EASY) Selection of samples closest to the centroid. (HARD) Selection of samples farthest from the centroid. (MODERATE) Selection of samples closest to the median distance from the centroid. (HERDING) Moment Matching, (GM MATCHING) Robust Moment (GM) Matching (6). We see that while EASY remains robust, it is clearly sampling from low-density areas – failing to capture the prototypical samples.

highlighting how increased degree improves representational capacity while potentially amplifying noise when data are corrupted 20% corruption.

### I.3. Image Classification

This set of experiments evaluates the performance of GM MATCHING when applied to Image Classification tasks. To ensure reproducibility, our experimental setup – including baselines, neural architectures, and choice of hyper-parameters – is identical to (Xia et al., 2022).

#### I.3.1. EXPERIMENTAL SETUP

We evaluate the performance of GM MATCHING against multiple popular data pruning baselines, namely Random selection, Moderate (Xia et al., 2022), Forgetting (Toneva et al., 2018), Kernel Herding (Chen et al., 2010), GraNd-score (Paul et al., 2021), EL2N-score (Paul et al., 2021), and Self-supervised Selection (Sorscher et al., 2022). These baselines are evaluated across both standard (uncorrupted) and robustness-focused experimental settings.

Our robust sampling experiments assess the resilience of each algorithm to a variety of data corruptions, measuring their generalization capabilities beyond conventional training conditions. Specifically, we investigate three key robustness scenarios: (1) direct corruptions applied to images (Appendix I.3.3), (2) label noise (Appendix I.3.4), and (3) adversarial attacks (Appendix I.3.5) across different corruption intensities.

Additionally, we conduct a series of ablation studies to analyze the impact of incorporating a Proxy Encoder within the GM MATCHING pipeline. These studies examine how changes in network architecture and distribution shifts between the pretraining distribution (proxy) and downstream tasks affect pruning efficacy. These experiments provide deeper insights into the role of the Proxy Encoder and its influence on the overall performance of GM MATCHING.

We conduct experiments on widely used image classification benchmarks, including CIFAR-10/100 (Krizhevsky, 2009),

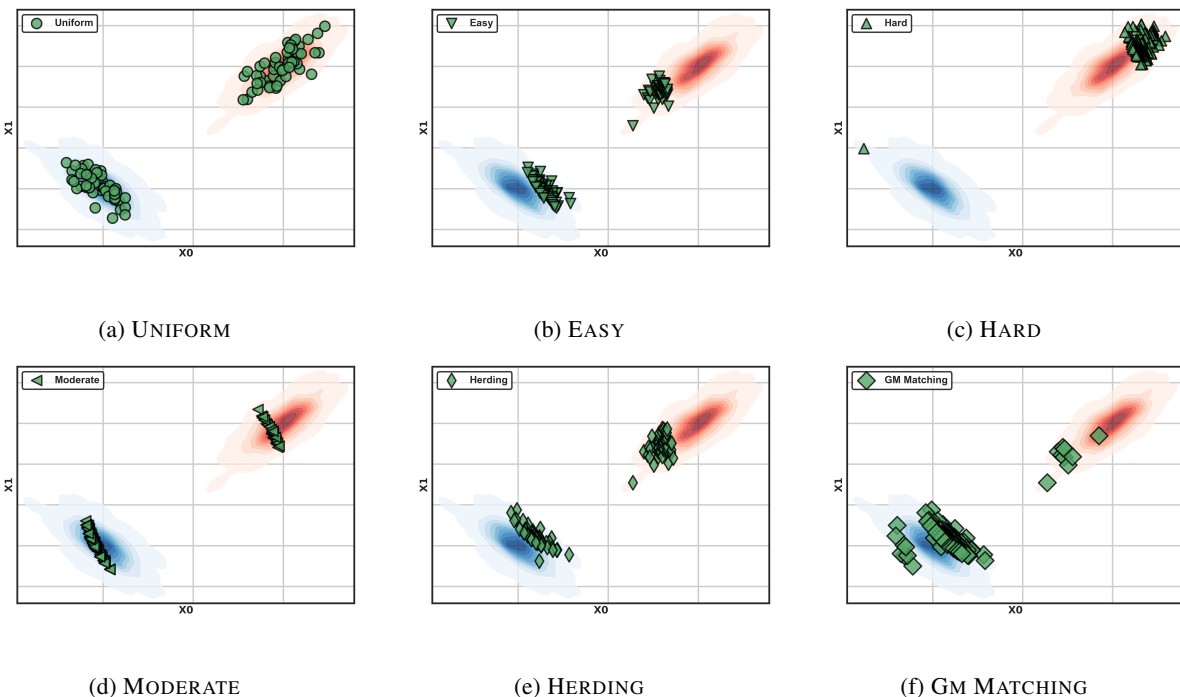

(a) UNIFORM          (b) EASY          (c) HARD

(d) MODERATE          (e) HERDING          (f) GM MATCHING

Figure 14: **Toy Example: 45% of the samples are corrupted** i.e. drawn from an adversary chosen distribution (red). We compare several baselines for choosing 10% samples: (UNIFORM) random sampling, (EASY) selects of samples closest to the centroid. (HARD) Selection of samples farthest from the centroid. (MODERATE) selects samples closest to the median distance from the centroid. (HERDING) moment matching, (GM MATCHING) robust moment (GM) matching (6). Clearly GM Matching is significantly more robust and diverse than the other approaches even at such high corruption rates.

Tiny-ImageNet (Le & Yang, 2015), and ImageNet-1K (Deng et al., 2009). Our study spans multiple popular neural network architectures, such as ResNet-18/50 (He et al., 2016), VGG-16 (Simonyan & Zisserman, 2014), ShuffleNet (Ma et al., 2018), SENet (Hu et al., 2018), EfficientNet-B0 (Tan & Le, 2019), and ViT-S (Dosovitskiy et al., 2020; Lee et al., 2021).

For CIFAR-10/100 datasets, the training configuration consists of a batch size of 128, SGD optimizer with momentum (0.9), weight decay of $5 \times 10^{-4}$, and an initial learning rate of 0.1. The learning rate undergoes step-wise decay by a factor of 5 at epochs 60, 120, and 160, totaling 200 epochs. Data augmentation strategies incorporate random cropping and random horizontal flipping.

For Tiny-ImageNet and ImageNet-1k experiments, we use a batch size of 256, SGD optimizer with momentum (0.9), weight decay of $1 \times 10^{-4}$, and an initial learning rate of 0.1. The learning rate decreases by a factor of 10 at epochs 30 and 60, across 90 total epochs, employing random horizontal flipping for data augmentation.

For computational practicality, particularly due to the scale of datasets and the complexity of geometric median computation, we employ the approximate Weiszfeld solver (Algorithm 2) for estimating the GM. Specifically, to further enhance computational efficiency without significant performance compromise, the solver computes the median over a randomly selected subset consisting of 50% of the training data points.

To account for variability and ensure statistical robustness, each experimental configuration is independently replicated across five distinct random seeds. Performance metrics are reported with variance to transparently capture and reflect the consistency of each method.

### I.3.2. IDEAL (NO CORRUPTION) SCENARIO

Our initial set of experiments evaluates the effectiveness of different data pruning strategies under an ideal, uncorrupted setting. We systematically prune datasets at selection ratios ranging from 20% to 80%, assessing the downstream classification performance across two widely used benchmarks: CIFAR-100 and Tiny ImageNet. The corresponding results, presented

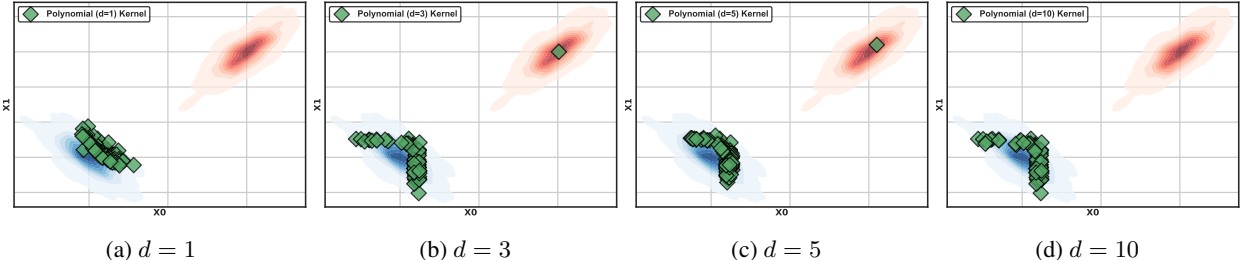

| (a) $d = 1$ | (b) $d = 3$ | (c) $d = 5$ | (d) $d = 10$ |

Figure 15: **POLYNOMIAL KERNEL**: GM MATCHING experiments under **20% corruption** using a polynomial kernel. Sub-figures (a)–(d) illustrate the effect of varying the polynomial degree (1, 3, 5, and 10, respectively) on the kernel mapping performance. The comparison highlights how increasing the degree can influence robustness and representational capacity in corrupted settings.

| | ImageNet-1k | | | | | |
|---|---|---|---|---|---|---|
| **Method / Ratio** | **60%** | **70%** | **80%** | **90%** | **100%** | **Mean ↑** |
| Random | $87.91 \pm 0.37$ | $88.63 \pm 0.95$ | $89.52 \pm 0.73$ | $89.57 \pm 0.60$ | $90.86 \pm 0.71$ | 89.30 |
| Herding | $88.25 \pm 2.16$ | $88.81 \pm 1.06$ | $89.60 \pm 0.58$ | $90.41 \pm 0.33$ | $90.86 \pm 0.71$ | 89.59 |
| Forgetting | $88.83 \pm 0.92$ | $89.81 \pm 0.97$ | $89.94 \pm 0.26$ | $90.41 \pm 0.58$ | $90.86 \pm 0.71$ | 89.97 |
| GraNd-score | $88.48 \pm 1.73$ | $89.82 \pm 2.07$ | $89.94 \pm 0.81$ | $90.41 \pm 0.62$ | $90.86 \pm 0.71$ | 89.90 |
| EL2N-score | $88.48 \pm 2.81$ | $89.82 \pm 1.14$ | $90.34 \pm 0.87$ | $90.57 \pm 0.46$ | $90.86 \pm 0.71$ | 90.01 |
| Self-sup.-selection | $87.59 \pm 2.61$ | $89.56 \pm 1.97$ | $\mathbf{90.74 \pm 0.27}$ | $90.49 \pm 0.98$ | $90.86 \pm 0.71$ | 89.49 |
| Moderate-DS | $89.23 \pm 0.96$ | $89.94 \pm 0.74$ | $90.65 \pm 0.51$ | $90.75 \pm 0.35$ | $90.86 \pm 0.71$ | 90.29 |
| **GM Matching** | $\mathbf{90.28 \pm 0.38}$ | $\mathbf{90.54 \pm 0.19}$ | $90.72 \pm 0.26$ | $\mathbf{90.84 \pm 0.32}$ | $90.86 \pm 0.71$ | **90.65** |

Table 4: **(CLEAN) IMAGE CLASSIFICATION**: Downstream Test Accuracy (Top-5) (%) of a ResNet-50 trained on a 60%, 70%, 80%, and 90% subset of ImageNet-1k, where the subset is selected via several benchmark data pruning algorithms(Appendix I.1). The best result in each case is in bold.

in Table 1, demonstrate that despite being designed with robustness-oriented applications in mind, GM MATCHING surpasses existing strong baselines even in the standard, clean setting. Across both datasets, GM MATCHING achieves an average improvement of over 2% compared to prior methods. Notably, its performance gains are particularly pronounced in the low-data selection regime (20%-40%), where it significantly outperforms competing pruning techniques. In Table 4, we observe similar improvements with GM MATCHING on ImageNet-1k. This suggests that GM MATCHING is especially effective in scenarios where data efficiency is critical, making it a promising approach for resource-constrained settings.

### I.3.3. ROBUSTNESS TO IMAGE CORRUPTION

In real-world applications, machine learning models are often deployed in environments where the input data is imperfect, degraded, or subject to various forms of corruption. This degradation can stem from sensor noise, environmental conditions, transmission artifacts, or adversarial perturbations.

To systematically evaluate the robustness of data pruning strategies under such conditions, we introduce structured image corruptions into the dataset and assess the downstream test accuracy across varying levels of pruning.

Specifically, we adopt the following practical perturbation types, drawing from established robustness benchmarks (Hendrycks & Dietterich, 2019; Szegedy et al., 2013) as -

- *Gaussian Noise*: Models real-world sensor noise by adding random perturbations sampled from a standard normal distribution.

- *Random Occlusion*: Mimics missing or occluded regions in images by replacing random patches with black or noisy pixels.

- *Resolution Reduction*: Simulates low-quality images by applying aggressive down-sampling and up-sampling, introducing

| CIFAR-100 | | | | | | | |
|---|---|---|---|---|---|---|---|
| **Method / Selection ratio** | **20%** | **30%** | **40%** | **60%** | **80%** | **100%** | **Mean ↑** |
| **No Corruption** | | | | | | | |
| Random | 50.26±3.24 | 53.61±2.73 | 64.32±1.77 | 71.03±0.75 | 74.12±0.56 | 78.14±0.55 | 62.67 |
| Herding | 48.39±1.42 | 50.89±0.97 | 62.99±0.61 | 70.61±0.44 | 74.21±0.49 | 78.14±0.55 | 61.42 |
| Forgetting | 35.57±1.40 | 49.83±0.91 | 59.65±2.50 | **73.34±0.39** | **77.50±0.53** | 78.14±0.55 | 59.18 |
| GraNd-score | 42.65±1.39 | 53.14±1.28 | 60.52±0.79 | 69.70±0.68 | 74.67±0.79 | 78.14±0.55 | 60.14 |
| EL2N-score | 27.32±1.16 | 41.98±0.54 | 50.47±1.20 | 69.23±1.00 | 75.96±0.88 | 78.14±0.55 | 52.99 |
| Optimization-based | 42.16±3.30 | 53.19±2.14 | 58.93±0.98 | 68.93±0.70 | 75.62±0.33 | 78.14±0.55 | 59.77 |
| Self-sup.-selection | 44.45±2.51 | 54.63±2.10 | 62.91±1.20 | 70.70±0.82 | 75.29±0.45 | 78.14±0.55 | 61.60 |
| Moderate-DS | 51.83±0.52 | 57.79±1.61 | 64.92±0.93 | 71.87±0.91 | 75.44±0.40 | 78.14±0.55 | 64.37 |
| **GM Matching** | **55.93± 0.48** | **63.08± 0.57** | **66.59± 1.18** | 70.82± 0.59 | 74.63± 0.86 | 78.14± 0.55 | **66.01** |
| **5% Feature Corruption** | | | | | | | |
| Random | 43.14±3.04 | 54.19±2.92 | 64.21±2.39 | 69.50±1.06 | 72.90±0.52 | 77.26±0.39 | 60.79 |
| Herding | 42.50±1.27 | 53.88±3.07 | 60.54±0.94 | 69.15±0.55 | 73.47±0.89 | 77.26±0.39 | 59.81 |
| Forgetting | 32.42±0.74 | 49.72±1.64 | 54.84±2.20 | 70.22±2.00 | 75.19±0.40 | 77.26±0.39 | 56.48 |
| GraNd-score | 42.24±0.57 | 53.48±0.76 | 60.17±1.66 | 69.16±0.81 | 73.35±0.81 | 77.26±0.39 | 59.68 |
| EL2N-score | 26.13±1.75 | 39.01±1.42 | 49.89±1.87 | 68.36±1.41 | 73.10±0.36 | 77.26±0.39 | 51.30 |
| Optimization-based | 38.25±3.04 | 50.88±6.07 | 57.26±0.93 | 68.02±0.39 | 73.77±0.56 | 77.26±0.39 | 57.64 |
| Self-sup.-selection | 44.24±0.48 | 55.99±1.21 | 61.03±0.59 | 69.96±1.07 | 74.56±1.17 | 77.26±0.39 | 61.16 |
| Moderate-DS | 46.78±1.90 | 57.36±1.22 | 65.40±1.19 | 71.46±0.19 | **75.64±0.61** | 77.26±0.39 | 63.33 |
| **GM Matching** | **49.50±0.72** | **60.23±0.88** | **66.25±0.51** | **72.91±0.26** | 75.10±0.29 | 77.26±0.39 | **64.80** |
| **10% Feature Corruption** | | | | | | | |
| Random | 43.27±3.01 | 53.94±2.78 | 62.17±1.29 | 68.41±1.21 | 73.50±0.73 | 76.50±0.63 | 60.26 |
| Herding | 44.34±1.07 | 53.31±1.49 | 60.13±0.38 | 68.20±0.74 | 74.34±1.07 | 76.50±0.63 | 60.06 |
| Forgetting | 30.43±0.70 | 47.50±1.43 | 53.16±0.44 | 70.36±0.82 | 75.11±0.71 | 76.50±0.63 | 55.31 |
| GraNd-score | 36.36±1.06 | 52.26±0.66 | 60.22±1.39 | 68.96±0.62 | 72.78±0.51 | 76.50±0.63 | 58.12 |
| EL2N-score | 21.75±1.56 | 30.80±2.23 | 41.06±1.23 | 64.82±1.48 | 73.47±1.30 | 76.50±0.63 | 46.38 |
| Optimization-based | 37.22±0.39 | 48.92±1.38 | 56.88±1.48 | 67.33±2.15 | 72.94±1.90 | 76.50±0.63 | 56.68 |
| Self-sup.-selection | 42.01±1.31 | 54.47±1.19 | 61.37±0.68 | 68.52±1.24 | 74.73±0.36 | 76.50±0.63 | 60.22 |
| Moderate-DS | 47.02±0.66 | 55.60±1.67 | 62.18±1.86 | 71.83±0.78 | **75.66±0.66** | 76.50±0.63 | 62.46 |
| **GM Matching** | **48.86±1.02** | **60.15±0.43** | **66.92±0.28** | **72.03±0.38** | 73.71±0.19 | 76.50±0.63 | **64.33** |
| **20% Feature Corruption** | | | | | | | |
| Random | 40.99±1.46 | 50.38±1.39 | 57.24±0.65 | 65.21±1.31 | 71.74±0.28 | 74.92±0.88 | 57.11 |
| Herding | 44.42±0.46 | 53.57±0.31 | 60.72±1.78 | 69.09±1.73 | 73.08±0.98 | 74.92±0.88 | 60.18 |
| Forgetting | 26.39±0.17 | 40.78±2.02 | 49.95±2.31 | 65.71±1.12 | 73.67±1.12 | 74.92±0.88 | 51.30 |
| GraNd-score | 36.33±2.66 | 46.21±1.48 | 55.51±0.76 | 64.59±2.40 | 70.14±1.36 | 74.92±0.88 | 54.56 |
| EL2N-score | 21.64±2.03 | 23.78±1.66 | 35.71±1.17 | 56.32±0.86 | 69.66±0.43 | 74.92±0.88 | 41.42 |
| Optimization-based | 33.42±1.60 | 45.37±2.81 | 54.06±1.74 | 65.19±1.27 | 70.06±0.83 | 74.92±0.88 | 54.42 |
| Self-sup.-selection | 42.61±2.44 | 54.04±1.90 | 59.51±1.22 | 68.97±0.96 | 72.33±0.20 | 74.92±0.88 | 60.01 |
| Moderate-DS | 42.98±0.87 | 55.80±0.95 | 61.84±1.96 | 70.05±1.29 | 73.67±0.30 | 74.92±0.88 | 60.87 |
| **GM Matching** | **47.12±0.64** | **59.17±0.92** | **63.45±0.34** | **71.70±0.60** | **74.60±1.03** | 74.92±0.88 | **63.21** |

Table 5: **Image Corruption ( CIFAR 100 ):** Comparing (Test Accuracy) pruning methods when 20% of the images are corrupted. ResNet-50 is used both as proxy and for downstream classification.

| Tiny ImageNet | | | | | | |
|---|---|---|---|---|---|---|
| **Method / Ratio** | **20%** | **30%** | **40%** | **60%** | **80%** | **100%** | **Mean ↑** |
| **No Corruption** | | | | | | |
| Random | 24.02±0.41 | 29.79±0.27 | 34.41±0.46 | 40.96±0.47 | 45.74±0.61 | 49.36±0.25 | 34.98 |
| Herding | 24.09±0.45 | 29.39±0.53 | 34.13±0.37 | 40.86±0.61 | 45.45±0.33 | 49.36±0.25 | 34.78 |
| Forgetting | 22.37±0.71 | 28.67±0.54 | 33.64±0.32 | 41.14±0.43 | **46.77±0.31** | 49.36±0.25 | 34.52 |
| GraNd-score | 23.56±0.52 | 29.66±0.37 | 34.33±0.50 | 40.77±0.42 | 45.96±0.56 | 49.36±0.25 | 34.86 |
| EL2N-score | 19.74±0.26 | 26.58±0.40 | 31.93±0.28 | 39.12±0.46 | 45.32±0.27 | 49.36±0.25 | 32.54 |
| Optimization-based | 13.88±2.17 | 23.75±1.62 | 29.77±0.94 | 37.05±2.81 | 43.76±1.50 | 49.36±0.25 | 29.64 |
| Self-sup.-selection | 20.89±0.42 | 27.66±0.50 | 32.50±0.30 | 39.64±0.39 | 44.94±0.34 | 49.36±0.25 | 33.13 |
| Moderate-DS | 25.29±0.38 | 30.57±0.20 | 34.81±0.51 | 41.45±0.44 | 46.06±0.33 | 49.36±0.25 | 35.64 |
| **GM Matching** | **27.88±0.19** | **33.15±0.26** | **36.92±0.40** | **42.48±0.12** | 46.75±0.51 | 49.36±0.25 | **37.44** |
| **5% Feature Corruption** | | | | | | |
| Random | 23.51±0.22 | 28.82±0.72 | 32.61±0.68 | 39.77±0.35 | 44.37±0.34 | 49.02±0.35 | 33.82 |
| Herding | 23.09±0.53 | 28.67±0.37 | 33.09±0.32 | 39.71±0.31 | 45.04±0.15 | 49.02±0.35 | 33.92 |
| Forgetting | 21.36±0.28 | 27.72±0.43 | 33.45±0.21 | 40.92±0.45 | 45.99±0.51 | 49.02±0.35 | 33.89 |
| GraNd-score | 22.47±0.23 | 28.85±0.83 | 33.81±0.24 | 40.40±0.15 | 44.86±0.49 | 49.02±0.35 | 34.08 |
| EL2N-score | 18.98±0.72 | 25.96±0.28 | 31.07±0.63 | 38.65±0.36 | 44.21±0.68 | 49.02±0.35 | 31.77 |
| Optimization-based | 13.65±1.26 | 24.02±1.35 | 29.65±1.86 | 36.55±1.84 | 43.64±0.71 | 49.02±0.35 | 29.50 |
| Self-sup.-selection | 19.35±0.57 | 26.11±0.31 | 31.90±0.37 | 38.91±0.29 | 44.43±0.42 | 49.02±0.35 | 32.14 |
| Moderate-DS | 24.63±0.78 | 30.27±0.16 | 34.84±0.24 | 40.86±0.42 | 45.60±0.31 | 49.02±0.35 | 35.24 |
| **GM Matching** | **27.46±1.22** | **33.14±0.61** | **35.76±1.14** | **41.62±0.71** | **46.83±0.56** | 49.02±0.35 | **36.96** |
| **10% Feature Corruption** | | | | | | |
| Random | 22.67±0.27 | 28.67±0.52 | 31.88±0.30 | 38.63±0.36 | 43.46±0.20 | 48.40±0.32 | 33.06 |
| Herding | 22.01±0.18 | 27.82±0.11 | 31.82±0.26 | 39.37±0.18 | 44.18±0.27 | 48.40±0.32 | 33.04 |
| Forgetting | 20.06±0.48 | 27.17±0.36 | 32.31±0.22 | 40.19±0.29 | 45.51±0.48 | 48.40±0.32 | 33.05 |
| GraNd-score | 21.52±0.48 | 26.98±0.43 | 32.70±0.19 | 40.03±0.26 | 44.87±0.35 | 48.40±0.32 | 33.22 |
| EL2N-score | 18.59±0.13 | 25.23±0.18 | 30.37±0.22 | 38.44±0.32 | 44.32±1.07 | 48.40±0.32 | 31.39 |
| Optimization-based | 14.05±1.74 | 29.18±1.77 | 29.12±0.61 | 36.28±1.88 | 43.52±0.31 | 48.40±0.32 | 29.03 |
| Self-sup.-selection | 19.47±0.26 | 26.51±0.55 | 31.78±0.14 | 38.87±0.54 | 44.69±0.29 | 48.40±0.32 | 32.26 |
| Moderate-DS | 23.79±0.16 | 29.56±0.16 | 34.60±0.12 | 40.36±0.27 | 45.10±0.23 | 48.40±0.32 | 34.68 |
| **GM Matching** | **27.41±0.23** | **32.84±0.98** | **36.27±0.68** | **41.85±0.29** | **46.35±0.44** | 48.40±0.32 | **36.94** |
| **20% Feature Corruption** | | | | | | |
| Random | 19.99±0.42 | 25.93±0.53 | 30.83±0.44 | 37.98±0.31 | 42.96±0.62 | 46.68±0.43 | 31.54 |
| Herding | 19.46±0.14 | 24.47±0.33 | 29.72±0.39 | 37.50±0.59 | 42.28±0.30 | 46.68±0.43 | 30.86 |
| Forgetting | 18.47±0.46 | 25.53±0.23 | 31.17±0.24 | 39.35±0.44 | 44.55±0.67 | 46.68±0.43 | 31.81 |
| GraNd-score | 20.07±0.49 | 26.68±0.40 | 31.25±0.40 | 38.21±0.49 | 42.84±0.72 | 46.68±0.43 | 30.53 |
| EL2N-score | 18.57±0.30 | 24.42±0.44 | 30.04±0.15 | 37.62±0.44 | 42.43±0.61 | 46.68±0.43 | 30.53 |
| Optimization-based | 13.71±0.26 | 23.33±1.84 | 29.15±2.84 | 36.12±1.86 | 42.94±0.52 | 46.88±0.43 | 29.06 |
| Self-sup.-selection | 20.22±0.23 | 26.90±0.50 | 31.93±0.49 | 39.74±0.52 | 44.27±0.10 | 46.68±0.43 | 32.61 |
| Moderate-DS | 23.27±0.33 | 29.06±0.36 | 33.48±0.11 | 40.07±0.36 | 44.73±0.39 | 46.68±0.43 | 34.12 |
| **GM Matching** | **27.19±0.92** | **31.70±0.78** | **35.14±0.19** | **42.04±0.31** | **45.12±0.28** | 46.68±0.43 | **36.24** |

Table 6: **Image Corruption ( Tiny ImageNet ):** Comparing (Test Accuracy) pruning methods under feature (image) corruption. ResNet-50 is used both as proxy and for downstream classification.

| Method / Ratio | CIFAR-100 (Label noise) | | Tiny ImageNet (Label noise) | | Mean ↑ |
| --- | --- | --- | --- | --- | --- |
| | 20% | 30% | 20% | 30% | |
| **20% Label Noise** | | | | | |
| Random | 34.47±0.64 | 43.26±1.21 | 17.78±0.44 | 23.88±0.42 | 29.85 |
| Herding | 42.29±1.75 | 50.52±3.38 | 18.98±0.44 | 24.23±0.29 | 34.01 |
| Forgetting | 36.53±1.11 | 45.78±1.04 | 13.20±0.38 | 21.79±0.43 | 29.33 |
| GraNd-score | 31.72±0.67 | 42.80±0.30 | 18.28±0.32 | 23.72±0.18 | 28.05 |
| EL2N-score | 29.82±1.19 | 33.62±2.35 | 13.93±0.69 | 18.57±0.31 | 23.99 |
| Optimization-based | 32.79±0.62 | 41.80±1.14 | 14.77±0.95 | 22.52±0.77 | 27.57 |
| Self-sup.-selection | 31.08±0.78 | 41.87±0.63 | 15.10±0.73 | 21.01±0.36 | 27.27 |
| Moderate-DS | 40.25±0.12 | 48.53±1.60 | 19.64±0.40 | 24.96±0.30 | 31.33 |
| **GM Matching** | **52.64±0.72** | **61.01±0.47** | **25.80±0.37** | **31.71±0.24** | **42.79** |
| **35% Label Noise** | | | | | |
| Random | 24.51±1.34 | 32.26±0.81 | 14.64±0.29 | 19.41±0.45 | 22.71 |
| Herding | 29.42±1.54 | 37.50±2.12 | 15.14±0.45 | 20.19±0.45 | 25.56 |
| Forgetting | 29.48±1.98 | 38.01±2.21 | 11.25±0.90 | 17.07±0.66 | 23.14 |
| GraNd-score | 23.03±1.05 | 34.83±2.01 | 13.68±0.46 | 19.51±0.45 | 22.76 |
| EL2N-score | 21.95±1.08 | 31.63±2.84 | 10.11±0.25 | 13.69±0.32 | 19.39 |
| Optimization-based | 26.77±0.15 | 35.63±0.92 | 12.37±0.68 | 18.52±0.90 | 23.32 |
| Self-sup.-selection | 23.12±1.47 | 34.85±0.68 | 11.23±0.32 | 17.76±0.69 | 22.64 |
| Moderate-DS | 28.45±0.53 | 36.55±1.26 | 15.27±0.31 | 20.33±0.28 | 25.15 |
| **GM Matching** | **43.33± 1.02** | **58.41± 0.68** | **23.14± 0.92** | **27.76± 0.40** | **38.16** |

Table 7: **Robustness to Label Noise:** Comparing (Test Accuracy) pruning methods on CIFAR-100 and TinyImageNet datasets, under 20% and 35% Symmetric Label Corruption, at 20% and 30% selection ratio. ResNet-50 is used both as proxy and for downstream classification.

pixelation artifacts.

• *Fog*: Emulates atmospheric distortions by overlaying a simulated fog effect – resulting in reduced contrast and visibility.

• *Motion Blur*: Models dynamic distortions caused by camera motion or moving objects during exposure.

To introduce diverse corruption across the dataset while ensuring a balanced distribution, we apply each corruption type uniformly at random to the corrupted samples. Instead of assigning fixed partitions, this approach ensures that each sample has an equal probability of being affected by any of the five corruption types. This stochastic allocation results in a heterogeneous mix of corruptions, compelling models to generalize across multiple degradation patterns rather than overfitting to a specific type of distortion.

The results of our experiments, presented in Tables 5 and 6, evaluate the impact of increasing corruption levels (5%, 10%, and 20%) on various pruning methods. Across both CIFAR-100 and Tiny ImageNet, GM MATCHING consistently outperforms all baselines, achieving an average accuracy improvement of ≈ 3% over the next-best approach.

This performance gap is amplified at higher corruption levels, where GM MATCHING maintains superior test accuracy. Moreover, the gains are also significant at aggressive pruning ratios (20%-40%), where GM MATCHING improves test accuracy by 2-4% over baselines. This trend aligns with its strong performance in data-scarce settings, reinforcing its ability to preserve robustness even when training data is significantly reduced.

Among baselines, Random Selection exhibits a steady performance drop with increasing corruption, confirming its inability to retain robustness. Herding and Moderate-DS, while effective in standard settings, struggle under high corruption levels. Forgetting and Optimization-based methods show inconsistent results, likely due to their reliance on training dynamics that become unstable when corrupted samples are introduced. GraNd-score and EL2N-score, which prioritize loss-based selection, perform well in clean settings but degrade significantly under corruption, suggesting vulnerability to adversarial perturbations. Self-supervised Selection remains competitive but fails to match GM MATCHING, particularly at higher corruption intensities.

| Tiny ImageNet (Label Noise) | | | | | | |
|---|---|---|---|---|---|---|
| **Method / Ratio** | **20%** | **30%** | **40%** | **60%** | **80%** | **100%** | **Mean ↑** |
| Random | 17.78±0.44 | 23.88±0.42 | 27.97±0.39 | 34.88±0.51 | 38.47±0.40 | 44.42±0.47 | 28.60 |
| Herding | 18.98±0.44 | 24.23±0.29 | 27.28±0.31 | 34.36±0.29 | 39.00±0.49 | 44.42±0.47 | 28.87 |
| Forgetting | 13.20±0.38 | 21.79±0.43 | 27.89±0.22 | **36.03±0.24** | **40.60±0.31** | 44.42±0.47 | 27.50 |
| GraNd-score | 18.28±0.32 | 23.72±0.18 | 27.34±0.33 | 34.91±0.19 | 39.45±0.45 | 44.42±0.47 | 28.34 |
| EL2N-score | 13.93±0.69 | 18.57±0.31 | 24.56±0.34 | 32.14±0.49 | 37.64±0.41 | 44.42±0.47 | 25.37 |
| Optimization-based | 14.77±0.95 | 22.52±0.77 | 25.62±0.90 | 34.18±0.79 | 38.49±0.69 | 44.42±0.47 | 27.12 |
| Self-sup.-selection | 15.10±0.73 | 21.01±0.36 | 26.62±0.22 | 33.93±0.36 | 39.22±0.12 | 44.42±0.47 | 27.18 |
| Moderate-DS | **19.64±0.40** | **24.96±0.30** | **29.56±0.21** | 35.79±0.36 | 39.93±0.23 | 44.42±0.47 | 30.18 |
| **GM Matching** | **25.80±0.37** | **31.71±0.24** | **34.87±0.21** | **39.76±0.71** | **41.94±0.23** | 44.42±0.47 | **34.82** |

Table 8: **Pruning with Label Noise (TinyImageNet):** Comparing (Test Accuracy) pruning methods under 20% Symmetric Label Corruption across wide array of selection ratio. ResNet-50 is used both as proxy and for downstream classification.

### I.3.4. ROBUSTNESS TO LABEL CORRUPTION

Next, we consider label noise – a prevalent issue in real-world datasets, where obtaining perfectly annotated data is impractical due to human labeling errors, dataset aggregation inconsistencies, or adversarial data manipulations. Consequently, it is crucial to assess the ability of data pruning methods to select informative samples while filtering out mislabeled examples, ensuring robustness to noisy annotations.

To systematically evaluate pruning methods under label noise, we introduce synthetically injected symmetric label noise, a widely used corruption paradigm in robust learning (Li et al., 2022; Patrini et al., 2017; Xia et al., 2020). In this setting, a fraction of training labels is randomly flipped to a different class, simulating annotation errors encountered in large-scale weakly labeled datasets. We experiment with two corruption levels (20% and 35% label noise) and report test accuracy across different pruning ratios.

Interestingly, in both CIFAR-100 and Tiny ImageNet, training ResNet50 in a subset selected by GM MATCHING outperforms all competing methods by a large margin ( Table 7,8). For instance, at 20% label noise, when selecting 30% of samples from CIFAR-100, GM MATCHING achieves 61.01% accuracy, compared to 48.53% (Moderate-DS) and 45.78% (Forgetting). At 35% label noise, the performance gap widens, with GM MATCHING achieving 58.41%, while the closest baseline (Moderate-DS) lags behind at 36.55%. Similar trends hold for Tiny ImageNet, where GM MATCHING outperforms the best baseline by 4-6% across different noise levels. Overall, on average GM MATCHING **outperforms the baselines by** $\approx$ **12%** when selecting a small $20 - 30\%$. Moreover, when Vision Transformers are trained on a 20% subset of CIFAR-10 data chosen by GM MATCHING, they achieve $\approx 10\%$ improvement over previous robust data selection algorithms (Table 10).

Since mislabeled examples originate from random class assignments, they tend to be spatially dissimilar from their intended class distributions. As a result, GM MATCHING is less likely to select such noisy examples, leading to improved generalization despite high noise levels.

### I.3.5. ROBUSTNESS TO ADVERSARIAL ATTACKS

Adversarial attacks pose a fundamental challenge to the reliability of deep learning models, as they introduce imperceptible but highly effective perturbations to input samples, forcing misaligned predictions (Szegedy et al., 2013; Huang et al., 2010).

To assess the robustness of data pruning methods in adversarial settings, we experiment with two widely used attack techniques:

- *Projected Gradient Descent (PGD)* (Madry et al., 2017): A strong iterative attack that optimizes perturbations by taking multiple gradient ascent steps, maximizing the model's loss function.

- *Gradient Sign Attack (GS)* (Goodfellow et al., 2014): A single-step adversarial attack that perturbs the input along the gradient of the loss function, often serving as a computationally efficient alternative to PGD.

Using these attack methods, we generate adversarial examples from models trained on CIFAR-100 and Tiny ImageNet. We then apply different data pruning strategies to these adversarial datasets and retrain models on the curated subsets to analyze

| Method / Ratio | CIFAR-100 (PGD Attack) | | CIFAR-100 (GS Attack) | | Mean ↑ |
|---|---|---|---|---|---|
| | 20% | 30% | 20% | 30% | |
| Random | 43.23±0.31 | 52.86±0.34 | 44.23±0.41 | 53.44±0.44 | 48.44 |
| Herding | 40.21±0.72 | 49.62±0.65 | 39.92±1.03 | 50.14±0.15 | 44.97 |
| Forgetting | 35.90±1.30 | 47.37±0.99 | 37.55±0.53 | 46.88±1.91 | 41.93 |
| GraNd-score | 40.87±0.84 | 50.13±0.30 | 40.77±1.11 | 49.88±0.83 | 45.41 |
| EL2N-score | 26.61±0.58 | 34.50±1.02 | 26.72±0.66 | 35.55±1.30 | 30.85 |
| Optimization-based | 38.29±1.77 | 46.25±1.82 | 41.36±0.92 | 49.10±0.81 | 43.75 |
| Self-sup.-selection | 40.53±1.15 | 49.95±0.50 | 40.74±1.66 | 51.23±0.25 | 45.61 |
| Moderate-DS | 43.60±0.97 | 51.66±0.39 | 44.69±0.68 | 53.71±0.37 | 48.42 |
| **GM Matching** | **45.41 ±0.86** | **51.80 ±1.01** | **49.78 ±0.27** | **55.50 ±0.31** | **50.62** |

| Method / Ratio | Tiny ImageNet (PGD Attack) | | Tiny ImageNet (GS Attack) | | Mean ↑ |
|---|---|---|---|---|---|
| | 20% | 30% | 20% | 30% | |
| Random | 20.93±0.30 | 26.60±0.98 | 22.43±0.31 | 26.89±0.31 | 24.21 |
| Herding | 21.61±0.36 | 25.95±0.19 | 23.04±0.28 | 27.39±0.14 | 24.50 |
| Forgetting | 20.38±0.47 | 26.12±0.19 | 22.06±0.31 | 27.21±0.21 | 23.94 |
| GraNd-score | 20.76±0.21 | 26.34±0.32 | 22.56±0.30 | 27.52±0.40 | 24.30 |
| EL2N-score | 16.67±0.62 | 22.36±0.42 | 19.93±0.57 | 24.65±0.32 | 20.93 |
| Optimization-based | 19.26±0.77 | 24.55±0.92 | 21.26±0.24 | 25.88±0.37 | 22.74 |
| Self-sup.-selection | 19.23±0.46 | 23.92±0.51 | 19.70±0.20 | 24.73±0.39 | 21.90 |
| Moderate-DS | 21.81±0.37 | 27.11±0.20 | 23.20±0.13 | 28.89±0.27 | 25.25 |
| **GM Matching** | **25.98 ±1.12** | **30.77 ±0.25** | **29.71 ±0.45** | **32.88 ±0.73** | **29.84** |

Table 9: **Robustness to Adversarial Attacks**. Comparing (Test Accuracy) pruning methods under PGD and GS attacks. ResNet-50 is used both as proxy and for downstream classification.

their effectiveness in retaining robustness. The results with ResNet-50 are summarized in Table 9, and the experiments with ViT are presented in Table 10.

Similar to other corruption scenarios, across both datasets and attack types, GM MATCHING consistently outperforms all baseline pruning methods, yielding an average accuracy improvement of 3% over the next-best approach. Specifically, under PGD attacks on CIFAR-100, GM MATCHING achieves 45.41% accuracy at 20% selection, compared to 43.60% (Moderate-DS) and 40.87% (GraNd-score). Similarly, in Tiny ImageNet under GS attacks, GM MATCHING maintains the highest mean accuracy of 29.84%, outperforming Moderate-DS (25.25%) and Self-supervised Selection (21.90%).

## I.3.6. ABLATIONS WITH PROXY ENCODER

Since the input features (e.g. images) often reside on a non-separable manifold, data pruning strategies rely on proxy models to map raw input samples into a separable embedding space, where importance scores can be assigned more effectively.

**A. IDEALIZED SETTING :** In the standard setting, the proxy model used for sample selection is identical to the model used for downstream training—both in terms of architecture and dataset. This represents an idealized scenario, where the feature representations used to evaluate sample importance remain consistent throughout training. Since the proxy and final models are identical, performance differences between pruning methods directly reflect the effectiveness of sample selection, rather than being confounded by architectural mismatches. Since no external factors (such as domain shifts) interfere, the results from this setup serve as a benchmark for understanding the best-case scenario for sample selection.

All the experiments reported in Tables 5–9 follow this framework: We perform data pruning on CIFAR-100 and Tiny ImageNet, using ResNet-50 as both the proxy model (for selecting samples) and the downstream classifier (for final training). The proxy model assigns importance scores to training samples, and a subset is selected accordingly. A new ResNet-50 is then trained from scratch on the pruned dataset, ensuring that no information leakage occurs between sample selection and final training. This methodology remains consistent across clean data, label noise, feature corruption, and adversarial attack settings, providing a direct comparison of how pruning strategies perform across diverse learning conditions.

By maintaining this controlled setup, we isolate the true impact of different pruning strategies, ensuring that their ability to retain the most valuable and generalizable training samples is the sole factor influencing performance. This establishes a strong foundation before extending the analysis to more complex settings involving distribution shifts and network

| CIFAR-100 (ViT-S) | | | | | |
|---|---|---|---|---|---|
| **Method** | **No Corruption** | **Noisy Feature** | **Label Noise** | **Adv. Attack** | **Mean ↑** |
| Random | 33.80±0.54 | 31.29±0.61 | 26.67±0.54 | 31.01±0.45 | 30.19 |
| Herding | 32.16±0.37 | 31.75±0.22 | 32.27±0.53 | 31.28±0.66 | 31.37 |
| Forgetting | 33.52±0.73 | 24.45±0.29 | 26.24±1.07 | 28.26±1.95 | 28.12 |
| GraNd-score | 22.49±0.47 | 18.40±0.11 | 22.13±0.90 | 19.27±1.27 | 20.07 |
| EL2N-score | 26.15±0.21 | 23.27±0.68 | 24.80±0.72 | 20.26±1.68 | 23.12 |
| Optimization-based | 31.84±0.63 | 30.12±0.73 | 30.12±0.70 | 29.36±0.75 | 30.36 |
| Self-sup.-selection | 33.35±0.31 | 30.72±0.90 | 29.16±0.27 | 28.49±0.56 | 30.93 |
| Moderate-DS | 34.43±0.32 | 32.73±0.35 | 31.86±0.49 | 32.61±0.40 | 32.91 |
| **GM MATCHING** | **40.81±0.87** | **38.26±0.68** | **42.11±0.36** | **39.45±0.82** | **40.66** |

Table 10: **IMAGE CLASSIFICATION (VIT-S)**: We compare the downstream test accuracy of various data selection methods on simulated CIFAR-100 (using ViT-S) under different corruption types: No Corruption, Noisy Feature, Label Noise, and Adversarial Attack. All experiments are performed with a fixed selection ratio of 20%, using ResNet-50 as both the proxy and downstream classifier. Our method, GM MATCHING, consistently outperforms all baselines—demonstrating superior robustness to corrupted data, with larger performance gains at higher corruption levels. Mean and standard deviation (%) are reported over multiple runs. The best result in each case is highlighted in **bold**.

| | ResNet-50→SENet | | ResNet-50→EfficientNet-B0 | | |
|---|---|---|---|---|---|
| **Method / Ratio** | **20%** | **30%** | **20%** | **30%** | **Mean ↑** |
| Random | 34.13±0.71 | 39.57±0.53 | 32.88±1.52 | 39.11±0.94 | 36.42 |
| Herding | 34.86±0.55 | 38.60±0.68 | 32.21±1.54 | 37.53±0.22 | 35.80 |
| Forgetting | 33.40±0.64 | 39.79±0.78 | 31.12±0.21 | 38.38±0.65 | 35.67 |
| GraNd-score | 35.12±0.54 | 41.14±0.42 | 33.20±0.67 | 40.02±0.35 | 37.37 |
| EL2N-score | 31.08±1.11 | 38.26±0.45 | 31.34±0.49 | 36.88±0.32 | 34.39 |
| Optimization-based | 33.18±0.52 | 39.42±0.77 | 32.16±0.90 | 38.52±0.50 | 35.82 |
| Self-sup.-selection | 31.74±0.71 | 38.45±0.39 | 30.99±1.03 | 37.96±0.77 | 34.79 |
| Moderate-DS | 36.04±0.15 | 41.40±0.20 | 34.26±0.48 | 39.57±0.29 | 37.82 |
| **GM Matching** | **37.93±0.23** | **42.59±0.29** | **36.31±0.67** | **41.03±0.41** | **39.47** |

Table 11: **NETWORK TRANSFER (NO CORRUPTION) PROXY ENCODER**: (Tiny-ImageNet) Model Transfer Results. A ResNet-50 proxy is used to find important samples which are then used to train SENet and EfficientNet.

mismatches.

However, a critical question remains:

*How well do samples selected by a proxy model generalize when trained on a different architecture or dataset ?*

In real-world applications, the model used for data selection (proxy model) is often different from the final model used for training due to hardware constraints, deployment considerations, or domain shifts. An ideal pruning strategy should ensure that the selected subset remains highly effective, even when the final model differs from the one used during selection.

To investigate this, we perform comprehensive ablation studies across multiple proxy selection scenarios, evaluating the robustness to distribution shifts and network mismatches i.e. to say that samples selected via a proxy network should generalize well when trained on unseen (during sample selection) networks / domains.

**B. GENERALIZATION TO UNSEEN NETWORK :** In this setting, the proxy model and downstream classifier are trained on the same dataset, meaning there is no distribution shift. However, the proxy model's architecture differs from the final training model, testing whether a pruned subset remains useful across different network designs. This scenario simulates practical constraints where a proxy model is used for data selection, but the final model needs to be optimized for different architectural properties (e.g., mobile-friendly architectures).

In Table 11, we use a ResNet-50 proxy trained on TinyImageNet (no corruption) to select samples from the same dataset for downstream training. Instead of using ResNet-50 for final training, we train with different architectures on the pruned subset:

| Method / Ratio | ResNet-50→ VGG-16 | | ResNet-50→ ShuffleNet | | Mean ↑ |
|---|---|---|---|---|---|
| | 20% | 30% | 20% | 30% | |
| **No Corruption** | | | | | |
| Random | 29.63±0.43 | 35.38±0.83 | 32.40±1.06 | 39.13±0.81 | 34.96 |
| Herding | 31.05±0.22 | 36.27±0.57 | 33.10±0.39 | 38.65±0.22 | 35.06 |
| Forgetting | 27.53±0.36 | 35.61±0.39 | 27.82±0.56 | 36.26±0.51 | 32.35 |
| GraNd-score | 29.93±0.95 | 35.61±0.39 | 29.56±0.46 | 37.40±0.38 | 33.34 |
| EL2N-score | 26.47±0.31 | 33.19±0.51 | 28.18±0.27 | 35.81±0.29 | 31.13 |
| Optimization-based | 25.92±0.64 | 34.82±1.29 | 31.37±1.14 | 38.22±0.78 | 32.55 |
| Self-sup.-selection | 25.16±1.10 | 33.30±0.94 | 29.47±0.56 | 36.68±0.36 | 31.45 |
| Moderate-DS | 31.45±0.32 | 37.89±0.36 | 33.32±0.41 | 39.68±0.34 | 35.62 |
| **GM Matching** | **35.86±0.41** | **40.56±0.22** | **35.51±0.32** | **40.30±0.58** | **38.47** |
| **20% Label Corruption** | | | | | |
| Random | 23.29±1.12 | 28.18±1.84 | 25.08±1.32 | 31.44±1.21 | 27.00 |
| Herding | 23.99±0.36 | 28.57±0.40 | 26.25±0.47 | 30.73±0.28 | 27.39 |
| Forgetting | 14.52±0.66 | 21.75±0.23 | 15.70±0.29 | 22.31±0.35 | 18.57 |
| GraNd-score | 22.44±0.46 | 27.95±0.29 | 23.64±0.10 | 30.85±0.21 | 26.22 |
| EL2N-score | 15.15±1.25 | 23.36±0.30 | 18.01±0.44 | 24.68±0.34 | 20.30 |
| Optimization-based | 22.93±0.58 | 24.92±2.50 | 25.82±1.70 | 30.19±0.48 | 25.97 |
| Self-sup.-selection | 18.39±1.30 | 25.77±0.87 | 22.87±0.54 | 29.80±0.36 | 24.21 |
| Moderate-DS | 23.68±0.19 | 28.93±0.19 | 28.82±0.33 | 32.39±0.21 | 28.46 |
| **GM Matching** | **28.77±0.77** | **34.87±0.23** | **32.05±0.93** | **37.43±0.25** | **33.28** |
| **20% Feature Corruption** | | | | | |
| Random | 26.33±0.88 | 31.57±1.31 | 29.15±0.83 | 34.72±1.00 | 30.44 |
| Herding | 18.03±0.33 | 25.77±0.34 | 23.33±0.43 | 31.73±0.38 | 24.72 |
| Forgetting | 19.41±0.57 | 28.35±0.16 | 18.44±0.57 | 31.09±0.61 | 24.32 |
| GraNd-score | 23.59±0.19 | 30.69±0.13 | 23.15±0.56 | 31.58±0.95 | 27.25 |
| EL2N-score | 24.60±0.81 | 31.49±0.33 | 26.62±0.34 | 33.91±0.56 | 29.16 |
| Optimization-based | 25.12±0.34 | 30.52±0.89 | 28.87±1.25 | 34.08±1.92 | 29.65 |
| Self-sup.-selection | 26.33±0.21 | 33.23±0.26 | 26.48±0.37 | 33.54±0.46 | 29.90 |
| Moderate-DS | 29.65±0.68 | 35.89±0.53 | 32.30±0.38 | 38.66±0.29 | 34.13 |
| GM Matching | **33.45±1.02** | **39.46±0.44** | **35.14±0.21** | **39.89±0.98** | **36.99** |
| **PGD Attack** | | | | | |
| Random | 26.12±1.09 | 31.98±0.78 | 28.28±0.90 | 34.59±1.18 | 30.24 |
| Herding | 26.76±0.59 | 32.56±0.35 | 28.87±0.48 | 35.43±0.22 | 30.91 |
| Forgetting | 24.55±0.57 | 31.83±0.36 | 23.32±0.37 | 31.82±0.15 | 27.88 |
| GraNd-score | 25.19±0.33 | 31.46±0.54 | 26.03±0.66 | 33.22±0.24 | 28.98 |
| EL2N-score | 21.73±0.47 | 27.66±0.32 | 22.66±0.35 | 29.89±0.64 | 25.49 |
| Optimization-based | 26.02±0.36 | 31.64±1.75 | 27.93±0.47 | 34.82±0.96 | 30.10 |
| Self-sup.-selection | 22.36±0.30 | 28.56±0.50 | 25.35±0.27 | 32.57±0.13 | 27.21 |
| Moderate-DS | 27.24±0.36 | 32.90±0.31 | 29.06±0.28 | 35.89±0.53 | 31.27 |
| **GM Matching** | **27.96±1.60** | **35.76±0.82** | **34.11±0.65** | **40.91±0.84** | **34.69** |

Table 12: **NETWORK TRANSFER (CORRUPTED) PROXY ENCODER**: : A ResNet-50 proxy (pretrained on TinyImageNet) is used to find important samples from Tiny-ImageNet; which is then used to train a VGGNet-16 and ShuffleNet. We repeat the experiment across multiple corruption settings - clean; 20% Feature / Label Corruption and PGD attack when 20% and 30% samples are selected.

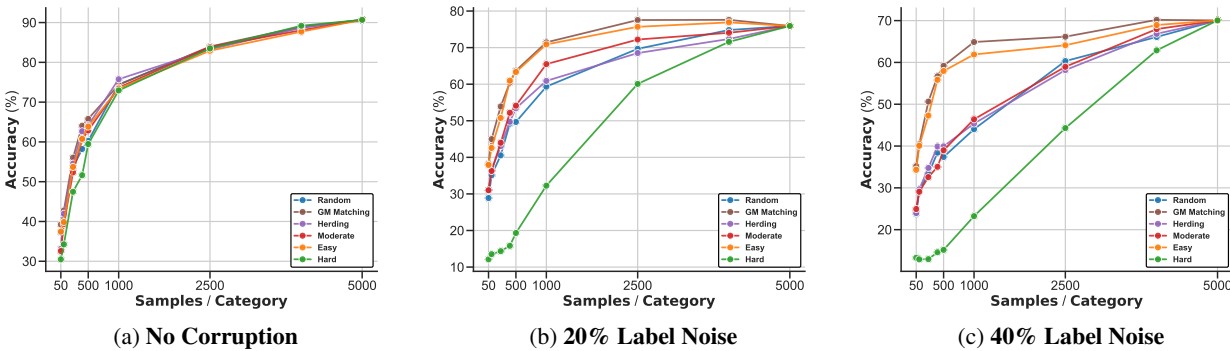

(a) **No Corruption**  (b) **20% Label Noise**  (c) **40% Label Noise**

Figure 16: DOMAIN TRANSFER ( IMAGENET-1K → CIFAR-10 ) PROXY ENCODER: CIFAR10, corrupted with label noise is pruned using a (proxy) ResNet-18 pretrained on ImageNet-1k. A ResNet-18 is trained from scratch on the subset. We compare our method GM MATCHING with geometric pruning baselines: UNIFORM, EASY, HARD, MODERATE, HERDING.

- SENet – A model optimized for channel-wise attention mechanisms, which enhances feature selection by adaptively recalibrating channel-wise responses.

- EfficientNet-B0 – A lightweight model designed for mobile and resource-efficient inference, using a combination of depthwise convolutions and width scaling to optimize performance while maintaining parameter efficiency.

- VGG-16 – A deep convolutional network with uniform architecture, known for its simple yet effective design, using fixed-size convolution filters and max pooling layers.

- ShuffleNet – A model specifically designed for speed and efficiency, utilizing group convolutions and channel shuffling to maximize accuracy while maintaining low computational overhead.

In Table 12, we also experiment with similar network mismatch scenarios, in the presence of various types and levels of corruption.

Both sets of experiments demonstrate that GM MATCHING consistently outperforms all pruning baselines, ensuring that selected samples remain informative and transferable across different network architectures. SENet and EfficientNet-B0 benefit the most from pruning, likely due to their adaptive feature recalibration and efficiency optimizations, while VGG-16 and ShuffleNet show greater sensitivity to pruning and corruption, struggling more under distribution shifts. Under corruption, loss-based selection methods (GraNd, EL2N) degrade significantly, whereas representative selection methods (Moderate-DS, Herding) hold up better under mild corruption but fail under severe noise.

These results highlight the importance of selecting subsets that generalize well, not only across different architectures but also under varying levels of corruption, reinforcing GM MATCHING as a highly robust and adaptable pruning strategy suitable for diverse deployment scenarios.

## C. GENERALIZATION TO UNSEEN DOMAIN.

In many real-world applications, deep learning models are pretrained on large-scale datasets (e.g., ImageNet) before being adapted to a different, often smaller, target dataset (e.g., CIFAR-10). This introduces a distribution shift between the dataset used for proxy-based sample selection and the dataset used for final training. If a pruning method is truly effective, it should be able to identify samples that generalize well across different data distributions, ensuring that the selected subset remains informative even if the proxy model was never explicitly trained on the target dataset.

To investigate this, we conduct experiments where the proxy model and downstream classifier share the same architecture, but the proxy model is pretrained on a different dataset, introducing a distribution shift. By keeping the architecture constant, we isolate the impact of dataset shift while avoiding confounding factors related to network differences.

In Figure 16, a ResNet-18 pretrained on ImageNet is used to select samples from CIFAR-10, which are then used to train a new ResNet-18 from scratch. This setup closely mirrors real-world transfer learning scenarios, where large-scale pretraining is leveraged to prune or curate training data for a smaller, domain-specific dataset before training a new model. The key

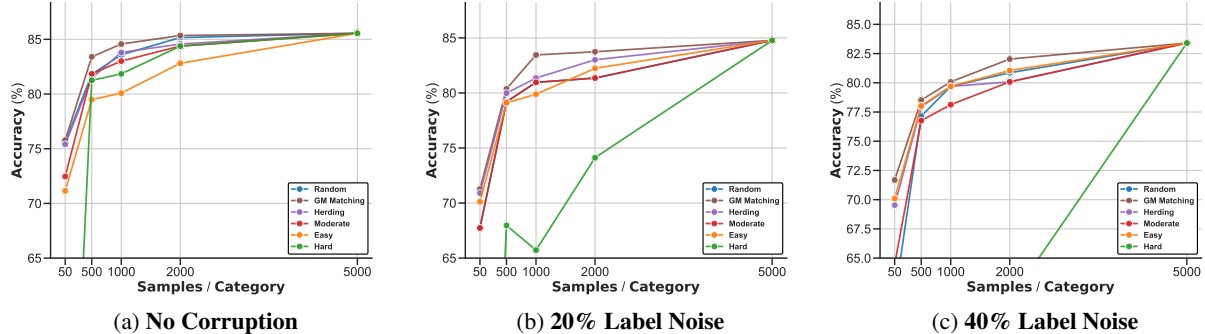

(a) **No Corruption**    (b) **20% Label Noise**    (c) **40% Label Noise**

Figure 17: DOMAIN TRANSFER ( IMAGENET-1K → CIFAR-10 ) PROXY ENCODER : We train a Linear Classifier on CIFAR10; over embeddings obtained from a frozen ResNet-18 pretrained on ImageNet-1k. The dataset was pruned using the same encoder. We compare GM MATCHING with several geometric pruning baselines ( Appendix I.1): Uniform, Easy, Hard, Moderate, Herding across no corruption, 20% and 30% label noise settings.

objective is to determine whether the subset selected by a proxy trained on a different dataset remains informative and structurally representative of the target dataset when used for full downstream training.

Furthermore, in Figure 17, we introduce an even more challenging setting by freezing the pretrained (on ImageNet) ResNet-18 and training only a linear classifier on top of the extracted CIFAR-10 features. Unlike the full fine-tuning approach, this method eliminates any feature adaptation between datasets, forcing the classifier to rely entirely on the quality of the selected samples. This makes it a more rigorous test of how well the pruned subsets inherently align with robust, transferable representations, ensuring that the selected data itself is informative, rather than merely benefiting from feature adaptation during training.

The results from Figure 16 17 demonstrate that GM MATCHING consistently outperforms all baselines, ensuring that selected samples remain highly informative and transferable despite the distribution shift. When fully fine-tuned, models trained on GM MATCHING pruned subsets achieve higher accuracy than those trained on subsets selected by other methods, indicating that it effectively identifies structurally important and generalizable samples. In the frozen feature setting, performance gaps between pruning strategies become even more pronounced, with Hard and Easy pruning strategies performing significantly worse, highlighting their reliance on feature adaptation rather than inherently meaningful sample selection. Loss-based methods (GraNd-score, EL2N-score) degrade under dataset shifts, suggesting that they may over-prioritize easy-to-learn samples that do not generalize well across domains. These findings emphasize that effective pruning strategies must go beyond dataset-specific heuristics and instead focus on selecting robust, transferable samples that remain useful even under feature extraction constraints, reinforcing the advantage of GM MATCHING in real-world transfer learning applications.

## D. GENERALIZATION TO BOTH UNSEEN DOMAIN AND NETWORK.

In many real-world scenarios, both domain and architecture mismatches occur simultaneously — a setting where the data selection model differs from the downstream training model both in terms of dataset and architecture. For instance, models like CLIP (Radford et al., 2021a) are often used as frozen, large-scale pretrained encoders on generic web-scale data (e.g., YFCC), whereas final deployment models may be lightweight CNNs (e.g., ResNet-18) trained from scratch on task-specific datasets like CIFAR.

To assess the robustness of pruning strategies in this challenging setting, we construct an experiment where a frozen CLIP ViT-B/32 encoder — pretrained on a generic dataset — is used as a proxy to embed CIFAR-10 samples and perform subset selection via GM MATCHING. A ResNet-18 model is then trained from scratch on the selected subset using only CIFAR-10 labels. This setup introduces both a *domain shift* (CLIP never trained on CIFAR-10) and a *network shift* (ViT to ResNet).

We evaluate performance across multiple corruption regimes: a clean dataset, as well as with 20% and 40% label noise. Table 13 presents the results across three subset sizes (10, 100, 1000 samples per class), showing the classification accuracy of ResNet-18 trained on these subsets.

Across all corruption levels and subset sizes, GM MATCHING consistently outperforms all baselines. For instance, in the 40% label noise setting, GM MATCHING achieves 43.7% mean accuracy — a full 3% higher than the next best method

| Method / Size | CIFAR-10 (ViT-B/32 $\rightarrow$ ResNet-18) | | | |
|---|---|---|---|---|
| | 10 | 100 | 1000 | Mean $\uparrow$ |
| **No Corruption** | | | | |
| Random | $23.2 \pm 3.9$ | $42.4 \pm 2.3$ | $86.8 \pm 0.7$ | 50.8 |
| Easy | $27.4 \pm 0.4$ | $42.5 \pm 0.5$ | $84.2 \pm 0.3$ | 51.4 |
| Hard | $19.8 \pm 1.7$ | $39.7 \pm 1.0$ | $86.1 \pm 0.2$ | 48.5 |
| Herding | $24.9 \pm 1.6$ | $45.7 \pm 0.6$ | $86.8 \pm 0.4$ | 52.5 |
| Moderate | $24.0 \pm 1.8$ | $44.5 \pm 2.7$ | $86.1 \pm 1.3$ | 51.5 |
| GM Matching | $\mathbf{25.6 \pm 0.2}$ | $\mathbf{47.6 \pm 1.9}$ | $\mathbf{86.9 \pm 0.3}$ | **53.4** |
| **20% Label Noise** | | | | |
| Random | $18.0 \pm 2.4$ | $36.4 \pm 0.9$ | $75.5 \pm 0.7$ | 43.3 |
| Easy | $24.2 \pm 0.6$ | $40.7 \pm 1.1$ | $76.5 \pm 1.9$ | 47.1 |
| Hard | $13.1 \pm 1.9$ | $22.7 \pm 0.7$ | $67.2 \pm 0.5$ | 34.3 |
| Herding | $22.7 \pm 0.3$ | $38.5 \pm 1.5$ | $76.6 \pm 1.3$ | 45.9 |
| Moderate | $23.0 \pm 1.3$ | $39.8 \pm 1.3$ | $75.9 \pm 1.3$ | 46.2 |
| GM Matching | $\mathbf{26.0 \pm 0.9}$ | $\mathbf{41.1 \pm 1.8}$ | $\mathbf{77.8 \pm 0.4}$ | **48.3** |
| **40% Label Noise** | | | | |
| Random | $16.8 \pm 2.0$ | $28.3 \pm 2.2$ | $66.2 \pm 0.8$ | 37.1 |
| Easy | $22.5 \pm 1.5$ | $34.1 \pm 1.5$ | $70.5 \pm 1.1$ | 42.4 |
| Hard | $12.8 \pm 1.3$ | $16.5 \pm 1.6$ | $51.4 \pm 1.9$ | 26.9 |
| Herding | $18.0 \pm 1.4$ | $30.1 \pm 0.9$ | $65.1 \pm 1.4$ | 37.7 |
| Moderate | $20.2 \pm 1.3$ | $34.0 \pm 1.7$ | $67.8 \pm 1.5$ | 40.7 |
| **GM Matching** | $\mathbf{23.3 \pm 1.8}$ | $\mathbf{36.8 \pm 1.4}$ | $\mathbf{71.0 \pm 1.3}$ | **43.7** |

Table 13: NETWORK AND DOMAIN TRANSFER - PROXY ENCODER: A pretrained CLIP ViT-B/32 proxy encoder is used to find (10, 100, 1000) samples per class from CIFAR-10. Consequently, a ResNet-18 is trained on the selected subset. We perform the experiment across - clean; 20% and 40% Label Noise.

(Moderate, 40.7%). Notably, the advantage of GM MATCHING is particularly prominent at smaller subset sizes (e.g., 10 samples/class), suggesting that it is more effective at isolating informative, transferable samples even when the proxy encoder and downstream model differ significantly.

These findings highlight a key strength of GM MATCHING: its ability to select subsets that retain semantic structure and predictive utility, even when selected in a drastically different representation space than where the model is ultimately trained. By anchoring the selection process on the geometric median — a robust estimator that captures the core of the data distribution — GM MATCHING exhibits strong generalization under simultaneous domain and architecture shifts, making it a practical and principled choice for real-world deployment settings involving large-scale, black-box, or frozen proxy encoders.

### I.4. Unconditional Image Generation

To further validate our approach, we conduct experiments on an unconditional image generation task using a diffusion model. Specifically, we train a U-Net on the MNIST dataset, with Denoising Diffusion Probabilistic Models (DDPM) (Ho et al., 2020), which learns to generate images through a gradual denoising process. We perform sample selection with CLIP ViT-B/32 (Radford et al., 2021a) embeddings as demonstrated in Figure 19.

The fundamental idea behind diffusion models is to learn the reverse of a gradual noise corruption process applied to training images. This forward diffusion process progressively adds Gaussian noise to an image across a sequence of time-steps $t = 1, \ldots, T$, following the Markovian formulation:

$$q(x_t|x_{t-1}) = \mathcal{N}(x_t; \sqrt{1 - \beta_t} x_{t-1}, \beta_t I), \tag{70}$$

where, $x_t$ represents the image at timestep $t$ and $\beta_t \in (0, 1)$ denotes the noise variance at timestep $t$, following a predefined

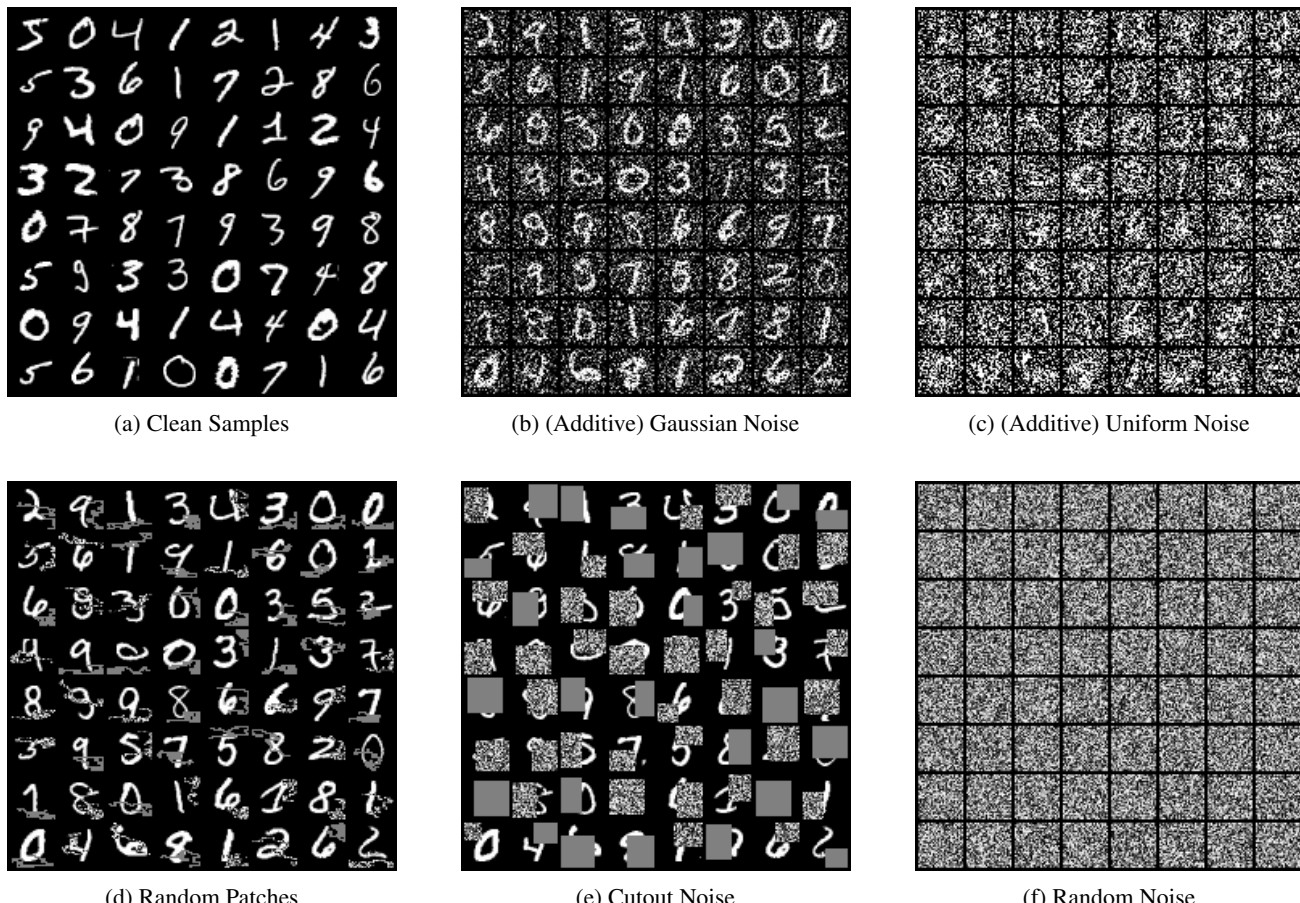

(a) Clean Samples      (b) (Additive) Gaussian Noise      (c) (Additive) Uniform Noise

(d) Random Patches      (e) Cutout Noise      (f) Random Noise

Figure 18: **Image Corruption:** Visualization of various noise types and sample categories. Each sub-figure demonstrates a distinct type of data corruption or clean sample used in the experiments.

schedule. The generative process aims to learn a function $p_\theta(x_{t-1}|x_t)$ that can reverse this diffusion, thereby reconstructing clean images from noise. The learned denoising function is parameterized using a deep neural network that predicts the noise component in a given sample, enabling the recovery of realistic image distributions.

To evaluate the quality of the generated samples, we use the Fréchet Inception Distance (FID), which measures the Wasserstein-2 distance between the feature distributions of real and generated images:

$$\text{FID} = \left\| \mu_r - \mu_g \right\|^2 + \text{Tr}\left( \Sigma_r + \Sigma_g - 2(\Sigma_r \Sigma_g)^{1/2} \right) \tag{71}$$

where $(\mu_r, \Sigma_r)$ and $(\mu_g, \Sigma_g)$ are the mean and covariance of the real and generated feature distributions respectively, and $\text{Tr}(\cdot)$ denotes the trace operator. Lower FID values indicate that the generated images more closely resemble the real data distribution.

### I.4.1. EXPERIMENTAL SETUP

For training stability and optimal convergence, we adopt specific hyperparameter settings. The batch size is set to 128 to ensure efficient mini-batch updates. The learning rate is fixed at $1 \times 10^{-4}$, tuned for stable convergence. We use the AdamW optimizer due to its adaptive learning rate properties and weight decay regularization. The number of diffusion time-steps is set to 1000, providing sufficient granularity for high-resolution generative refinement. A linear noise schedule is applied where $\beta_t$ increases linearly over time-steps, preventing abrupt changes in noise levels. To ensure robustness in our conclusions, we conduct multiple training runs with different random seeds, mitigating the impact of initialization biases

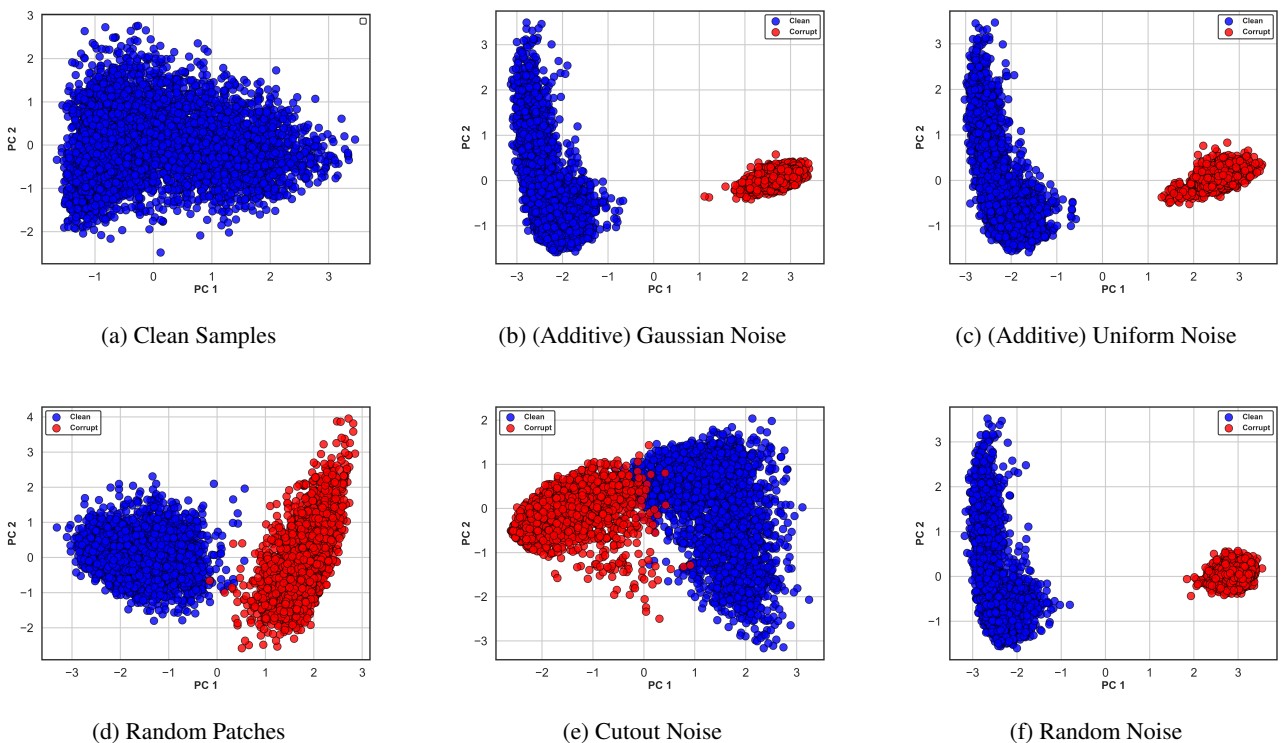

Figure 19: **(PROXY EMBEDDING SPACE)** t-SNE Visualization of CLIP ViT-B/32 Embeddings of a subset of MNIST images from Figure 18: (a) Clean (baseline), and where 45% samples corrupted with (b) Gaussian noise, (c) Uniform noise, (d) Random patches, (e) Cutout noise.

and training variability.

Our experiments involve running the model across multiple random seeds to ensure statistical robustness. Results are compared using FID scores to determine the effectiveness of different data selection strategies in training the diffusion model. We compare GM MATCHING against multiple baseline geometric subset selection strategies(Appendix I.1): Random, Easy, Hard, Moderate, and Kernel Herding.

To further stress-test these selection methods, we introduce structured perturbations into the training data, simulating realistic noise and adversarial conditions. These perturbations include Gaussian noise, which applies additive white noise pixel-wise; uniform noise, which perturbs pixel intensities randomly; random patches, which corrupt localized regions with random pixel values; cutout augmentations, which mask out rectangular sections of images; and completely random images, which introduce purely random noise samples into training batches. Visual examples of these corrupted data samples are presented in Figure 18.

### I.4.2. DISCUSSION

Figure 6 depicts the generative performance of the diffusion model when trained on the selected subset. Evidently, GM MATCHING consistently achieves FID scores across all subset sizes and corruption scenarios, clearly demonstrating its superior performance in terms of generating high-quality samples. *In the clean (no corruption) setting*, at smaller subset sizes (10%-20%), GM MATCHING substantially outperforms the baseline methods. While all methods eventually converge to similar performance levels at full dataset size (100%), the early advantage of GM MATCHING highlights its efficiency in selecting representative subsets. *Under moderate (20%)corruption*, GM MATCHING maintains a significant performance advantage, especially at smaller subset sizes (10%-40%). This result suggests that GM MATCHING effectively identifies high-quality training subsets even in the presence of moderately corrupted data, thereby preserving generative quality. *At higher (30%)corruption rates*, GM MATCHING demonstrates remarkable robustness against higher corruption levels. In contrast, methods such as "Hard" sampling significantly degrade in performance due to sensitivity to corrupted data.

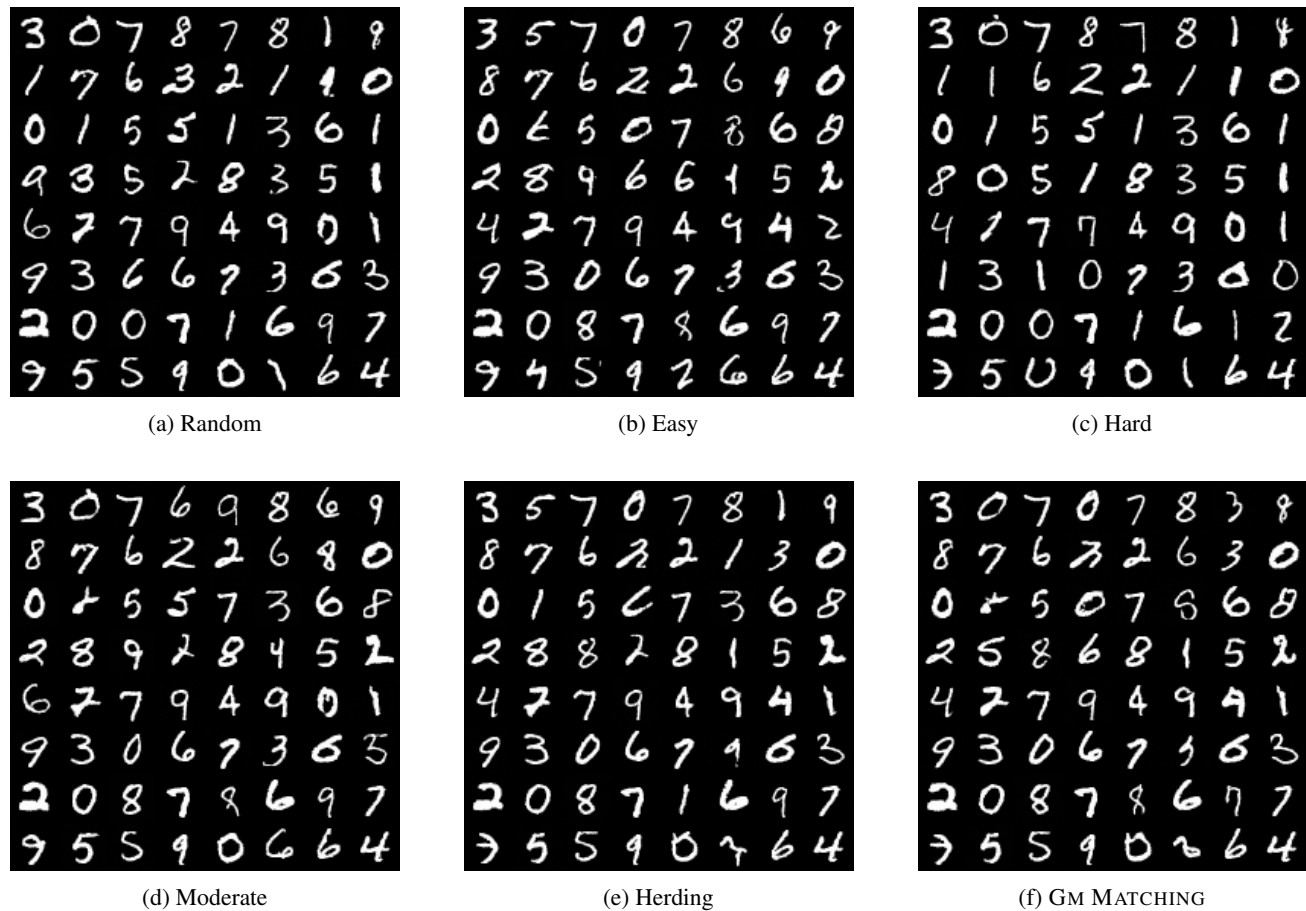

(a) Random        (b) Easy        (c) Hard

(d) Moderate        (e) Herding        (f) GM MATCHING

Figure 20: (No Corruption) Visualization of Generated Samples 40% sampling

GM MATCHING, however, remains stable and consistently achieves superior FID scores, emphasizing its robustness in more challenging scenarios.

Visual analyses provide additional insights into the effectiveness of GM Matching, particularly when comparing subsets of clean versus noisy training data. In Figure 20, we present visualizations of generated samples obtained from the UNet trained via DDPM on a 40% subset of MNIST. All subset selection methods produce visually comparable and clear digit images under these clean conditions, demonstrating that, in an ideal scenario without data corruption, the impact of different subset selection methods on visual quality is minimal. However, Figure 21 highlights significant differences in generated samples when the same model is trained on a 40% subset derived from data subjected to severe corruption (30%). Baseline methods, such as Random, Easy, and Hard selection, result in samples with substantial visual distortion and ambiguity. In contrast, GM MATCHING notably produces clear and well-defined digit representations, emphasizing its superior robustness and capability to effectively mitigate the adverse effects of corrupted data during training. Collectively, these visual results reinforce our quantitative findings (Figure 6), clearly underscoring the suitability and robustness of GM MATCHING in challenging data corruption scenarios for training diffusion models.

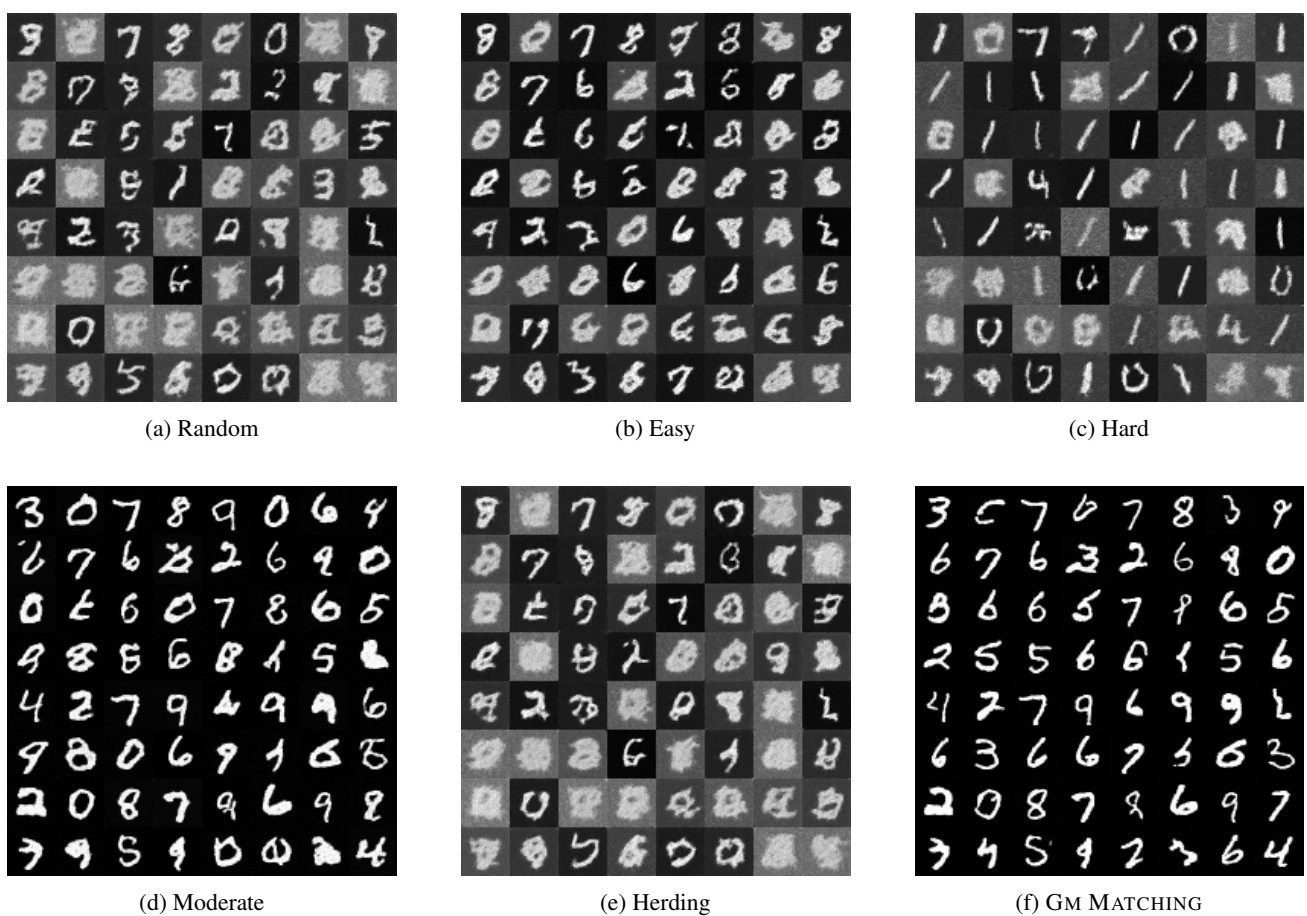

(a) Random (b) Easy (c) Hard

(d) Moderate (e) Herding (f) GM MATCHING

Figure 21: (30% Corruption) Visualization of Generated Samples 40% sampling

