# OpenReview forum: "Geometric Median (GM) Matching for Robust k-Subset Selection from Noisy Data"
_ICML.cc/2025/Conference — ICML 2025 poster_

### Official Review · Reviewer_C6QM · 2025-03-13

**Overall Recommendation:** 3

**Summary:**

The paper proposes the use of the Geometric Median to robustly identify subsets of a dataset that best represents the full dataset. The main goal is to reduce sensitivity of the selection algorithm to outliers in the corrupted data, a problem that traditional subset selectors that rely on the empirical mean suffer from. In this regard, the geometric mean is proposed as more robust surrogate. An iterative selection algorithm is proposed that selects subsets that minimizes the discrepancy between the subset and the geometric mean. Theoretical guarantees are given on the convergence properties of the proposed algorithm.

**Claims And Evidence:**

The claims made in the paper are well supported in theoretical and empirical analysis.

**Essential References Not Discussed:**

N/A

**Experimental Designs Or Analyses:**

The experimental setup appears standard for the problem domain.

**Methods And Evaluation Criteria:**

The evaluation criteria is standard for the subset selection problem.

**Other Comments Or Suggestions:**

The captions of Figure 6 are obscured by vspacing.

**Other Strengths And Weaknesses:**

The main strength of the proposed method is its extension of subset selection methods using the GM, which is well grounded in theory and in so doing , answers the question of how to obtain subset selector that are robust to outliers. On the other side, one may point out that the main contribution is the substitution of the empirical mean with the geometric mean in an already existing subset selection method.

**Questions For Authors:**

- How well does the proposed method perform when the corruption rate is larger than 20% in table 2?

- The current experiments are performed on small scale datasets. How does the proposed method scale when applied on larger datasets?

**Relation To Broader Scientific Literature:**

The paper builds on classical results in robust estimation and by utilizing the Geometric Mean, extend these workds into the current subset selection domain.

**Theoretical Claims:**

I do not find any issues with the proofs of the theoretical claims in the paper, although i should stress that i only did a cursory pass over the proofs.

---

> ### Author Rebuttal · Authors · 2025-04-01
>
> We thank the reviewer for their thoughtful and detailed feedback. We are encouraged that the reviewer finds the theoretical claims well supported, the empirical design appropriate, and the core contribution valuable.
> Below, we address the specific points raised:
>
> **Q1: Performance at Higher Corruption Rates (>20%) in Table 2**
>
> Thank you for raising this. We now include experiments with 30% image corruption across two datasets (CIFAR-100 and TinyImageNet), at 20% and 30% selection rate:
>
> | Method / Ratio       | C(20%)     | C(30%)     | T(20%)  | T(30%)  |
> |----------------------|-------------------|--------------------|--------------------|--------------------|
> | Random               | 24.51 ± 1.34       | 32.26 ± 0.81       | 14.64 ± 0.29       | 19.41 ± 0.45       |
> | Herding              | 29.42 ± 1.54       | 37.50 ± 2.12       | 15.14 ± 0.45       | 20.19 ± 0.45       |
> | Forgetting           | 29.48 ± 1.98   | 38.01 ± 2.11       | 11.25 ± 0.90       | 17.07 ± 0.66       |
> | GraNd-score          | 23.03 ± 1.15       | 34.83 ± 1.22       | 13.68 ± 0.42       | 15.40 ± 0.52       |
> | EL2N-score           | 21.95 ± 1.08       | 31.63 ± 2.84       | 10.11 ± 0.25       | 13.52 ± 0.32       |
> | Optimization-based   | 26.77 ± 1.05       | 35.63 ± 0.63       | 13.20 ± 0.56       | 18.52 ± 0.33       |
> | Self-sup.-selection  | 23.12 ± 1.47       | 34.85 ± 0.68       | 11.23 ± 0.32       | 17.76 ± 0.69       |
> | Moderate-DS          | 28.45 ± 0.53       | 36.55 ± 1.26       | 15.27 ± 0.38   | 20.33 ± 0.28       |
> | **GM Matching**      | **40.28 ± 1.02**   | **50.71 ± 0.86**   | **20.38 ± 1.09**   | **25.93 ± 0.98**   |
>
> GM Matching, significantly outperforms all baselines — the gains are even more pronounced compared to milder corruption (Tables 2, 4, 5), further validating the robustness of our approach.
>
> **Q2. Scaling on larger datasets?**
>
> We now include results ( Test Accuracy @ Top-5) on ImageNet-1k (1.2M training samples), using ResNet-50 and selection ratios from 60% to 90%:
>
> | Method / Ratio       | 60%             | 70%             | 80%             | 90%             |
> |----------------------|-----------------|------------------|------------------|------------------|
> | Random               | 87.91 ± 0.37     | 88.63 ± 0.95     | 89.52 ± 0.73     | 89.57 ± 0.60     |
> | Herding              | 88.25 ± 2.16     | 88.81 ± 1.06     | 89.60 ± 0.58     | 90.41 ± 0.33     |
> | Forgetting           | 88.83 ± 0.92     | 89.81 ± 0.97     | 89.94 ± 0.26     | 90.41 ± 0.58     |
> | GraNd-score          | 88.48 ± 1.73     | 89.82 ± 2.07     | 90.24 ± 0.81     | 90.41 ± 0.62     |
> | EL2N-score           | 88.48 ± 2.81     | 89.82 ± 1.14     | 90.34 ± 0.87     | 90.46 ± 0.96     |
> | Self-sup.-selection  | 87.59 ± 2.61     | 89.56 ± 1.97     | 90.74 ± 0.27     | 90.49 ± 0.98     |
> | Moderate-DS          | 89.23 ± 0.96     | 89.94 ± 0.74     | 90.65 ± 0.51     | 90.75 ± 0.35     |
> | **GM Matching**      | **90.28 ± 0.38** | **90.54 ± 0.19** | **90.72 ± 0.26** | **90.84 ± 0.32** |
>
> GM Matching achieves the best performance across all pruning levels — outperforming all baselines. Beyond empirical scalability, kindly also refer to Section 5.4 and Appendix F, where we break down the time complexity of both GM estimation and greedy selection and motivating the batched variant (Algorithm 1). Figure 9-11 show wall-clock time scaling vs. dataset size, embedding dim, and batch size.
>
> **Note:** For the new experiments in Q1 and Q2, we follow the setup and reuse baselines from Moderate Coreset (ICLR 2023). Code: https://github.com/tmllab/2023_ICLR_Moderate-DS
>
> **Q3. Contribution**
>
> We thank the reviewer for acknowledging the strength of our theoretical grounding and robustness to outliers. We respectfully clarify that our contribution goes well beyond a simple substitution: Fundamentally, we propose a new combinatorial formulation for Robust Moment Matching, enabling systematic study of subset selection in noisy settings. While we instantiate our method with the GM, the framework supports a wide class of robust estimators (e.g., trimmed means, M-estimators), opening a principled direction in robust subset selection under noise. This formulation is fundamental, as it decouples selection from fragile mean estimation and instead aligns with a robust signal. Moreover, since the framework applies to general Hilbert spaces, it can extend to gradient space, enabling integration with methods like CRAIG that perform gradient matching for coreset selection. In summary, we present a general, theoretically grounded framework for robust coreset selection — not limited to a single estimator or modality — paving the way for future advances in robust data summarization.
>
> **Q4. Figure 6 formatting**
>
> Thank you for pointing this out. We will correct the spacing issue in the camera-ready version.
>
> We are grateful for the reviewer’s insights and look forward to constructive discussion and refining the paper accordingly.

---

### Official Review · Reviewer_7RaC · 2025-03-14

**Overall Recommendation:** 4

**Summary:**

The paper introduces Geometric Median (GM) Matching, a novel approach for robust k-subset selection from noisy datasets. The key contribution is replacing the empirical mean, which is sensitive to outliers, with the Geometric Median (GM), a robust estimator with an optimal breakdown point of 1/2. The GM Matching algorithm iteratively selects a subset whose mean approximates the GM of the potentially noisy dataset. Theoretical guarantees demonstrate that GM Matching achieves $O(1/k)$ scaling, outperforming traditional $O(1/\sqrt{k})$ scaling of uniform sampling, even under high corruption. Extensive experiments on image classification and generation tasks show that GM Matching significantly outperforms existing pruning methods, particularly in high-corruption settings, making it a strong baseline for robust data pruning.

--------------- Updated ------------
Thank you for your response, based on other reviews and response, I feel confident in my rating of accept.

**Claims And Evidence:**

The paper makes several claims, which are mostly well-supported by theoretical analysis and empirical validation:

* GM Matching is robust under high corruption rates: Theoretical guarantees (Theorem 1) prove that GM Matching remains stable even when up to 50% of data is arbitrarily corrupted. The experiments on CIFAR-100 and Tiny ImageNet confirm that GM Matching consistently outperforms other selection methods in corrupted environments.
* O(1/k) convergence rate: The authors provide a mathematical proof that GM Matching converges at a quadratic improvement over uniform sampling. Empirical results (Fig. 3) support this claim, showing GM Matching achieving better moment matching error than herding and random sampling.
* Superior performance in real-world scenarios: The experiments across multiple datasets (Tables 1, 2, 3) demonstrate that GM Matching consistently outperforms alternatives in both clean and noisy settings.

**Essential References Not Discussed:**

Garg et al. (2023): This work discusses low-loss curvature and its impact on data efficiency, which could strengthen the theoretical backing of GM Matching. While not directly related to robustness, it is worth including in Related Work.

[1] Garg, Isha, and Kaushik Roy. "Samples with low loss curvature improve data efficiency." In Proceedings of the IEEE/CVF Conference on Computer Vision and Pattern Recognition, pp. 20290-20300. 2023.

**Experimental Designs Or Analyses:**

The experimental design is solid, with extensive comparisons across different datasets, architectures, and corruption settings. The main concern is the use of FID for diffusion model evaluation. While acceptable in this case, the authors should consider moving to more robust generative evaluation metrics such as those proposed by Stein et al. (2023) and Jayasumana et al. (2024).

[2] Stein, George, Jesse Cresswell, Rasa Hosseinzadeh, Yi Sui, Brendan Ross, Valentin Villecroze, Zhaoyan Liu, Anthony L. Caterini, Eric Taylor, and Gabriel Loaiza-Ganem. "Exposing flaws of generative model evaluation metrics and their unfair treatment of diffusion models." Advances in Neural Information Processing Systems 36 (2023): 3732-3784.

[3] Jayasumana, Sadeep, Srikumar Ramalingam, Andreas Veit, Daniel Glasner, Ayan Chakrabarti, and Sanjiv Kumar. "Rethinking fid: Towards a better evaluation metric for image generation." In Proceedings of the IEEE/CVF Conference on Computer Vision and Pattern Recognition, pp. 9307-9315. 2024.

**Methods And Evaluation Criteria:**

The proposed method makes sense given the problem at hand, as it directly addresses the weaknesses of empirical mean-based selection methods in noisy datasets. The evaluation is mostly robust, with well-chosen benchmarks (CIFAR-100, Tiny ImageNet, MNIST), pruning ratios (20%-100%), and corruption scenarios (label noise, feature corruption, adversarial attacks).

One minor issue: The baseline comparison might be missing Garg et al. (2023) for core set selection [1]. While Garg et al. is not explicitly designed for noisy datasets, this work could be relevant and should be included in Related Work.


[1] Garg, Isha, and Kaushik Roy. "Samples with low loss curvature improve data efficiency." In Proceedings of the IEEE/CVF Conference on Computer Vision and Pattern Recognition, pp. 20290-20300. 2023.

**Other Comments Or Suggestions:**

* Add adversarial attack parameters: Without this information, it is hard to replicate the results.
* Clarify when to use GM Matching: The paper should discuss specific conditions under which GM Matching is preferable over other methods (e.g., what levels of corruption?).

**Other Strengths And Weaknesses:**

Strengths:

* Strong theoretical guarantees: The breakdown point of 1/2 makes GM Matching highly robust compared to mean-based approaches.
* Extensive experiments: The authors test multiple datasets, architectures, and corruption types, making the results more generalizable.
* Practical relevance: Given the increasing interest in data pruning and robust training, GM Matching is a strong baseline for future research.

Weaknesses:

*  Forgetting method performance needs more discussion: The results indicate that Forgetting performs better at higher corruption rates, but the paper does not fully explain why. A more detailed discussion of when to use GM Matching vs. other approaches would be useful.
* Under feature corruption, when should GM Matching be used versus other methods? This should be more explicitly stated.
* Adversarial attack settings are unclear: The paper does not provide details on PGD attack parameters—e.g., what was the epsilon bound? How many iterations were used? This should be stated in the main paper or appendix.

**Questions For Authors:**

1. What were the specific attack parameters? Please provide details on the epsilon bound and number of iterations.

2. Why does Forgetting perform better at higher corruption rates? The paper suggests GM Matching is robust, yet Forgetting seems to outperform it at low pruning rates. Why is this the case? Some discussion would be helpful.

**Relation To Broader Scientific Literature:**

The work builds upon robust statistics and subset selection techniques, integrating Geometric Median estimation into a scalable data pruning method. The key contribution is applying GM to data selection for deep learning robustness, which is novel and well-grounded in theoretical and empirical research. GM Matching provides a formal and solid approach to robustness against label noise and feature corruption.

**Theoretical Claims:**

I checked the correctness of the theoretical proofs, and they seem sound. The key theoretical claim (GM Matching achieves O(1/k) convergence under gross corruption) is well-supported by rigorous mathematical derivations and empirical validation. While low-level details may have been missed, the results seem correct at the high level.

---

> ### Author Rebuttal · Authors · 2025-04-01
>
> We sincerely thank the reviewer for the detailed, thoughtful, and encouraging feedback. We are pleased that you found our theoretical results, empirical validation, and overall framing of GM Matching to be strong contributions. We respond to your key points below:
>
> **Q1. Missing Related Work**
>
> Thank you for pointing this out. Garg et al. (2023) presents an interesting and well-motivated approach based on low-loss curvature, which offers valuable insights into data efficiency and sample selection. While it is not explicitly designed for noisy settings, we agree that it is a promising direction for future work, particularly in exploring whether curvature-based selection methods can be made robust to corruption. We will cite and briefly discuss this work in the revised Related Work section.
>
> **Q2. Evaluation Metrics for Diffusion**
>
> We agree that FID has known limitations, especially when evaluating modern diffusion models. In our case, we used FID following standard practice for proof-of-concept (PoC) experiments on simple generative setups, and we appreciate that the reviewer finds this acceptable. That said, we thank the reviewer for pointing us to Stein et al. (2023) and Jayasumana et al. (2024), and we will reference both in the revised manuscript.
>
> **Q3. Adversarial attack parameters:**
>
> We appreciate the reviewer’s careful attention to reproducibility and thank you for this valuable feedback. Clarifying attack settings is indeed essential for replicating adversarial corruption experiments.
>
> As mentioned in the Experiments Section, our setup is identical to Moderate Coreset [ICLR 23]. The code is available at: https://github.com/tmllab/2023_ICLR_Moderate-DS .
>
> Specifically,  In the adversarial corruption experiments, we used Projected Gradient Descent (PGD) as implemented in torchattacks, with the following parameters: $\epsilon$ = 8/255, step size  = 2/255, number of iterations = 10, and random start = True. These are standard settings in the robust training literature (e.g., Madry et al., 2018). Additionally, we used Gradient Sign Attack (GSA)  (Goodfellow et al., 2014) via advertorch, with the same $\epsilon$ = 8/255. Adversarial examples were generated using a pretrained ResNet-50 on CIFAR-100 with standard normalization.
>
> We will ensure these details are clearly stated in the revised manuscript - Appendix.
>
> Furthermore, we will release the code of GM Matching for reproducibility.
>
> **Q4.. Discussion on forgetting**
>
> We appreciate the reviewer highlighting this nuanced behavior.
>
> Forgetting-based methods, which leverage early training dynamics, can perform well in clean or mildly corrupted settings at high retain ratios (e.g., 80–90%), where noise has limited impact. In such regimes, GM Matching may be modestly disadvantaged due to the bias introduced by the geometric median, while Forgetting can surface informative, hard-to-learn examples.
> However, Forgetting is known to be sensitive to corruption, unstable across architectures, and often requires careful tuning and access to full training traces. In contrast, GM Matching is simple, tuning-free, and inherently robust—making it particularly effective in severe or mixed corruption scenarios.
>
> We will expand on this comparison in the revised manuscript and include a discussion highlighting when each method is most appropriate.
>
> **Q5. Settings where GM Matching is preferable.**
>
> We thank the reviewer for this thoughtful question.
> Theoretically, in clean settings, at low pruning rates: the GM is a biased estimator of the uncorrupted mean, and as such, GM Matching may perform similar to or slightly worse than algorithms that leverage informative or hard examples—e.g., those with high loss, large gradient norms, or that are frequently forgotten—can be more valuable, making dynamic or score-based methods (e.g., Forgetting, EL2N) highly competitive.
>
> However, GM Matching has strong advantage in two key regimes:
>
>  (1) Clean settings with high pruning rates, where its $\mathcal{O}(1/k)$ convergence offers a sharp advantage over random sampling (a very strong baseline); and
>
>  (2) Noisy settings, where its robustness to outliers leads to consistent improvements over dynamic or score-based methods, which often degrade under corruption.
>
> Even on real-world datasets without synthetic noise (e.g., CIFAR-100, TinyImageNet, ImageNet-1k), we observe consistent gains, underscoring its practical relevance—making it a robust, tuning-free default across varied conditions.
>
> We will include this discussion in our revised manuscript.
>
> We appreciate the thoughtful comments and suggestions, and we look forward to continued dialogue to further strengthen the work.

---

> > ### Comment · Reviewer_7RaC · 2025-04-03
> >
> > Thank you for your response, based on other reviews and response, I feel confident in my rating of accept.

---

> > > ### Author Response · Authors · 2025-04-04
> > >
> > > Thank you again for your thoughtful review and for actively and promptly engaging with our rebuttal.
> > > Your feedback helped us strengthen the paper, and we appreciate your support and confidence in our work.

---

### Official Review · Reviewer_1B9r · 2025-03-17

**Overall Recommendation:** 2

**Summary:**

This paper proposes a dataset pruning method with subset selection. The proposed method utilizes the geometric median moment matching allowing a small amount of an approximation error. Also, the authors provide the theoretical guarantee that the proposed geometric median moment matching leads to a good approximation of the mean. Finally, several experiments on the benchmark datasets are conducted to demonstrate the efficacy of the proposed method on the clean dataset as well as noisy data.

**Claims And Evidence:**

The author's claims are clear.

**Essential References Not Discussed:**

-

**Experimental Designs Or Analyses:**

The experiments are conducted based on the previous literature.

**Methods And Evaluation Criteria:**

The proposed methods seem to make sense.

**Other Comments Or Suggestions:**

- The terminology $k$-subset selection might be misunderstood as choosing a few handful number of data instances, while the authors are selecting a portion of the dataset.
- Regarding that, what happens to the experimental result if the authors set $k=10, 100, 1000$, for example in the CIFAR case?

**Other Strengths And Weaknesses:**

- The paper is clearly written, and easy to follow.
- Extensive experiments are conducted to demonstrate the superiority of the proposed method.
- The DDPM data generation experiment on the MNIST dataset is too simple. I am aware that training the diffusion model requires a long training time, but it would be more persuasive if the authors utilized complex datasets for training the generative model.

**Questions For Authors:**

- What was the intuition behind utilizing the geometric median moment matching?

**Relation To Broader Scientific Literature:**

-

**Theoretical Claims:**

I haven't read the proof of the proposed theorem, but it seems to be straightforward maths.

---

> ### Author Rebuttal · Authors · 2025-04-01
>
> **Q1. The terminology k-subset selection.**
>
> We appreciate the reviewer’s observation regarding potential ambiguity in the term k-subset selection. Our usage follows common conventions in the subset selection and coreset literature, where k typically denotes the size of the selected subset—either as an absolute number or as a fraction of the dataset i.e., $k = \rho n $ for some  $\rho \in (0,1)$ interchangeably.
> Importantly, this terminology is not only standard but also theoretically grounded in our work. Our results are stated in terms of selecting k samples from a dataset of size n, and the error bounds naturally depend on k (e.g. Theorem 1).
> Thus, we feel our formulation justifies the use of k-subset selection both terminologically and theoretically.
>
> However, based on your suggestion, we will clarify this distinction explicitly in the revised manuscript to avoid any confusion.
>
> **Q2. Performance for small fixed subset sizes (k=10,100,1000)**
>
> We thank the reviewer for this valuable suggestion.
>
> We conduct additional experiments on CIFAR-10 using fixed subset sizes 10, 100, 1000 samples per class  under three settings: clean, 20% label noise, and 40% label noise. A CLIP ViT-B/32 proxy encoder is used to select subsets, and a ResNet-18 is trained from scratch on the pruned data.
>
> | **Method / Subset** | **10** | **100** | **1000** | **Mean ↑** |
> |---------------------|--------|---------|----------|------------|
> | **No Corruption** |||||
> | Random              | 23.2 ± 3.9 | 42.4 ± 2.3 | 86.8 ± 0.7 | 50.8 |
> | Easy                | 27.4 ± 0.4 | 42.5 ± 0.5 | 84.2 ± 0.3 | 51.4 |
> | Hard                | 19.8 ± 1.7 | 39.7 ± 1.0 | 86.1 ± 0.2 | 48.5 |
> | Herding             | 24.9 ± 1.6 | 45.7 ± 0.6 | 86.8 ± 0.4 | 52.5 |
> | Moderate            | 24.0 ± 1.8 | 44.5 ± 2.7 | 86.1 ± 1.3 | 51.5 |
> | **GM Matching**     | **25.6 ± 0.2** | **47.6 ± 1.9** | **86.9 ± 0.3** | **53.4** |
> | **20% Label Noise** |||||
> | Random              | 18.0 ± 2.4 | 36.4 ± 0.9 | 75.5 ± 0.7 | 43.3 |
> | Easy                | 24.2 ± 0.6 | 40.7 ± 1.1 | 76.5 ± 1.9 | 47.1 |
> | Hard                | 13.1 ± 1.9 | 22.7 ± 0.7 | 67.2 ± 0.5 | 34.3 |
> | Herding             | 22.7 ± 0.3 | 38.5 ± 1.5 | 76.6 ± 1.3 | 45.9 |
> | Moderate            | 23.0 ± 1.3 | 39.8 ± 1.3 | 75.9 ± 1.3 | 46.2 |
> | **GM Matching**     | **26.0 ± 0.9** | **41.1 ± 1.8** | **77.8 ± 0.4** | **48.3** |
> | **40% Label Noise** |||||
> | Random              | 16.8 ± 2.0 | 28.3 ± 2.2 | 66.2 ± 0.8 | 37.1 |
> | Easy                | 22.5 ± 1.5 | 34.1 ± 1.5 | 70.5 ± 1.1 | 42.4 |
> | Hard                | 12.8 ± 1.3 | 16.5 ± 1.6 | 51.4 ± 1.9 | 26.9 |
> | Herding             | 18.0 ± 1.4 | 30.1 ± 0.9 | 65.1 ± 1.4 | 37.7 |
> | Moderate            | 20.2 ± 1.3 | 34.0 ± 1.7 | 67.8 ± 1.5 | 40.7 |
> | **GM Matching**     | **23.3 ± 1.8** | **36.8 ± 1.4** | **71.0 ± 1.3** | **43.7** |
>
> As evident, GM Matching remains consistently strong even under extreme data reduction, particularly in noisy settings. This further supports the versatility of the method across the spectrum of subset sizes.
>
> **Q3. Simplicity of DDPM exp**
>
> We agree that MNIST is a relatively simple dataset. Our goal with this experiment was to provide a proof of concept (PoC) demonstrating the applicability of GM Matching to generative modeling — specifically, to show that high-quality subsets selected via our method improve generation quality, even under severe corruption. This controlled setup was intentionally chosen to isolate the effect of subset selection, avoiding confounding factors from architectural or optimization complexity.  We believe this early result is a meaningful step in connecting robust selection to generative tasks. A thorough investigation of GM Matching in diffusion models on more complex datasets (e.g., CIFAR-10, CelebA) and with modern backbones (e.g., ADM) is an exciting direction for future work, as is exploring its applicability to LLMs.
>
> **Q4. Intuition behind GM**
>
> Great question — thank you for the opportunity to clarify.
> At the core of our approach is the observation that the empirical mean is highly sensitive to outliers and corrupted data, which can significantly distort subset selection. In contrast, the geometric median has a breakdown point of 50%, making it a far more robust estimate of central tendency in noisy settings. This intuition leads naturally to our Robust Moment Matching formulation: instead of chasing the (fragile) dataset mean, we align with a robust estimator that resists the influence of corrupted or adversarial data points. This makes the selection process resilient across a variety of noise settings — as our theory and experiments confirm.
>
> We thank the reviewer once again and look forward to engaging further in discussion in strengthening the paper.

---

### Decision · Program_Chairs · 2025-05-01

**Decision:**

Accept (poster)

**Comment:**

The paper proposes Geometric Median (GM) Matching, a robust method for k-subset selection from noisy data. By replacing the outlier-sensitive empirical mean with the GM, which has an optimal breakdown point of 1/2, the algorithm selects subsets whose means approximate the GM. It achieves an improved theoretical error scaling of $O(1/k)$, surpassing the $O(1/\sqrt{k})$ of uniform sampling, even under heavy corruption. Experiments on image classification and generation confirm its superior performance over existing pruning methods, establishing it as a strong baseline for robust data selection. The majority of reviewers appreciates its novelty and soundness of the paper, especially its theoretical contribution. Although one reviewer raised concerns about the parameters used in the CIFAR-10 experiment, I find the rebuttal sufficiently convincing. As a side note, while the reviewer found the term ``k-subset'' potentially misleading, I consider it standard terminology in the community. Therefore, I recommend accepting the paper.